# Growing Tiny Networks: Spotting Expressivity Bottlenecks and Fixing Them Optimally

## Abstract

Machine learning tasks are generally formulated as optimization problems, where one searches for an optimal function within a certain functional space. In practice, parameterized functional spaces are considered, in order to be able to perform gradient descent. Typically, a neural network architecture is chosen and fixed, and its parameters (connection weights) are optimized, yielding an architecture-dependent result. This way of proceeding however forces the evolution of the function during training to lie within the realm of what is expressible with the chosen architecture, and prevents any optimization across architectures. Costly architectural hyper-parameter optimization is often performed to compensate for this. Instead, we propose to adapt the architecture on the fly during training.

We show that the information about desirable architectural changes, due to expressivity bottlenecks when attempting to follow the functional gradient, can be extracted from backpropagation. To do this, we propose a mathematical definition of expressivity bottlenecks, which enables us to detect, quantify and solve them while training, by adding suitable neurons when and where needed. Thus, while the standard approach requires large networks, in terms of number of neurons per layer, for expressivity and optimization reasons, we are able to start with very small neural networks and let them grow appropriately. As a proof of concept, we show results on the CIFAR dataset, matching large neural network accuracy, with competitive training time, while removing the need for standard architectural hyper-parameter search.

## 1 Introduction

**Issues with the fixed-architecture paradigm.**    Universal approximation theorems such as (Hornik et al., 1989; Cybenko, 1989) are historically among the first theoretical results obtained on neural networks, stating the family of neural networks with arbitrary width as a good candidate for a parameterized space of functions to be used in machine learning. However the current common practice in neural network training consists in choosing a fixed architecture, and training it, without any possible architecture modification meanwhile. This inconveniently prevents the direct application of these universal approximation theorems, as expressivity bottlenecks that might arise in a given layer during training will not be able to be fixed. There are two approaches to circumvent this in daily practice. Either one chooses a (very) large width, to be sure to avoid expressivity and optimization issues (Hanin & Rolnick, 2019b; Raghu et al., 2017), to the cost of extra computational power consumption for training and applying such big models; to mitigate this cost, model reduction techniques are often used afterwards, using pruning, tensor factorization, quantization (Louizos et al., 2017) or distillation (Hinton et al., 2015). Or one tries different architectures and keeps the most suitable one (in terms of performance-size compromise for instance), which multiplies the computational burden by the number of trials. This latter approach relates to the Auto-DeepLearning field (Liu et al., 2020), where different exploration strategies over the space of architecture hyper-parameters (among other ones) have been tested, including reinforcement learning (Baker et al., 2017; Zoph & Le, 2016), Bayesian optimization techniques (Mendoza et al., 2016), and evolutionary approaches (Miller et al., 1989; Stanley et al., 2009; Miikkulainen et al., 2017; Bennet et al., 2021), that all rely on random tries and consequently take time for exploration. Within that line, Net2Net (Chen et al., 2015), AdaptNet (Yang et al., 2018) and MorphNet (Gordon et al., 2018) propose different strategies to explore possible variations of a given architecture, possibly guided by model size con-

straints. Instead, we aim at providing a way to locate precisely expressivity bottlenecks in a trained network, which might speed up neural architecture search significantly. Moreover, based on such observations, we aim at modifying the architecture *on the fly* during training, in a single run (no re-training), using first-order derivatives only, while avoiding neuron redundancy. Related work on architecture adaptation while training includes probabilistic edges (Liu et al., 2019) or sparsifying priors (Wolinski et al., 2020). Yet the training is done on the largest architecture allowed, which is resource-consuming. On the opposite we aim at starting from the simplest architecture possible.

**Optimization properties.** An important reason for common practice to choose wide architectures is the associated optimization properties: sufficiently larger networks are proved theoretically and shown empirically to be better optimized than small ones (Jacot et al., 2018). Typically, small networks exhibit issues with spurious local minima, while wide ones find good nearly-global minima. One of our goals is to train small networks without suffering from such optimization difficulties.

**Neural architecture growth.** A related line of work consists in growing networks neuron by neuron, by iteratively estimating the best possible neurons to add, according to a certain criterion. For instance, approaches such as (Wu et al., 2019) or Firefly (Wu et al., 2020) aim at escaping local minima by adding neurons that minimize the loss under neighborhood constraints. These neurons are found by gradient descent or by solving quadratic problems involving second-order derivatives. Other approaches (Causse et al., 2019; Bashtova et al., 2022), including GradMax (Evci et al., 2022), seek to minimize the loss as fast as possible and involve another quadratic problem. However the neurons added by these approaches are possibly redundant with existing neurons, especially if one does not wait for training convergence to a local minimum (which is time consuming) before adding neurons, therefore producing larger-than-needed architectures.

**Redundancy.** To our knowledge, the only approach tackling redundancy in neural architecture growth adds random neurons that are orthogonal in some sense to the ones already present (Maile et al., 2022). More precisely, the new neurons are picked within the *kernel* (preimage of $\{0\}$) of an application describing already existing neurons. Two such applications are proposed, respectively the matrix of fan-in weights and the pre-activation matrix, yielding two different notions of orthogonality. The latter formulation is close to the one of GradMax, in that both study first-order loss variations and use the same pre-activation matrix, with an important difference though: GradMax optimally decreases the loss without caring about redundancy, while the other one avoids redundancy but picks random directions instead of optimal ones. In this paper we bridge the gap between these two approaches, picking optimal directions that avoid redundancy in the pre-activation space.

**Notions of expressivity.** Several concepts of expressivity or complexity exist in the Machine Learning literature, ranging from Vapnik-Chervonenkis dimension (Vapnik & Chervonenkis, 1971) and Rademacher complexity (Koltchinskii, 2001) to the number of pieces in a piecewise affine function (as networks with ReLU activations are) (Serra et al., 2018; Hanin & Rolnick, 2019a). Bottlenecks have been also studied from the point of view of Information Theory, through mutual information between the activities of different layers (Tishby & Zaslavsky, 2015; Dai et al., 2018); this quantity is difficult to estimate though. Also relevant and from Information Theory, the Minimum Description Length paradigm and Kolmogorov complexity (Kolmogorov, 1965; Li et al., 2008) enable to define trade-offs between performance and model complexity.

In this article, we aim at measuring lacks of expressivity as the difference between what the backpropagation asks for and what can be done by a small parameter update (such as a gradient step), that is, between the desired variation for each activation in each layer (for each sample) and the best one that can be realized by a parameter update. Intuitively, differences arise when a layer does not have sufficient expressive power to realize the desired variation. Our main contributions are that we:

- take a functional analysis viewpoint over gradient descent on neural networks, suggesting to attempt to follow the functional gradient. We optimize not only the weights of the current architecture, but also the architecture itself on the fly, in order to progressively move towards more suitable parameterized functional spaces. This removes the optimization issues (local minima) that are due to thin architectures;

- properly define and quantify the notion of expressivity bottlenecks, globally at the neural network output as well as at each layer, and this in an easily computable way. This allows to localize expressivity bottlenecks in a neural network;

- mathematically define the best possible neurons to add to a given layer to decrease lacks of expressivity as a quadratic problem; compute them and their associated expressivity gain;

- automatically adapt the architecture to the task at hand by making it grow where needed, and this in a single run, in competitive computational complexity with respect to classically training a large model just once. To remove any need for layer width hyper-optimization, one could define a target accuracy and stop adding neurons when it is reached.

## 2 MAIN CONCEPTS

### 2.1 NOTATIONS

Let $\mathcal{F}$ be a functional space, e.g. $L_2(\mathbb{R}^p \to \mathbb{R}^d)$, and a loss function $\mathcal{L} : \mathcal{F} \to \mathbb{R}$ defined on it, of the form $\mathcal{L}(f) = \mathbb{E}_{(\boldsymbol{x},\boldsymbol{y})\sim\mathcal{D}}\left[\ell(f(\boldsymbol{x}),\boldsymbol{y})\right]$, where $\ell$ is the per-sample loss, assumed to be differentiable, and where $\mathcal{D}$ is the sample distribution, from which the dataset $\{(\boldsymbol{x}_1,\boldsymbol{y}_1),...,(\boldsymbol{x}_N,\boldsymbol{y}_N)\}$ is sampled, with $\boldsymbol{x}_i \in \mathbb{R}^p$ and $\boldsymbol{y}_i \in \mathbb{R}^d$.

For the sake of simplicity we consider a feedforward neural network $f_\theta : \mathbb{R}^p \to \mathbb{R}^d$ with $L$ hidden layers, each of which consisting of an affine layer with weights $\boldsymbol{W}_l$ followed by a differentiable activation function $\sigma_l$ which satisfies $\sigma_l(0) = 0$. The network parameters are then $\theta := (\boldsymbol{W}_l)_{l=1...L}$. The network iteratively computes:

$$\boldsymbol{b}_0(\boldsymbol{x}) = \begin{pmatrix} \boldsymbol{x} \\ 1 \end{pmatrix}$$

$$\forall l \in [1, L], \quad \begin{cases} \boldsymbol{a}_l(\boldsymbol{x}) &= \boldsymbol{W}_l\,\boldsymbol{b}_{l-1}(\boldsymbol{x}) \\ \boldsymbol{b}_l(\boldsymbol{x}) &= \begin{pmatrix} \sigma_l(\boldsymbol{a}_l(\boldsymbol{x})) \\ 1 \end{pmatrix} \end{cases}$$

$$f_\theta(\boldsymbol{x}) = \sigma_L(\boldsymbol{a}_L(\boldsymbol{x}))$$

Figure 1: Notations

To any vector-valued function noted $\boldsymbol{t}(\boldsymbol{x})$ and any batch of inputs $\boldsymbol{X} := [\boldsymbol{x}_1,...,\boldsymbol{x}_n]$, we associate the concatenated matrix $\boldsymbol{T}(\boldsymbol{X}) := (\boldsymbol{t}(\boldsymbol{x}_1) \quad ... \quad \boldsymbol{t}(\boldsymbol{x}_n)) \in \mathbb{R}^{|\boldsymbol{t}(.)|\times n}$. The matrices of pre-activation and post-activation activities at layer $l$ over a minibatch $\boldsymbol{X}$ are thus respectively: $\boldsymbol{A}_l(\boldsymbol{X}) = (\boldsymbol{a}_l(\boldsymbol{x}_1) \quad ... \quad \boldsymbol{a}_l(\boldsymbol{x}_n))$ and $\boldsymbol{B}_l(\boldsymbol{X}) = (\boldsymbol{b}_l(\boldsymbol{x}_1) \quad ... \quad \boldsymbol{b}_l(\boldsymbol{x}_n))$.

*NB: convolutions can also be considered, with appropriate representations (cf matrix $\boldsymbol{b}_l^c(\boldsymbol{x})$ in 10).*

### 2.2 APPROACH

**Functional gradient descent.** We take a functional perspective on the use of neural networks. Ideally in a machine learning task, one would search for a function $f : \mathbb{R}^p \to \mathbb{R}^d$ that minimizes the loss $\mathcal{L}$ by gradient descent: $\frac{\partial f}{\partial t} = -\nabla_f \mathcal{L}(f)$ for some metric on the functional space $\mathcal{F}$ (typically, $L_2(\mathbb{R}^p \to \mathbb{R}^d)$), where $\nabla_f$ denotes the functional gradient and $t$ denotes the evolution time of the gradient descent. The descent direction $\boldsymbol{v}_{\text{goal}} := -\nabla_f \mathcal{L}(f)$ is a function of the same type as $f$ and whose value at $\boldsymbol{x}$ is easily computable as $\boldsymbol{v}_{\text{goal}}(\boldsymbol{x}) = -(\nabla_f \mathcal{L}(f))(\boldsymbol{x}) = -\nabla_{\boldsymbol{u}} \ell(\boldsymbol{u}, \boldsymbol{y}(\boldsymbol{x}))\big|_{\boldsymbol{u}=f(\boldsymbol{x})}$ (see Appendix A.1 for more details). This direction $\boldsymbol{v}_{\text{goal}}$ is the best infinitesimal variation in $\mathcal{F}$ to add to $f$ to decrease the loss $\mathcal{L}$.

**Parametric gradient descent reminder.** However in practice, to represent functions and to compute gradients, the infinite-dimensional functional space $\mathcal{F}$ has to be replaced with a finite-dimensional parametric space of functions, which is usually done by choosing a particular neural network architecture $\mathcal{A}$ with weights $\theta \in \Theta_\mathcal{A}$. The associated parametric search space $\mathcal{F}_\mathcal{A}$ then consists of all possible functions $f_\theta$ that can be represented with such a network for any parameter value $\theta$. Under standard weak assumptions (see Appendix A.2), the gradient descent is of the form:

$$\frac{\partial \theta}{\partial t} = -\nabla_\theta \mathcal{L}(f_\theta) = -\mathbb{E}_{(\boldsymbol{x},\boldsymbol{y})\sim\mathcal{D}}\left[\nabla_\theta \ell(f_\theta(\boldsymbol{x}),\boldsymbol{y})\right].$$

Using the chain rule, these parameter updates yield a functional evolution :

$$\boldsymbol{v}_{\text{GD}} := \frac{\partial f_\theta}{\partial t} = \frac{\partial f_\theta}{\partial \theta}\frac{\partial \theta}{\partial t} = \frac{\partial f_\theta}{\partial \theta} \underset{(\boldsymbol{x},\boldsymbol{y})\sim\mathcal{D}}{\mathbb{E}}\left[\frac{\partial f_\theta}{\partial \theta}^T(\boldsymbol{x})\,\boldsymbol{v}_{\text{goal}}(\boldsymbol{x})\right]$$

which significantly differs from the original functional gradient descent. We will aim to augment the neural network architecture so that parametric gradient descents can get closer to the functional one.

**Optimal move direction.** We name $\mathcal{T}_{\mathcal{A}}^{f_\theta}$, or just $\mathcal{T}_{\mathcal{A}}$, the tangent space of $\mathcal{F}_{\mathcal{A}}$ at $f_\theta$, that is, the set of all possible infinitesimal variations around $f_\theta$ under small parameter variations:

$$\mathcal{T}_{\mathcal{A}}^{f_\theta} := \left\{ \frac{\partial f_\theta}{\partial \theta}\,\delta\theta \ \middle| \ \text{s.t. } \delta\theta \in \Theta_{\mathcal{A}} \right\}$$

This linear space is a first-order approximation of the neighborhood of $f_\theta$ within $\mathcal{F}_{\mathcal{A}}$. The direction $\boldsymbol{v}_{\text{GD}}$ obtained above by gradient descent is actually not the best one to consider within $\mathcal{T}_{\mathcal{A}}$. Indeed, the best move $\boldsymbol{v}^*$ would be the orthogonal projection of the desired direction $\boldsymbol{v}_{\text{goal}} := -\nabla_{f_\theta}\mathcal{L}(f_\theta)$ onto $\mathcal{T}_{\mathcal{A}}$. This projection is what a (generalization of the notion of) natural gradient would compute (Ollivier, 2017).

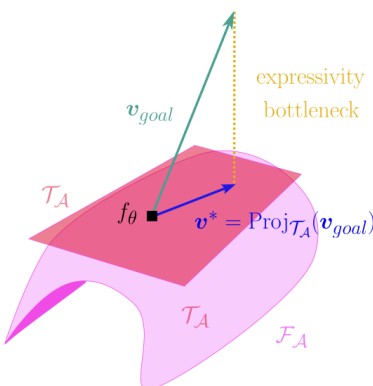

Figure 2: Expressivity bottleneck

Indeed, the parameter variation $\delta\theta^*$ associated to the functional variation $\boldsymbol{v}^* = \frac{\partial f_\theta}{\partial \theta}\,\delta\theta^*$ is the gradient $-\nabla_\theta^{\mathcal{T}_{\mathcal{A}}}\mathcal{L}(f_\theta)$ of $\mathcal{L}\circ f_\theta$ w.r.t. parameters $\theta$ when considering the $L_2$ metric on *functional* variations $\|\frac{\partial f_\theta}{\partial \theta}\,\delta\theta\|_{L_2(\mathcal{T}_{\mathcal{A}})}$, not to be confused with the usual gradient $\nabla_\theta\mathcal{L}(f_\theta)$, based on the $L_2$ metric on *parameter* variations $\|\delta\theta\|_{L_2(\mathbb{R}^{|\Theta_{\mathcal{A}}|})}$. This can be seen in a proximal formulation as:

$$\boldsymbol{v}^* = \underset{\boldsymbol{v}\in\mathcal{T}_{\mathcal{A}}}{\arg\min}\,\|\boldsymbol{v}-\boldsymbol{v}_{\text{goal}}\|^2 = \underset{\boldsymbol{v}\in\mathcal{T}_{\mathcal{A}}}{\arg\min}\left\{D_f\mathcal{L}(f)(\boldsymbol{v})+\frac{1}{2}\|\boldsymbol{v}\|^2\right\} \tag{1}$$

where $D$ is the directional derivative (see details in Appendix A.3), or equivalently as:

$$\delta\theta^* = \underset{\delta\theta\in\Theta_{\mathcal{A}}}{\arg\min}\left\|\frac{\partial f_\theta}{\partial \theta}\,\delta\theta-\boldsymbol{v}_{\text{goal}}\right\|^2 = \underset{\delta\theta\in\Theta_{\mathcal{A}}}{\arg\min}\left\{D_\theta\mathcal{L}(f_\theta)(\delta\theta)+\frac{1}{2}\left\|\frac{\partial f_\theta}{\partial \theta}\,\delta\theta\right\|^2\right\} =: -\nabla_\theta^{\mathcal{T}_{\mathcal{A}}}\mathcal{L}(f_\theta)\,.$$

**Lack of expressivity.** When $\boldsymbol{v}_{\text{goal}}$ does not belong to the reachable subspace $\mathcal{T}_{\mathcal{A}}$, there is a lack of expressivity, that is, the parametric space $\mathcal{A}$ is not rich enough to follow the ideal functional gradient descent. This happens frequently with small neural networks (see Appendix A.4 for an example). The expressivity bottleneck is then quantified as the distance $\|\boldsymbol{v}^* - \boldsymbol{v}_{\text{goal}}\|$ between the functional gradient $\boldsymbol{v}_{\text{goal}}$ and the optimal functional move $\boldsymbol{v}^*$ given the architecture $\mathcal{A}$ (in the sense of Eq. 1).

### 2.3 GENERALIZING TO ALL LAYERS

**Ideal updates.** The same reasoning can be applied to the pre-activations $\boldsymbol{a}_l$ at each layer $l$, seen as functions $\boldsymbol{a}_l : \boldsymbol{x} \in \mathbb{R}^p \mapsto \boldsymbol{a}_l(\boldsymbol{x}) \in \mathbb{R}^{d_l}$ defined over the input space of the neural network. The optimal parameter update for a given layer $l$ then follows the projection of the desired update $-\nabla_{\boldsymbol{a}_l}\mathcal{L}(f_\theta)$ of the pre-activation functions $\boldsymbol{a}_l$ onto the linear subspace $\mathcal{T}_{\mathcal{A}}^{\boldsymbol{a}_l}$ of pre-activation variations that are possible with the architecture, as we will detail now.

Given an sample $(\boldsymbol{x},\boldsymbol{y}) \in \mathcal{D}$, standard backpropagation already iteratively computes $\boldsymbol{v}_{\text{goal}}^l(\boldsymbol{x}) := -\left(\nabla_{\boldsymbol{a}_l}\mathcal{L}(f_\theta)\right)(\boldsymbol{x}) = -\nabla_{\boldsymbol{u}}\ell\left(\sigma_L(\boldsymbol{W}_L\,\sigma_{L-1}(\boldsymbol{W}_{L-1}...\sigma_l(\boldsymbol{u}))),\,\boldsymbol{y}\right)|_{\boldsymbol{u}=\boldsymbol{a}_l(\boldsymbol{x})}$, which is the derivative of the loss $\ell(f_\theta(\boldsymbol{x}),\boldsymbol{y})$ with respect to the pre-activations $\boldsymbol{u} = \boldsymbol{a}_l(\boldsymbol{x})$ of each layer. This is usually performed in order to compute the gradients w.r.t. model parameters $\boldsymbol{W}_l$, as $\nabla_{\boldsymbol{W}_l}\ell(f_\theta(\boldsymbol{x}),\boldsymbol{y}) = \frac{\partial \boldsymbol{a}_l(\boldsymbol{x})}{\partial W_l}\,\nabla_{\boldsymbol{a}_l}\ell(f_\theta(\boldsymbol{x}),\boldsymbol{y})$.

$\boldsymbol{v}_{\text{goal}}^l(\boldsymbol{x}) := -\left(\nabla_{\boldsymbol{a}_l}\mathcal{L}(f_\theta)\right)(\boldsymbol{x})$ indicates the direction in which one would like to change the layer pre-activations $\boldsymbol{a}_l(\boldsymbol{x})$ in order to decrease the loss at point $\boldsymbol{x}$. However, given a minibatch of points $(\boldsymbol{x}_i)$, most of the time no parameter move $\delta\theta$ is able to induce this progression for each $\boldsymbol{x}_i$ simultaneously, because the $\theta$-parameterized family of functions $\boldsymbol{a}_l$ is not expressive enough.

**Activity update resulting from a parameter change.** Given a subset of parameters $\tilde{\theta}$ (such as the ones specific to a layer: $\tilde{\theta} = \boldsymbol{W}_l$), and an incremental direction $\delta\tilde{\theta}$ to update these parameters (e.g. the one resulting from a gradient descent: $\delta\tilde{\theta} = -\eta \sum_{(\boldsymbol{x},\boldsymbol{y}) \in \text{minibatch}} \nabla_{\tilde{\theta}} \ell(f_\theta(\boldsymbol{x}), \boldsymbol{y})$ for some learning rate $\eta$), the impact of the parameter update $\delta\tilde{\theta}$ on the pre-activations $\boldsymbol{a}_l$ at layer $l$ at order 1 in $\delta\tilde{\theta}$ is $\boldsymbol{v}^l(\boldsymbol{x}_i, \delta\tilde{\theta}) := \dfrac{\partial \boldsymbol{a}_l(\boldsymbol{x})}{\partial \tilde{\theta}} \, \delta\tilde{\theta}$.

*Note:* given a learning rate $\eta$, in the sequel we will rather consider $\boldsymbol{v}^l_{\text{goal}}(\boldsymbol{x}) := -\eta \nabla_{\boldsymbol{a}_l} \mathcal{L}(f_\theta)(\boldsymbol{x})$.

## 3 EXPRESSIVITY BOTTLENECKS

We now quantify expressivity bottlenecks at any layer $l$ as the distance between the desired activity update $\boldsymbol{v}^l_{\text{goal}}(.)$ and the best realizable one $\boldsymbol{v}^l(.)$ (cf Figure 2):

**Definition 3.1** (Lack of expressivity). *For a neural network $f_\theta$ and a minibatch of points $\boldsymbol{X} = \{(\boldsymbol{x}_i, \boldsymbol{y}_i)\}_{i=1}^n$, we define the lack of expressivity at layer $l$ as how far the desired activity update $\boldsymbol{V}^l_{goal} = (\boldsymbol{v}^l_{goal}(\boldsymbol{x}_1), \boldsymbol{v}^l_{goal}(\boldsymbol{x}_2), ...)$ is from the closest possible activity update $V^l = (\boldsymbol{v}^l(\boldsymbol{x}_1), \boldsymbol{v}^l(\boldsymbol{x}_2), ...)$ realizable by a parameter change $\delta\theta$:*

$$\Psi^l := \min_{\boldsymbol{v}^l \in \mathcal{T}_\mathcal{A}^{\boldsymbol{a}_l}} \frac{1}{n} \sum_{i=1}^n \left\| \boldsymbol{v}^l(\boldsymbol{x}_i) - \boldsymbol{v}^l_{\text{goal}}(\boldsymbol{x}_i) \right\|^2 = \min_{\delta\theta} \frac{1}{n} \left\| \boldsymbol{V}^l(\boldsymbol{X}, \delta\theta) - \boldsymbol{V}^l_{\text{goal}}(\boldsymbol{X}) \right\|^2_{\text{Tr}} \quad (2)$$

where $||.||$ stands for the $L_2$ norm, $||.||_{\text{Tr}}$ for the Frobenius norm, and $\boldsymbol{V}^l(\boldsymbol{X}, \delta\theta)$ is the activity update resulting from parameter change $\delta\theta$ as defined in previous section. In the two following parts we fix the minibatch $\boldsymbol{X}$ and simplify notations accordingly by removing the dependency on $\boldsymbol{X}$.

### 3.1 BEST MOVE WITHOUT MODIFYING THE ARCHITECTURE OF THE NETWORK

Let $\delta\boldsymbol{W}_l^*$ be the solution of 2 when the parameter variation $\delta\theta$ is restricted to involve only layer $l$ parameters, i.e. $\boldsymbol{W}_l$. This move is sub-optimal in that it does not result from an update of all architecture parameters but only of the current layer ones:

$$\delta\boldsymbol{W}_l^* = \arg\min_{\delta\mathbf{W}_l} \frac{1}{n} \left\| \boldsymbol{V}^l(\delta\mathbf{W}_l) - \boldsymbol{V}^l_{\text{goal}} \right\|^2_{\text{Tr}} \quad (3)$$

**Proposition 3.1.** *The solution of Problem (3) is:*

$$\delta\boldsymbol{W}_l^* = \frac{1}{n} \boldsymbol{V}^l_{goal} \boldsymbol{B}_{l-1}^T \left( \frac{1}{n} \boldsymbol{B}_{l-1} \boldsymbol{B}_{l-1}^T \right)^+$$

*where $P^+$ denotes the generalized inverse of matrix $P$.*

This update $\delta\boldsymbol{W}_l^*$ is not equivalent to the usual gradient descent update, whose form is $\delta\boldsymbol{W}_l^{\text{GD}} \propto \boldsymbol{V}^l_{\text{goal}} \boldsymbol{B}_{l-1}^T$. In fact the associated activity variation, $\delta\boldsymbol{W}_l^* \boldsymbol{B}_{l-1}$, is the projection of $\boldsymbol{V}^l_{\text{goal}}$ on the post-activation matrix of layer $l-1$, that is to say onto the span of all possible post-activation directions, through the projector $\frac{1}{n} \boldsymbol{B}_{l-1}^T (\frac{1}{n} \boldsymbol{B}_{l-1} \boldsymbol{B}_{l-1}^T)^+ \boldsymbol{B}_{l-1}$. To increase expressivity if needed, we will aim at increasing this span with the most useful directions to close the gap between this best update and the desired one. Note that the update $\delta\boldsymbol{W}_l^*$ consists of a standard gradient ($\boldsymbol{V}^l_{\text{goal}} \boldsymbol{B}_{l-1}^T$) and of a (kind of) natural gradient only for the last part (projector), as we consider metrics in the pre-activation space.

### 3.2 REDUCING EXPRESSIVITY BOTTLENECK BY MODIFYING THE ARCHITECTURE

To get as close as possible to $\boldsymbol{V}^l_{\text{goal}}$ and to increase the expressive power of the current neural network, we modify each layer of its structure. At layer $l-1$, we add $K$ neurons $n_1, ..., n_K$ with input weights $\boldsymbol{\alpha}_1, ..., \boldsymbol{\alpha}_k$ and output weights $\boldsymbol{\omega}_1, ..., \boldsymbol{\omega}_K$ (cf Figure 3). We have the following expansions by concatenation : $\boldsymbol{W}_{l-1}^T \leftarrow \begin{pmatrix} \boldsymbol{W}_{l-1}^T & \boldsymbol{\alpha}_1 & ... & \boldsymbol{\alpha}_K \end{pmatrix}$ and $\boldsymbol{W}_l \leftarrow \begin{pmatrix} \boldsymbol{W}_l & \boldsymbol{\omega}_1 & ... & \boldsymbol{\omega}_K \end{pmatrix}$.

We note this architecture modification $\theta \leftarrow \theta \oplus \theta_{\leftrightarrow}^K$ where $\oplus$ is the concatenation sign and $\theta_{\leftrightarrow}^K := (\boldsymbol{\alpha}_k, \boldsymbol{\omega}_k)_{k=1}^K$ are the $K$ added neurons.

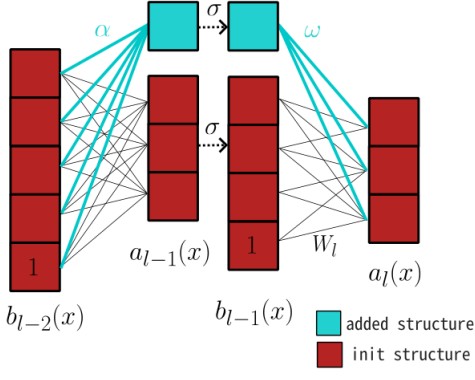
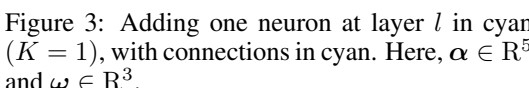

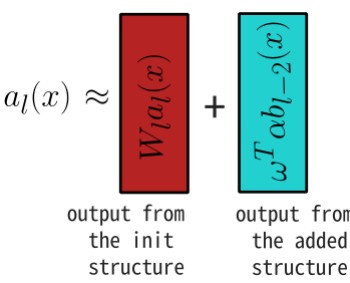

Figure 3: Adding one neuron at layer $l$ in cyan ($K = 1$), with connections in cyan. Here, $\boldsymbol{\alpha} \in \mathrm{R}^5$ and $\boldsymbol{\omega} \in \mathrm{R}^3$.

Figure 4: Sum of functional moves

The added neurons could be chosen randomly, as in usual neural network initialization, but this would not yield any guarantee regarding the impact on the system loss. Another possibility would be to set either input weights $(\boldsymbol{\alpha}_k)_{k=1}^K$ or output weights $(\boldsymbol{\omega}_k)_{k=1}^K$ to 0, so that the function $f_\theta(.)$ would not be modified, while its gradient w.r.t. $\theta$ would be enriched from the new parameters. Another option is to solve a optimization problem as in the previous section with the modified structure $\theta \leftarrow \theta \oplus \theta_\leftrightarrow^K$ and jointly search for both the optimal new parameters $\theta_\leftrightarrow^K$ and the optimal variation $\delta \mathbf{W}_l$ of the old ones:

$$\underset{\theta_\leftrightarrow^K, \, \delta \mathbf{W}_l}{\arg\min} \left\| \boldsymbol{V}^l(\delta \mathbf{W}_l \oplus \theta_\leftrightarrow^K) - \boldsymbol{V}_{\text{goal}}^l \right\|_{\text{Tr}}^2 \tag{4}$$

As shown in figure 4, the displacement $\boldsymbol{V}^l$ at layer $l$ is actually a sum of the moves induced by the neurons already present ($\delta \mathbf{W}_l$) and by the added neurons ($\theta_\leftrightarrow^K$), our problem rewrites as :

$$\underset{\theta_\leftrightarrow^K, \, \delta \mathbf{W}_l}{\arg\min} \left\| \boldsymbol{V}^l(\theta_\leftrightarrow^K) + \boldsymbol{V}^l(\delta \mathbf{W}_l) - {\boldsymbol{V}_{\text{goal}}}^l \right\|_{\text{Tr}}^2 \tag{5}$$

with $\boldsymbol{v}^l(\boldsymbol{x}, \theta_\leftrightarrow^K) := \sum_{j=1}^K \boldsymbol{\omega}_k \, (b_{l-2}(\boldsymbol{x})^T \boldsymbol{\alpha}_k)$ (See A.5). We choose $\delta \mathbf{W}_l$ as the best move of already-existing parameters as defined in Proposition 3.1 and we note $\boldsymbol{V}_{\text{goal}_{proj}}^l := \boldsymbol{V}_{\text{goal}}^l - \boldsymbol{V}^l(\delta \mathbf{W}_l^*)$. We are looking for the solution $(K^*, \theta_\leftrightarrow^{K*})$ of the optimization problem :

$$\underset{K, \, \theta_\leftrightarrow^K}{\arg\min} \left\| \boldsymbol{V}^l(\theta_\leftrightarrow^K) - \boldsymbol{V}_{\text{goal}_{proj}}^l \right\|_{\text{Tr}}^2 . \tag{6}$$

This quadratic optimization problem can be solved thanks to the low-rank matrix approximation theorem (Eckart & Young, 1936), using matrices $\boldsymbol{N} := \frac{1}{n} \boldsymbol{B}_{l-2} \left( \boldsymbol{V}_{\text{goal}_{proj}}^l \right)^T$ and $\boldsymbol{S} := \frac{1}{n} \boldsymbol{B}_{l-2} \boldsymbol{B}_{l-2}^T$. As $\boldsymbol{S}$ is semi-positive definite, let its truncated SVD be $\boldsymbol{S} = \boldsymbol{U} \Sigma \boldsymbol{U}^T$, and define $S^{-\frac{1}{2}} := \boldsymbol{U} \sqrt{\Sigma}^{-1} \boldsymbol{U}^T$, with the convention that the inverse of 0 eigenvalues is 0. Finally, consider the truncated SVD of matrix $S^{-\frac{1}{2}} \boldsymbol{N} = \sum_{k=1}^R \lambda_k \boldsymbol{u}_k \boldsymbol{v}_k^T$, where $R$ is the rank of the matrix $S^{-\frac{1}{2}} \boldsymbol{N}$. Then:

**Proposition 3.2.** *The solution of Problem (6) is:*
- *optimal number of neurons: $K^* = R$*
- *their optimal weights: $(\boldsymbol{\alpha}_k^*, \boldsymbol{\omega}_k^*) = \left( \operatorname{sign}(\lambda_k) \sqrt{|\lambda_k|} S^{-\frac{1}{2}} \boldsymbol{u}_k, \, \sqrt{|\lambda_k|} \boldsymbol{v}_k \right)$ for $k = 1, ..., R$.*

*Moreover for any number of neurons $K \leqslant R$, and associated weights $\theta_\leftrightarrow^{K,*}$, the expressivity gain and the first order in $\eta$ of the loss improvement due to the addition of these $K$ neurons are equal and can be quantified very simply as a function of the singular values $\lambda_k$:*

$$\Psi_{\theta \oplus \theta_\leftrightarrow^{K,*}}^l = \Psi_\theta^l - \sum_{k=1}^K \lambda_k^2 \qquad and \qquad \mathcal{L}(f_{\theta \oplus \theta_\leftrightarrow^{K,*}}) = \mathcal{L}(f_\theta) + \frac{\sigma_l'(0)}{\eta} \sum_{k=1}^K \lambda_k^2 + o(||\theta_\leftrightarrow^{K,*}||^2)$$

**Proposition 3.3.** *If $S$ is positive definite, then solving (5) is equivalent to taking $\omega_k = N\alpha_k$ and finding the $K$ first eigenvectors $\alpha_k$ associated to the $K$ largest eigenvalues $\lambda$ of the generalized eigenvalue problem :*

$$NN^T\alpha_k = \lambda S\alpha_k$$

**Corollary 1.** *For all integers $m, m'$ such that $m + m' \leqslant R$, at order one in $\eta$, adding $m + m'$ neurons simultaneously according to the previous method is equivalent to adding $m$ neurons then $m'$ neurons by applying successively the previous method twice.*

*Note:* Problems (5) and (6) are generally not equivalent, though similar (cf C.4).
*Note 2:* Minimizing the distance (4), ie the distance between $V^l_{\text{goal}}$ and $V^l$, is equivalent to minimizing the loss $\mathcal{L}$ at order one in $V^l$:

$$\mathcal{L}(f_{\theta \oplus \theta^K_{\leftrightarrow}}) \approx \mathcal{L}(f_\theta) - \frac{\sigma'_{l-1}(0)}{\eta\,n}\left\langle V^l_{\text{goal}},\, V^l \right\rangle_{\text{Tr}} \tag{7}$$

The family $\{V^{l+1}((\alpha_k, \omega_k))\}^K_{k=1}$ of pre-activity variations induced by adding the neurons $\theta^{K,*}_{\leftrightarrow}$ is orthogonal for the trace scalar product. We could say that the added neurons are orthogonal to each other (and to the already-present ones) in that sense. Interestingly, the GradMax method (Evci et al., 2022) also aims at minimizing the loss 7, but without avoiding redundancy (see Appendix B.1 for more details).

**Addition of new neurons.** In practice before adding new neurons $(\alpha, \omega)$, we multiply them by an amplitude factor $\gamma$ found by a simple line search (see Appendix D.4), i.e. we add $(\sqrt{\gamma}\alpha, \sqrt{\gamma}\omega)$. The addition of each neuron $k$ has an impact on the loss of the order of $\gamma\lambda^2_k$ provided $\gamma$ is small. This performance gain could be used in a selection criterion realizing a trade-off with computational complexity. A selection based on statistical significance of singular values can also be performed (Appendix D.3). The full algorithm and its complexity are detailed in Appendices D.5 and D.6.

## 4 ABOUT GREEDY GROWTH SUFFICIENCY

One might wonder whether a greedy approach on layer growth might get stuck in a non-optimal state. We derive the following series of propositions in this regard. Since in this work we add neurons layer per layer independently, we study here the case of a single hidden layer network, to spot potential layer growth issues. For the sake of simplicity, we consider the task of least square regression towards an explicit continuous target $f^*$, defined on a compact set. That is, we aim at minimizing the loss:

$$\inf_f \sum_{x \in \mathcal{D}} \|f(x) - f^*(x)\|^2$$

where $f(x)$ is the output of the neural network and $\mathcal{D}$ is the training set.

**Proposition 4.1** (Greedy completion of an existing network). *If $f$ is not $f^*$ yet, then there exists a set of neurons to add to the hidden layer such that the new function $f'$ will have a lower loss than $f$.*

One can even choose the added neurons such that the loss is arbitrarily well minimized. Furthermore:

**Proposition 4.2** (Greedy completion by one single neuron). *If $f$ is not $f^*$ yet, there exists a neuron to add to the hidden layer such that the new function $f'$ will have a lower loss than $f$.*

As a consequence, there exists no situation where one would need to add many neurons simultaneously to decrease the loss: it is always feasible with a single neuron. One can express a lower bound on how much the loss has improved (for the best such neuron), but it is not a very good bound without further assumptions on $f$. Furthermore, finding the optimal neuron to add is actually NP-hard (Bach, 2017), so we will not necessarily search for the optimal one.

**Proposition 4.3** (Greedy completion by one infinitesimal neuron). *The neuron in the previous proposition can be chosen to have arbitrarily small input weights.*

This detail is important in that our approach is based on the tangent space of the function $f$ and thus manipulates infinitesimal quantities. Our optimization problem indeed relies on the linearization of the activation function by requiring the added neuron to have infinitely small input weights, to make the problem easier to solve. This proposition confirms that such neuron exists indeed.

**Correlations and higher orders.** Note that, as a matter of fact, our approach exploits linear correlations between inputs of a layer and desired output variations. It might happen that the loss is not minimized yet but there is no such correlation to exploit anymore. In that case the optimization problem (6) will not find neurons to add. Yet following Prop. 4.3 there does exist a neuron with arbitrarily small input weights that can reduce the loss. This paradox can be explained by pushing further the Taylor expansion of that neuron output in terms of weight amplitude (single factor $\varepsilon$ on all of its input weights), for instance $\sigma(\varepsilon\boldsymbol{\alpha}\cdot\boldsymbol{x}) \simeq \sigma(0) + \sigma'(0)\varepsilon\boldsymbol{\alpha}\cdot\boldsymbol{x} + \frac{1}{2}\sigma''(0)\varepsilon^2(\boldsymbol{\alpha}\cdot\boldsymbol{x})^2 + O(\varepsilon^3)$. Though the linear term $\boldsymbol{\alpha}\cdot\boldsymbol{x}$ might be uncorrelated over the dataset with desired output variation, i.e. $\mathbb{E}_{\boldsymbol{x}\sim\mathcal{D}}[\boldsymbol{\alpha}\cdot\boldsymbol{x}] = 0$, the quadratic term $(\boldsymbol{\alpha}\cdot\boldsymbol{x})^2$, or higher-order ones otherwise, might be correlated. Finding neurons with such higher-order correlations can be done by increasing accordingly the power of $(\boldsymbol{\alpha}\cdot\boldsymbol{x})$ in the optimization problem (5). Note that one could consider other function bases than the polynomials from Taylor expansion. In all cases, one does not need to solve such problems exactly but just to find an approximate solution, i.e. a neuron improving the loss.

**Adding random neurons.** Another possibility to suggest additional neurons, when expressivity bottlenecks are detected but no correlation (up to order $p$) can be exploited anymore, is to add random neurons. The first $p$ order Taylor expansions will show 0 correlation with desired output variation, hence no loss improvement nor worsening, but the correlation of the $p+1$-th order will be non-0, with probability 1, in the spirit of random projections. Furthermore, in the spirit of common neural network training practice, one could consider brute force combinatorics by adding many random neurons and hoping some will be close enough to the desired direction (Frankle & Carbin, 2018). The difference with usual training is that we would perform such computationally-costly searches only when and where relevant, exploiting all simple information first (linear correlations in each layer).

## 5 RESULTS

We evaluate our method, called **TINY**, on the CIFAR-10 dataset (Krizhevsky et al., 2009). We start with an architecture of two blocks consisting of 2 convolutions and 1 MaxPooling each, followed by two fully-connected layers, using the the selu activation function. The training criterion is the cross-entropy loss. We increase the size of the network on the fly (during training), with a batch size of $500$ for training, see D.2. More experiments can be found in Appendix E.

In Figure 5, the TINY method is compared to GradMax (**GM**) (Evci et al., 2022) We imitate this method by skipping the projection step of the desired update on the current architecture. The *Left* and *Right* plot respectively correspond to the plots of the accuracy $acc_0$ on validation set after adding all the neurons, and the accuracy $acc_\infty$ at the end of training. On the $x$-axis, we measure the complexity of each model as the number of basic operations performed during test.

At a fixed complexity, before and after the final training, our method out-performs GradMax because of the redundancy of that procedure. While GradMax neurons align with the gradient when expressivity bottlenecks can be mostly solved by the current architecture, we add new directions and decrease the expressivity bottleneck as defined in (2) by projecting the desired update. In the GradMax paper, to avoid redundancy, neurons can be added only once the gradient of the current structure vanishes to 0. Our method allows to skip this waiting by projecting the desired update on the current structure using (3).

In Figure 6 we compare our results with a baseline (**FS**) where the final TINY architecture is retrained from start with a new set of weights, initialized with Xavier Normal distribution and trained until convergence. The Figure shows the evolution of train/test accuracy for the TINY and from scratch (**FS**) methods. No new neurons are added after 4000 steps in TINY. It is possible to achieve 100% accuracy on training dataset, thus overfitting the data, meaning that TINY could effectively bypass any expressivity bottleneck, and this with fewer parameters than (Zhang et al., 2017)'s theorem. While usual methods have to train until convergence different architectures to find the optimal one, our method achieves this result in a single run.

## 6 LIMITATIONS

At a given layer, our method reduces expressivity bottlenecks, unless the required information has been lost in previous layers. This situation might arise e.g. with very thin, deep neural networks

with ReLU activations. In such a case, the lack of information flow might prevent any update. This limitation however disappears once new layers or skip connections can be added to the architecture. In Appendix E.3 we are thus able to train a ResNet-18 model on CIFAR-100.

## 7 CONCLUSION

We have properly defined lacks of expressivity, and their minimization has allowed us to optimize the architecture on the fly, to better follow the functional gradient, enabling architecture growth, and escaping optimization issues usually associated to layer thinness. Work in progress includes the extension to layer addition in order to fully automatically develop architectures, a statistical test to assess the significance of estimated neurons to add, and a strategy for neuron addition based on a compromise between performance gain and added computational complexity.

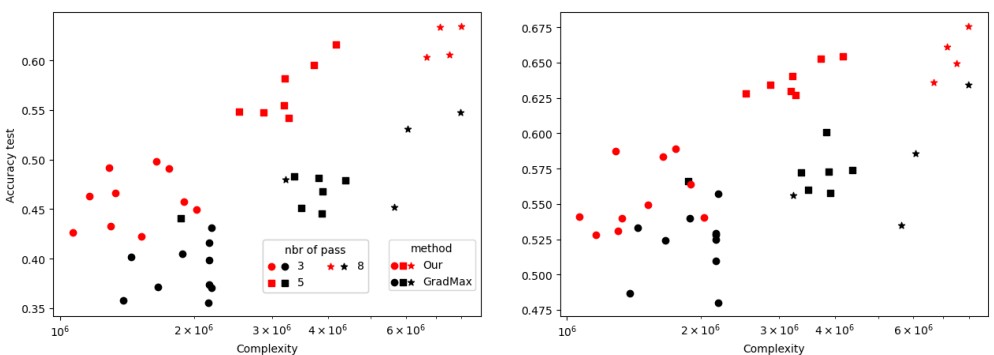

Figure 5: Experiments on CIFAR Dataset: accuracies (*Left*) $acc_0$, (*Right*) $acc_\infty$ on test set against the complexity for **GM** and **Our** method. The starting architecture $N_{init}$ has the structure of 3 neurons per convolutional layers and 5 neurons per linear layer which is of complexity $2 \times 10^5$.

---

**Algorithm 1:** Algorithm to plot Figure 5.

**for** *each method [Ours, GradMax]* **do**
    Start from a given small neural network $NN$;
    **for** *j in $nb_{pass}$* **do**
        **for** *each layer $l$* **do**
            Compute the first three neurons to add at layer $l$ ;
                *// with our method the above also yields $\delta W_l^*$ as a by-product*;
            Select among them the ones with significant singular value (Appendix D.3);
            Compute the amplitude factor $\gamma$ for the new neurons (Appendix D.4) ;
            Add the neurons if $\gamma > 0$;
            Update the architecture with the best update $\delta W_l^*$ at $l+1$ (only for our method)
        **end**
    **end**
    Get accuracy $acc_0$ of $NN$;
    Train $NN$ for 15 epochs ;
    Get final accuracy $acc_\infty$ of $NN$;
**end**

---

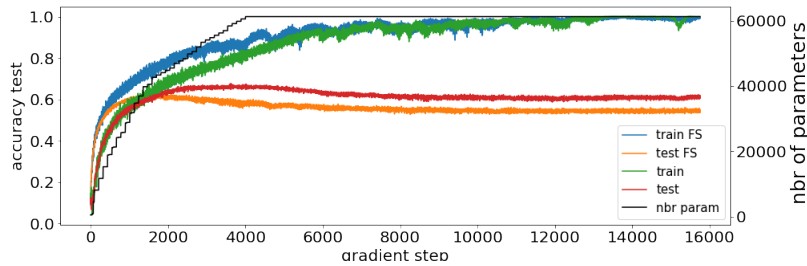

Figure 6: Evolution of the train/test accuracy wrt gradient steps for TINY and FS.

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

We provide additional details, following the same order as the sections in the paper.

# A  THEORETICAL DETAILS FOR PART 2

## A.1  FUNCTIONAL GRADIENT

The functional loss $\mathcal{L}$ is a functional that takes as input a function $f \in \mathcal{F}$ and outputs a real score:

$$\mathcal{L} : \ f \in \mathcal{F} \ \mapsto \ \mathcal{L}(f) = \mathop{\mathbb{E}}_{(\boldsymbol{x}, \boldsymbol{y}) \sim \mathcal{D}} \Big[ \ell(f(\boldsymbol{x}), \boldsymbol{y}) \Big] \ \in \mathbb{R} \ .$$

The function space $\mathcal{F}$ can typically be chosen to be $L_2(\mathbb{R}^p \to \mathbb{R}^d)$, which is a Hilbert space. The directional derivative (or Gateaux derivative, or Fréchet derivative) of functional $\mathcal{L}$ at function $f$ in direction $v$ is defined as:

$$D\mathcal{L}(f)(v) = \lim_{\varepsilon \to 0} \frac{\mathcal{L}(f + \varepsilon v) - \mathcal{L}(f)}{\varepsilon}$$

if it exists. Here $v$ denotes any function in the Hilbert space $\mathcal{F}$ and stands for the direction in which we would like to update $f$, following an infinitesimal step (of size $\varepsilon$), resulting in a function $f + \varepsilon v$.

If this directional derivative exists in all possible directions $v \in \mathcal{F}$ and moreover is continuous in $v$, then the Riesz representation theorem implies that there exists a unique direction $v^* \in \mathcal{F}$ such that:

$$\forall v \in \mathcal{F}, \ \ D\mathcal{L}(f)(v) = \langle v^*, v \rangle \ .$$

This direction $v^*$ is named the *gradient* of the functional $\mathcal{L}$ at function $f$ and is denoted by $\nabla_f \mathcal{L}(f)$.

Note that while the inner product $\langle, \rangle$ considered is usually the $L_2$ one, it is possible to consider other ones, such as Sobolev ones (e.g., $H^1$). The gradient $\nabla_f \mathcal{L}(F)$ depends on the chosen inner product and should consequently rather be denoted by $\nabla_f^{L_2} \mathcal{L}(f)$ for instance.

Note that continuous functions from $\mathbb{R}^p$ to $\mathbb{R}^d$, as well as $C^\infty$ functions, are dense in $L_2(\mathbb{R}^p \to \mathbb{R}^d)$.

Let us now study properties specific to our loss design: $\mathcal{L}(f) = \mathbb{E}_{(\boldsymbol{x}, \boldsymbol{y}) \sim \mathcal{D}} \Big[ \ell(f(\boldsymbol{x}), \boldsymbol{y}) \Big]$. Assuming sufficient $\ell$-loss differentiability and integrability, we get, for any function update direction $v \in \mathcal{F}$ and infinitesimal step size $\varepsilon \in \mathbb{R}$:

$$\mathcal{L}(f + \varepsilon v) - \mathcal{L}(f) = \mathop{\mathbb{E}}_{(\boldsymbol{x}, \boldsymbol{y}) \sim \mathcal{D}} \Big[ \ell(f(\boldsymbol{x}) + \varepsilon v(\boldsymbol{x}), \boldsymbol{y}) - \ell(f(\boldsymbol{x}), \boldsymbol{y}) \Big]$$

$$= \mathop{\mathbb{E}}_{(\boldsymbol{x}, \boldsymbol{y}) \sim \mathcal{D}} \Big[ \nabla_{\boldsymbol{u}} \ell(\boldsymbol{u}, \boldsymbol{y}) \big|_{\boldsymbol{u} = f(\boldsymbol{x})} \cdot \varepsilon v(\boldsymbol{x}) + O(\varepsilon^2 \|v(\boldsymbol{x})\|^2) \Big]$$

using the usual gradient of function $\ell$ at point $(\boldsymbol{u} = f(\boldsymbol{x}), \boldsymbol{y})$ w.r.t. its first argument $\boldsymbol{u}$, with the standard Euclidean dot product $\cdot$ in $\mathbb{R}^p$. Then the directional derivative is:

$$D\mathcal{L}(f)(v) = \mathop{\mathbb{E}}_{(\boldsymbol{x}, \boldsymbol{y}) \sim \mathcal{D}} \Big[ \nabla_{\boldsymbol{u}} \ell(\boldsymbol{u}, \boldsymbol{y}) \big|_{\boldsymbol{u} = f(\boldsymbol{x})} \cdot v(\boldsymbol{x}) \Big] = \mathop{\mathbb{E}}_{\boldsymbol{x} \sim \mathcal{D}} \Big[ \mathop{\mathbb{E}}_{\boldsymbol{y} \sim \mathcal{D}|\boldsymbol{x}} \Big[ \nabla_{\boldsymbol{u}} \ell(\boldsymbol{u}, \boldsymbol{y}) \big|_{\boldsymbol{u} = f(\boldsymbol{x})} \Big] \cdot v(\boldsymbol{x}) \Big]$$

and thus the functional gradient for the inner product $\langle v, v' \rangle_{\mathbb{E}} := \mathbb{E}_{\boldsymbol{x} \sim \mathcal{D}} \Big[ v(\boldsymbol{x}) \cdot v'(\boldsymbol{x}) \Big]$ is the function:

$$\nabla_f^{\mathbb{E}} \mathcal{L}(f) : \boldsymbol{x} \mapsto \mathop{\mathbb{E}}_{\boldsymbol{y} \sim \mathcal{D}|\boldsymbol{x}} \Big[ \nabla_{\boldsymbol{u}} \ell(\boldsymbol{u}, \boldsymbol{y}) \big|_{\boldsymbol{u} = f(\boldsymbol{x})} \Big]$$

which simplifies into:

$$\nabla_f^{\mathbb{E}} \mathcal{L}(f) : \boldsymbol{x} \mapsto \nabla_{\boldsymbol{u}} \ell(\boldsymbol{u}, \boldsymbol{y}(\boldsymbol{x})) \big|_{\boldsymbol{u} = f(\boldsymbol{x})}$$

if there is no ambiguity in the dataset, i.e. if for each $\boldsymbol{x}$ there is a unique $\boldsymbol{y}(\boldsymbol{x})$.

Note that by considering the $L_2(\mathbb{R}^p \to \mathbb{R}^d)$ inner product $\int v \cdot v'$ instead, one would respectively get:

$$\nabla_f^{L_2} \mathcal{L}(f) : \boldsymbol{x} \mapsto p_{\mathcal{D}}(\boldsymbol{x}) \mathop{\mathbb{E}}_{\boldsymbol{y} \sim \mathcal{D}|\boldsymbol{x}} \Big[ \nabla_{\boldsymbol{u}} \ell(\boldsymbol{u}, \boldsymbol{y}) \big|_{\boldsymbol{u} = f(\boldsymbol{x})} \Big]$$

and

$$\nabla_f^{L_2} \mathcal{L}(f) : \boldsymbol{x} \mapsto p_{\mathcal{D}}(\boldsymbol{x}) \nabla_{\boldsymbol{u}} \ell(\boldsymbol{u}, \boldsymbol{y}(\boldsymbol{x})) \big|_{\boldsymbol{u} = f(\boldsymbol{x})}$$

instead, where $p_{\mathcal{D}}(\boldsymbol{x})$ is the density of the dataset distribution at point $\boldsymbol{x}$. In practice one estimates such gradients using a minibatch of samples $(\boldsymbol{x}, \boldsymbol{y})$, obtained by picking uniformly at random within a finite dataset, and thus the formulas for the two inner products coincide (up to a constant factor).

## A.2 DIFFERENTIATION UNDER THE INTEGRAL SIGN

Let $X$ be an open subset of $\mathbb{R}$, and $\Omega$ be a measure space. Suppose $f : X \times \Omega \to \mathbb{R}$ satisfies the following conditions:

- $f(x, \omega)$ is a Lebesgue-integrable function of $\omega$ for each $x \in X$.
- For almost all $\omega \in \Omega$, the partial derivative $f_x$ of $f$ according to $x$ exists for all $x \in X$.
- There is an integrable function $\theta : \Omega \to \mathbb{R}$ such that $|f_x(x, \omega)| \leq \theta(\omega)$ for all $x \in X$ and almost every $\omega \in \Omega$.

Then, for all $x \in X$,

$$\frac{d}{dx} \int_\Omega f(x, \omega) \, d\omega = \int_\Omega f_x(x, \omega) \, d\omega$$

See proof and details :Flanders (1973).

## A.3 GRADIENTS AND PROXIMAL POINT OF VIEW

Gradients with respect to standard variables such as vectors are defined the same way as functional gradients above: given a sufficiently smooth loss $\widetilde{\mathcal{L}} : \theta \in \Theta_\mathcal{A} \mapsto \widetilde{\mathcal{L}}(\theta) = \mathcal{L}(f_\theta) \in \mathbb{R}$, and an inner product $\cdot$ in the space $\Theta_\mathcal{A}$ of parameters $\theta$, the gradient $\nabla_\theta \widetilde{\mathcal{L}}(\theta)$ is the unique vector $\boldsymbol{\tau} \in \Theta_\mathcal{A}$ such that:

$$\forall \delta\theta \in \Theta_\mathcal{A}, \quad \boldsymbol{\tau} \cdot \delta\theta = D_\theta \widetilde{\mathcal{L}}(\theta)(\delta\theta)$$

where $D_\theta \widetilde{\mathcal{L}}(\theta)(\delta\theta)$ is the directional derivative of $\widetilde{\mathcal{L}}$ at point $\theta$ in the direction $\delta\theta$, defined as in the previous section. This gradient depends on the inner product chosen, which can be highlighted by the following property. The opposite $-\nabla_\theta \widetilde{\mathcal{L}}(\theta)$ of the gradient is the unique solution of the problem:

$$\arg \min_{\delta\theta \in \Theta_\mathcal{A}} \left\{ D_\theta \widetilde{\mathcal{L}}(\theta)(\delta\theta) + \frac{1}{2} \|\delta\theta\|_P^2 \right\}$$

where $\| \ \|_P$ is the norm associated to the chosen inner product. Changing the inner product obviously changes the way candidate directions $\delta\theta$ are penalized, leading to different gradients. This proximal formulation can be obtained as follows. For any $\delta\theta$, its distance to the gradient descent direction is:

$$\left\| \delta\theta - \left( -\nabla_\theta \widetilde{\mathcal{L}}(\theta) \right) \right\|^2 = \|\delta\theta\|^2 + 2\,\delta\theta \cdot \nabla_\theta \widetilde{\mathcal{L}}(\theta) + \left\| \nabla_\theta \widetilde{\mathcal{L}}(\theta) \right\|^2$$

$$= 2 \left( \frac{1}{2} \|\delta\theta\|^2 + D_\theta \widetilde{\mathcal{L}}(\theta)(\delta\theta) \right) + K$$

where $K$ does not depend on $\delta\theta$. For the above to hold, the inner product used has to be the one from which the norm is derived. By minimizing this expression with respect to $\delta\theta$, one obtains the desired property.

In our case of study, for the norm over the space $\Theta_\mathcal{A}$ of parameter variations, we consider a norm of in the space of associated functional variations, i.e.:

$$\|\delta\theta\|_P := \left\| \frac{\partial f_\theta}{\partial \theta} \delta\theta \right\|$$

which makes more sense from a physical point of view, as it is more intrinsic to the task to solve and depends as little as possible on the parameterization (i.e. on the architecture chosen). This results in a functional move that is the projection of the functional one to the set of possible moves given the architecture. On the opposite, the standard gradient (using Euclidean parameter norm $\|\delta\theta\|$ in parameter space) yields a functional move obtained not only by projecting the functional gradient but also by multiplying it by a matrix $\frac{\partial f_\theta}{\partial \theta} \frac{\partial f_\theta}{\partial \theta}^T$ which can be seen as a strong architecture bias over optimization directions.

We consider here that the loss $\mathcal{L}$ to be minimized is the real loss that the user wants to optimize, possibly including regularizers to avoid overfitting, and since the architecture is evolving during training, possibly to architectures far from usual manual design and never tested before, one cannot

assume architecture bias to be desirable. We aim at getting rid of it in order to follow the functional gradient descent as closely as possible.

Searching for

$$\boldsymbol{v}^* \;=\; \underset{\boldsymbol{v}\in\mathcal{T}_\mathcal{A}}{\arg\min}\,\|\boldsymbol{v}-\boldsymbol{v}_{\text{goal}}\|^2 \;=\; \underset{\boldsymbol{v}\in\mathcal{T}_\mathcal{A}}{\arg\min}\left\{D_f\mathcal{L}(f)(\boldsymbol{v}) + \frac{1}{2}\|\boldsymbol{v}\|^2\right\}$$

or equivalently for:

$$\delta\theta^* \;=\; \underset{\delta\theta\in\Theta_\mathcal{A}}{\arg\min}\,\left\|\frac{\partial f_\theta}{\partial\theta}\,\delta\theta - \boldsymbol{v}_{\text{goal}}\right\|^2 \;=\; \underset{\delta\theta\in\Theta_\mathcal{A}}{\arg\min}\left\{D_\theta\mathcal{L}(f_\theta)(\delta\theta) + \frac{1}{2}\left\|\frac{\partial f_\theta}{\partial\theta}\,\delta\theta\right\|^2\right\} =: -\nabla_\theta^{\mathcal{T}_\mathcal{A}}\mathcal{L}(f_\theta)$$

then appears as a natural goal.

## A.4 Example of expressivity bottleneck

**Example.** Suppose one tries to estimate the function $y = f_{\text{true}}(x) = 2\sin(x) + x$ with a linear model $f_{\text{predict}}(x) = ax + b$. Consider $(a, b) = (1, 0)$ and the square loss $\mathcal{L}$. For the dataset of inputs $(x_0, x_1, x_2, x_3) = (0, \frac{\pi}{2}, \pi, \frac{3\pi}{2})$, there exists no parameter update $(\delta a, \delta b)$ that would improve prediction at $x_0, x_1, x_2$ and $x_3$ simultaneously, as the space of linear functions $\{f : x \to ax + b \mid a, b \in \mathbb{R}\}$ is not expressive enough. To improve the prediction at $x_0, x_1, x_2$ **and** $x_3$, one should look for another, more expressive functional space such that for $i = 0, 1, 2, 3$ the functional update $\Delta f(x_i) := f^{t+1}(x_i) - f^t(x_i)$ goes into the same direction as the functional gradient $\boldsymbol{v}_{\text{goal}}(x_i) := -\nabla_{f(x_i)}\mathcal{L}(f(x_i), y_i) = -2(f(x_i) - y_i)$ where $y_i = f_{\text{true}}(x_i)$.

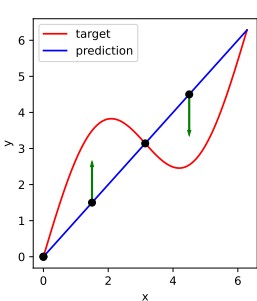

Figure 7: Linear interpolation

## A.5 Problem formulation and choice of pre-activities

There are several ways to design the problem of adding neurons, which we discuss now, in order to explain our choice of the pre-activities to express expressivity bottlenecks.

Suppose one wishes to add $K$ neurons $\theta_\leftrightarrow^K := (\boldsymbol{\alpha}_k, \boldsymbol{\omega}_k)_{k=1}^K$ to layer $l - 1$, which impacts the activities $\boldsymbol{a}_l$ at the next layer, in order to improve its expressivity. These neurons could be chosen to have only nul weights, or nul input weights $\boldsymbol{\alpha}_k$ and non-nul output weights $\boldsymbol{\omega}_k$, or the opposite, or both non-nul weights. Searching for the best neurons to add for each of these cases will produce different optimization problems.

Let us remind first that adding such $K$ neurons with weights $\theta_\leftrightarrow^K := (\boldsymbol{\alpha}_k, \boldsymbol{\omega}_k)_{k=1}^K$ changes the activities $\boldsymbol{a}_l$ of the (next) layer by

$$\delta\boldsymbol{a}_l \;=\; \sum_{k=1}^K \boldsymbol{\omega}_k\,\sigma(\boldsymbol{b}_{l-2}(\boldsymbol{x})^T\boldsymbol{\alpha}_k) \tag{8}$$

**Small weights approximation** Under the hypothesis of small input weights $\alpha_k$, the activity variation 8 can be approximated by:

$$\sigma'(0)\sum_{k=1}^K \boldsymbol{\omega}_k\boldsymbol{b}_{l-2}(\boldsymbol{x})^T\boldsymbol{\alpha}_k$$

at first order in $\|\boldsymbol{\alpha}_k\|$. We will drop the constant $\sigma'(0)$ in the sequel.

This quantity is linear both in $\boldsymbol{\alpha}_k$ and $\boldsymbol{\omega}_k$, therefore the first-order parameter-induced activity variations are easy to compute:

$$\boldsymbol{v}^l(\boldsymbol{x}, (\boldsymbol{\alpha}_k)_{k=1}^K) = \frac{\partial \boldsymbol{a}_l(\boldsymbol{x})}{\partial(\,(\boldsymbol{\alpha}_k)_{k=1}^K\,)}_{|(\boldsymbol{\alpha}_k)_{k=1}^K=0}(\boldsymbol{\alpha}_k)_{k=1}^K = \sum_{k=1}^K \boldsymbol{\omega}_k \boldsymbol{b}_{l-2}(\boldsymbol{x})^T \boldsymbol{\alpha}_k$$

$$\boldsymbol{v}^l(\boldsymbol{x}, (\boldsymbol{\omega}_k)_{k=1}^K) = \frac{\partial \boldsymbol{a}_l(\boldsymbol{x})}{\partial(\,(\boldsymbol{\omega}_k)_{k=1}^K\,)}_{|(\boldsymbol{\omega}_k)_{k=1}^K=0}(\boldsymbol{\omega}_k)_{k=1}^K = \sum_{k=1}^K \boldsymbol{\omega}_k \boldsymbol{b}_{l-2}(\boldsymbol{x})^T \boldsymbol{\alpha}_k$$

so with a slight abuse of notation we have:

$$\boldsymbol{v}^l(\boldsymbol{x}, \theta_\leftrightarrow^K) = \sum_{k=1}^K \boldsymbol{\omega}_k \boldsymbol{b}_{l-2}(\boldsymbol{x})^T \boldsymbol{\alpha}_k$$

Note also that technically the quantity above is first-order in $\boldsymbol{\alpha}_k$ and in $\boldsymbol{\omega}_k$ but second-order in the joint variable $\theta_\leftrightarrow^K = (\boldsymbol{\alpha}_k, \boldsymbol{\omega}_k)$.

**Adding neurons with 0 weights (both input and output weights).** In that case, one increases the number of neurons in the layer, but without changing the function (since only nul quantities are added) and also without changing the gradient with respect to the parameters, thus not improving expressivity. Indeed, the added quantity (Eq. 8) involves $0 \times 0$ multiplications, and consequently the derivative $\frac{\partial \boldsymbol{a}_l(\boldsymbol{x})}{\partial \theta_\leftrightarrow^K}\Big|_{\theta_\leftrightarrow^K=0}$ w.r.t. these new parameters, that is, $\boldsymbol{b}_{l-2}(\boldsymbol{x})^T \boldsymbol{\alpha}_k$ w.r.t. $\boldsymbol{\omega}_k$ and $\boldsymbol{\omega}_k \, \boldsymbol{b}_{l-2}(\boldsymbol{x})^T$ w.r.t. $\boldsymbol{a}_k$ is 0, as both $\boldsymbol{a}_k$ and $\boldsymbol{\omega}_k$ are 0.

**Adding neurons with non-0 input weights and 0 output weights or the opposite.** In these cases, the addition of neurons will not change the function (because of multiplications by 0), but just the gradient. One of the 2 gradients (w.r.t. $\boldsymbol{a}_k$ or w.r.t $\boldsymbol{\omega}_k$) will be non-0, as the variable that is 0 has non-0 derivatives.

The question is then how to pick the best non-0 variable, ($\boldsymbol{a}_k$ or $\boldsymbol{\omega}_k$) such that the added gradient will be the most useful. The problem can then be formulated similarly as what is done in the paper.

**Adding neurons with small yet non-0 weights.** In this case, both the function and its gradient will change when adding the neurons. Fortunately, Proposition 3.2 states that the best neurons to add in terms of expressivity (to get the gradient closer to the variation desired by the backpropagation) are also the best neurons to add to decrease the loss, i.e. the function change they will imply goes into the right direction.

For each family $(\boldsymbol{\omega}_k)_{k=1}^K$, the tangent space in $\boldsymbol{a}_l$ restricted to the family $(\boldsymbol{\alpha}_k)_{k=1}^K$, ie $\mathcal{T}_\mathcal{A}^{\boldsymbol{a}_l} := \{\frac{\partial \boldsymbol{a}_l}{\partial(\boldsymbol{\alpha}_k)_{k=1}^K}|_{(\boldsymbol{\alpha}_k)_{k=1}^K=0}(.)(\boldsymbol{\alpha}_k)_{k=1}^K|(\boldsymbol{\alpha}_k)_{k=1}^K \in (\mathbb{R}^{|\boldsymbol{b}_{l-2}(\boldsymbol{x})|})^K\}$ varies with the family $(\boldsymbol{\omega}_k)_{k=1}^K$, ie $\mathcal{T}_\mathcal{A}^{\boldsymbol{a}_l} := \mathcal{T}_\mathcal{A}^{\boldsymbol{a}_l}((\boldsymbol{\omega}_k)_{k=1}^K)$. Optimizing w.r.t. the $\boldsymbol{\omega}_k$ is equivalent to search for the best tangent space for the $\boldsymbol{\alpha}_k$, while symmetrically optimizing w.r.t. the $\boldsymbol{\alpha}_k$ is equivalent to find the best projection on the tangent space defined by the $\boldsymbol{\omega}_k$.

**Pre-activities vs. post-activities.** The space of pre-activities $\boldsymbol{a}_l$ is a natural space for this framework, as they are formed with linear operations and we compute first-order variation quantities. Considering the space of post-activities $\boldsymbol{b}_l = \sigma(\boldsymbol{a}_l)$ is also possible, though computing variations will be more complex. Indeed, without first-order approximation, the obtained problem is not manageable, because of the non-linear activation function $\sigma$ added in front of all quantities (while in the case of pre-activations, quantity 8 is linear in $\boldsymbol{\omega}_k$ and thus does not require approximation in $\boldsymbol{\omega}_k$, which allow considering large $\boldsymbol{\omega}_k$), and, with first-order approximation, it would add the derivative of the activation function, taken at various locations $\sigma'(\boldsymbol{a}_l)$ (while in the previous case the derivatives of the activation function were always taken at 0).

### A.6 ADDING CONVOLUTIONAL NEURONS

To add a convolutional neuron at layer $l-1$, one should add a kernel at layer $l-1$ and expand one dimension to all the kernels in layer $l$ to match the new dimension of the post-activity.

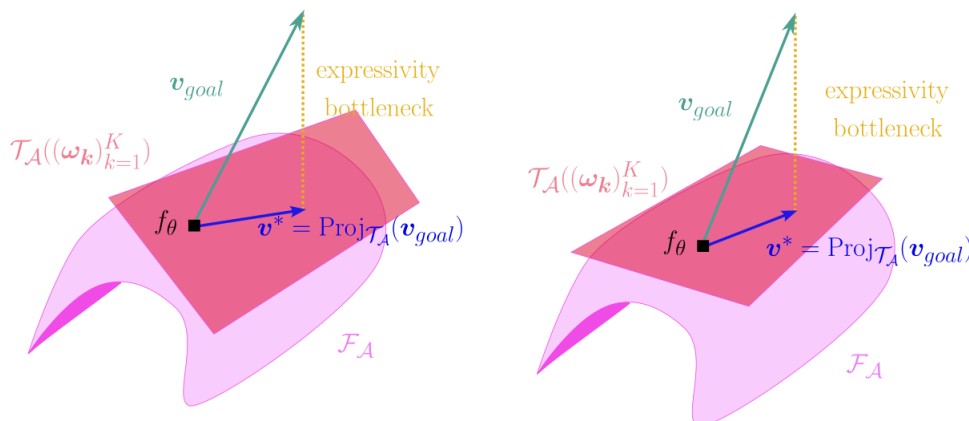

Figure 8: Changing the tangent space with different values of $(\omega_k)_{k=1}^{K}$.

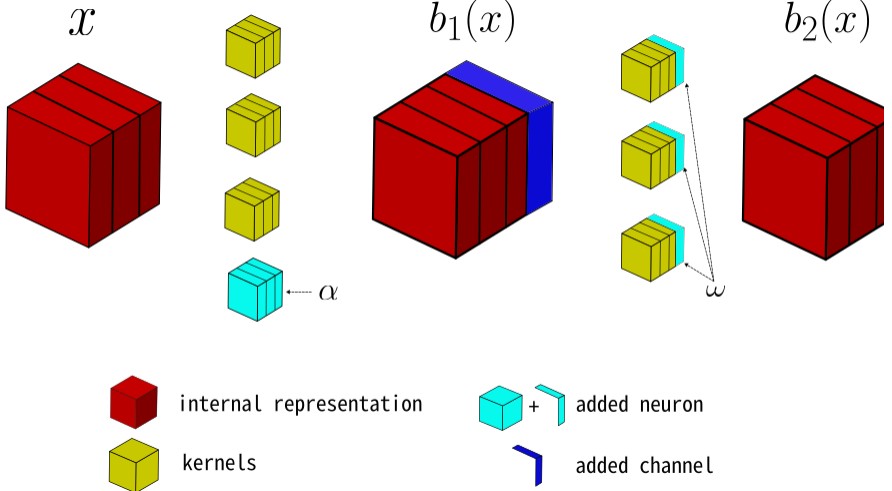

Figure 9: Adding one convolutional neuron at layer one for a input with tree channels.

# B    THEORETICAL COMPARISON WITH OTHER APPROACHES

## B.1    GRADMAX METHOD

Using the notation of the paper Evci et al. (2022) : $\boldsymbol{z}_l(x) := \boldsymbol{a}_l(x)$ and $\boldsymbol{h}_l(x) := \boldsymbol{b}_l(x)$. We have that $\frac{\partial \mathcal{L}}{\partial \boldsymbol{z}_l}(x) := \boldsymbol{v}_{\text{goal}}^l(x)$. When adding $K$ neurons at layer $l$, fan-in weights $\{\boldsymbol{\alpha}_l\}_{j=1}^{k}$ are initialized to zeros and fan-out weights are initialized as the solution of the following optimization problem :

$$
(\boldsymbol{\omega}_1^*, ..., \boldsymbol{\omega}_K^*) := \Omega^* = \arg\max_{\Omega} ||\sum_i \boldsymbol{b}_{l-2}(\boldsymbol{x}_i)\boldsymbol{v}_{\text{goal}}^{l+1}{}^T(x_i)\Omega||_F^2 \qquad s.t.\ ||\Omega||_F^2 < c
$$

$$
:= \arg\max_{\Omega} ||\boldsymbol{B}_{l-2}\boldsymbol{V}_{\text{goal}}^{l+1}{}^T\Omega||_F^2 \qquad s.t.\ ||\Omega||_F^2 < c
$$

$$
:= \arg\max_{\Omega} \text{Tr}\left(\Omega^T \tilde{\boldsymbol{N}}^T \tilde{\boldsymbol{N}}\Omega\right) \qquad s.t.\ ||\Omega||_F^2 < c
$$

While our method is equivalent to the following optimisation problem :

$$
\begin{aligned}
(\boldsymbol{\omega}_1^*, ..., \boldsymbol{\omega}_K^*) = \Omega^* &= \arg\max_{\Omega} || \sum_i \boldsymbol{b}_{l-2}(x_i) \boldsymbol{v}_{\text{goal}_{proj}}^{l+1\ T}(x_i)\Omega ||_F^2 && s.t.\ \ ||\boldsymbol{B}_{l-1}\Omega||_F^2 < c \\
&= \arg\max_{\Omega} || \boldsymbol{B}_{l-2} \boldsymbol{V}_{\text{goal}_{proj}}^{l\ T} \Omega ||_F^2 && s.t.\ \ ||\boldsymbol{B}_{l-1}\Omega||_F^2 < c \\
&= \arg\max_{\hat{\Omega}} \operatorname{Tr}\left( \tilde{\Omega}^T \boldsymbol{S}^{-\frac{1}{2}} \boldsymbol{N}^T \boldsymbol{N} \boldsymbol{S}^{-\frac{1}{2}} \tilde{\Omega} \right) && s.t.\ \ ||\tilde{\Omega}||_F^2 < c,\ \tilde{\Omega} := \boldsymbol{S}^{\frac{1}{2}}\Omega \\
&= \arg\max_{\Omega} \operatorname{Tr}\left( \Omega \boldsymbol{S}^{-1} \boldsymbol{N}^T \boldsymbol{N} \boldsymbol{S}^{-1} \Omega \right) && s.t.\ \ ||\Omega||_F^2 < \tilde{c}
\end{aligned}
$$

One can note three differences between those optimization problems:

- First, the matrix $\tilde{\boldsymbol{N}}$ is not defined using the projection of the desired update $\boldsymbol{V}_{\text{goal}_{proj}}^{l+1}$. As a consequence, GradMax does not take into account redundancy, and on the opposite will actually try to add new neurons that are as redundant as possible with the part of the goal update that is already feasible with already-existing neurons.

- Second, the constraint lies in the weight space for GradMax method while it lies in the pre-activation space in our case. The difference is that GradMax relies on the Euclidean metric in the space of parameters, which arguably offers less meaning that the Euclidean metric in the space of activities. Essentially this is the same difference as between the standard L2 gradient w.r.t. parameters and the natural gradient, which takes care of parameter redundancy and measures all quantities in the output space in order to be independent from the parameterization. In practice we do observe that the "natural" gradient direction improves the loss better than the usual L2 gradient.

- Third, our fan-in weights are not set to 0 but directly to their optimal values (at first order).

## B.2 NORTH PREACTIVATION

In paper Maile et al. (2022), fan-out weights are initialized to 0 while fan-in weights are initialized as $\boldsymbol{\alpha}_i = \boldsymbol{S}^{-1} \boldsymbol{B}_{l-1} \mathbf{V}_{\mathbf{Z}_l}^T r_i$ where $r_i$ is a random vector and $\mathbf{V}_{\mathbf{Z}_l} \in \mathcal{M}(|Ker(\boldsymbol{B}_{l-1}^T)|, |\boldsymbol{b}_{l-1}(\boldsymbol{x})|)$ is a matrix consisting of the orthogonal vectors of the kernel of pre-activations $\boldsymbol{B}_l$, i.e $\{z \mid \boldsymbol{B}_l^T z = 0\}$. In our paper fan-in weights are initialized as $\boldsymbol{\alpha}_i = \boldsymbol{S}^{-1} \boldsymbol{B}_{l-1} \boldsymbol{V}_{\text{goal}_{proj}}^T \boldsymbol{v}_i = \boldsymbol{S}^{-1} \boldsymbol{B}_{l-1} \boldsymbol{V}_{\text{goal}}^T \mathbf{V}_{\mathbf{Z}_l} \mathbf{V}_{\mathbf{Z}_l}^T \boldsymbol{v}_i$, where the $\boldsymbol{v}_i$ are right eigenvectors of the matrix $\boldsymbol{S}^{-\frac{1}{2}} \boldsymbol{N}$.

The main difference is thus that we use the backpropagation to find the best $\boldsymbol{v}_i$ or $r_i$ directly, while the NORTH approach tries random directions $r_i$ to explore the space of possible neuron additions.

## C PROOFS OF PART 3 AND 4

### C.1 PROPOSITION 3.1

Denoting by $\delta \mathbf{W}_l^+$ the generalized (pseudo-)inverse of $\delta \mathbf{W}_l$, we have:

$$
\delta \boldsymbol{W}_l^* = \frac{1}{n} \boldsymbol{V}_{\text{goal}}^l \boldsymbol{B}_{l-1}^T \left( \frac{1}{n} \boldsymbol{B}_{l-1} \boldsymbol{B}_{l-1}^T \right)^+ \text{ and } \boldsymbol{V}_0^l = \frac{1}{n} \boldsymbol{V}_{\text{goal}}^l \boldsymbol{B}_{l-1}^T \left( \frac{1}{n} \boldsymbol{B}_{l-1} \boldsymbol{B}_{l-1}^T \right)^+ \boldsymbol{B}_{l-1}
$$

*Proof*
Consider the function

$$
g(\delta \mathbf{W}_l) = || \boldsymbol{V}_{\text{goal}}^l - \delta \mathbf{W}_l \boldsymbol{B}_{l-1} ||_{\text{Tr}}^2 \tag{9}
$$

then:

$$
\begin{aligned}
g(\delta\mathbf{W}_l + \boldsymbol{H}) &= ||\boldsymbol{V}_{\text{goal}}{}^l - \delta\mathbf{W}_l\boldsymbol{B}_{l-1} - \boldsymbol{H}\boldsymbol{B}_{l-1}||^2_{\text{Tr}} \\
&= g(\delta\mathbf{W}_l) - 2\langle\boldsymbol{V}_{\text{goal}}{}^l - \delta\mathbf{W}_l\boldsymbol{B}_{l-1}, \boldsymbol{H}\boldsymbol{B}_{l-1}\rangle_{\text{Tr}} + o(||\boldsymbol{H}||) \\
&= g(\delta\mathbf{W}_l) - 2\operatorname{Tr}\left(\left(\boldsymbol{V}_{\text{goal}}{}^l - \delta\mathbf{W}_l\boldsymbol{B}_{l-1}\right)^T\boldsymbol{H}\boldsymbol{B}_{l-1}\right) + o(||\boldsymbol{H}||) \\
&= g(\delta\mathbf{W}_l) - 2\operatorname{Tr}\left(\boldsymbol{B}_{l-1}\left(\boldsymbol{V}_{\text{goal}}{}^l - \delta\mathbf{W}_l\boldsymbol{B}_{l-1}\right)^T\boldsymbol{H}\right) + o(||\boldsymbol{H}||) \\
&= g(\delta\mathbf{W}_l) - 2\langle\left(\boldsymbol{V}_{\text{goal}}{}^l - \delta\mathbf{W}_l\boldsymbol{B}_{l-1}\right)\boldsymbol{B}_{l-1}^T, \boldsymbol{H}\rangle_{\text{Tr}} + o(||\boldsymbol{H}||)
\end{aligned}
$$

By identification $\nabla_{\delta\mathbf{W}_l}g(\delta\mathbf{W}_l) = 2\left(\boldsymbol{V}_{\text{goal}}{}^l - \delta\mathbf{W}_l\boldsymbol{B}_{l-1}\right)\boldsymbol{B}_{l-1}^T$, and thus:

$$
\nabla_{\delta\mathbf{W}_l}g(\delta\mathbf{W}_l) = 0 \implies \boldsymbol{V}_{\text{goal}}{}^l\boldsymbol{B}_{l-1}^T = \delta\mathbf{W}_l\boldsymbol{B}_{l-1}\boldsymbol{B}_{l-1}^T
$$

Using the definition of the generalized inverse of $M^+$, we get:

$$
\delta\boldsymbol{W}_l^* = \frac{1}{n}\boldsymbol{V}_{\text{goal}}{}^l\boldsymbol{B}_{l-1}^T\left(\frac{1}{n}\boldsymbol{B}_{l-1}\boldsymbol{B}_{l-1}^T\right)^+
$$

as one solution. For convolutional layers, considering 2D images of size $p$, using index $i = 1, ..., n$ for the samples, and index $k = 1, ..., ch$ for the out-channels, and considering the convolutional kernel size to be $(2, 2)$, then :

$$
\boldsymbol{b}_i^k = \begin{pmatrix} b_i^{1,k} & b_i^{2,k} & \cdot & b_i^{p,k} \\ b_i^{p+1,k} & b_i^{p+2,k} & \cdot & b_i^{2p,k} \\ \cdot & & \cdot & \cdot \\ b_i^{p(p-1)+1,k} & \cdot & \cdot & b_i^{p^2,k} \end{pmatrix} \tag{10}
$$

$$
\boldsymbol{B}_i^c = \begin{pmatrix} b_i^{1,1} & b_i^{2,1} & b_i^{p+1,1} & b_i^{p+2,1} & b_i^{1,2} & b_i^{2,2} & b_i^{p+1,2} & b_i^{p+2,2} & b_i^{1,3} & \cdot \\ b_i^{2,1} & b_i^{3,1} & b_i^{p+2,1} & b_i^{p+3,1} & b_i^{2,2} & b_i^{3,2} & b_i^{p+2,2} & b_i^{p+3,2} & b_i^{2,3} & \cdot \\ \cdot & \cdot & \cdot & \cdot & \cdot & \cdot & \cdot & \cdot & \cdot & \cdot \end{pmatrix}
$$

Then the function to minimize is

$$
g(\delta\mathbf{W}_l) = ||\boldsymbol{V}_{\text{goal}}{}^l - \boldsymbol{B}^c\delta\mathbf{W}_l||^2_{\text{Tr}}
$$

where $\boldsymbol{B}^c := \begin{pmatrix} \boldsymbol{B}_1^c & ... & \boldsymbol{B}_n^c \end{pmatrix}$. $\qquad\square$

---

## C.2 PROPOSITION 3.2

We define the matrices $\boldsymbol{N} := \frac{1}{n}\boldsymbol{B}_{l-2}\left(\boldsymbol{V}_{\text{goal}\,proj}{}^l\right)^T$ and $\boldsymbol{S} := \frac{1}{n}\boldsymbol{B}_{l-2}\boldsymbol{B}_{l-2}^T$. Let us denote its SVD by $\boldsymbol{S} = \boldsymbol{U}\Sigma\boldsymbol{U}^T$, and note $S^{-\frac{1}{2}} := \boldsymbol{U}\sqrt{\Sigma}^{-1}\boldsymbol{U}^T$ and consider the SVD of the matrix $S^{-\frac{1}{2}}\boldsymbol{N} = \sum_{k=1}^R\lambda_k\boldsymbol{u}_k\boldsymbol{v}_k^T$ with $\lambda_1 \geq ... \geq \lambda_R \geq 0$, where $R$ is the rank of the matrix $\boldsymbol{N}$. Then:

**Proposition C.1** (3.2). *The solution of (5) can be written as:*

- *optimal number of neurons: $K^* = R$*

- *their optimal weights: $(\boldsymbol{\alpha}_k^*, \boldsymbol{\omega}_k^*) = (sign(\lambda_k)\sqrt{\lambda_k}S^{-\frac{1}{2}}\boldsymbol{u}_k, \sqrt{\lambda_k}\boldsymbol{v}_k)$ for $k = 1, ..., R$.*

*Moreover for any number of neurons $K \leqslant R$, and associated scaled weights $\theta_{\leftrightarrow}^{K,*}$, the expressivity gain and the first order in $\eta$ of the loss improvement due to the addition of these $K$ neurons are equal and can be quantified very simply as a function of the eigenvalues $\lambda_k$:*

$$
\Psi^l_{\theta\oplus\theta_{\leftrightarrow}^{K,*}} = \Psi^l_\theta - \sum_{k=1}^K\lambda_k^2
$$

$$
\mathcal{L}(f_{\theta\oplus\theta_{\leftrightarrow}^{K,*}}) = \mathcal{L}(f_\theta) + \frac{\sigma_l'(0)}{\eta}\sum_{k=1}^K\lambda_k^2 + o(||\theta_{\leftrightarrow}^{K,*}||^2)
$$

*Proof*

We first compute the proof for a linear layer.

The optimal neurons $n_1, ..., n_K$ are defined by $n_i := (\boldsymbol{\alpha}_i, \boldsymbol{\omega}_i)$ and are the solution of the optimization problem :

$$\underset{\delta \boldsymbol{W}_l, \delta \theta_{l-1 \leftrightarrow l}^K}{\arg\min} \; ||\overbrace{\boldsymbol{V}_{\text{goal}}^l - \delta \boldsymbol{W}_l \boldsymbol{B}_{l-1}}^{\boldsymbol{V}_{\text{goal}}^l{}_{proj}} - \boldsymbol{\Omega} \boldsymbol{A}^T \boldsymbol{B}_{l-2}||_{\text{Tr}}^2$$

$$\text{where } \boldsymbol{\Omega} := (\boldsymbol{\omega}_1 \quad ... \quad \boldsymbol{\omega}_K) \; \text{ and } \; \boldsymbol{A} := (\boldsymbol{\alpha}_1 \quad ... \quad \boldsymbol{\alpha}_K)$$

This is equivalent to :

$$\underset{\boldsymbol{C} = \boldsymbol{\Omega} \boldsymbol{A}^T}{\arg\min} ||\overbrace{\boldsymbol{V}_{\text{goal}}^l - \delta \boldsymbol{W}_l \boldsymbol{B}_{l-1}}^{\boldsymbol{V}_{\text{goal}}^l{}_{proj}} - \boldsymbol{C} \boldsymbol{B}_{l-2}||_{\text{Tr}}^2 \tag{11}$$

Then $\boldsymbol{C}^* = \boldsymbol{S}^+ \boldsymbol{N}$. Taking $K = rank(\boldsymbol{S}^+ \boldsymbol{N})$ and a family $(\boldsymbol{\alpha}_k, \boldsymbol{\omega}_k)_{1,...,R}$ such that $\boldsymbol{\Omega} \boldsymbol{A}^T = \sum_k \boldsymbol{\omega}_k \alpha_k^T = \boldsymbol{S}^+ \boldsymbol{N}$ is a optimal solution. But what if we decide to only add $K < R$ neurons ?

$$\underset{\theta_{\leftrightarrow}^K}{\arg\min} \left\{ \frac{1}{n} ||\boldsymbol{V}_{\text{goal}}^l{}_{proj} - \boldsymbol{V}^l(\theta_{\leftrightarrow}^K)||_{\text{Tr}}^2 \right\} = \underset{\theta_{\leftrightarrow}^K}{\arg\min} \left\{ -\frac{2}{n} \left\langle \boldsymbol{V}_{\text{goal}}^l{}_{proj}, \boldsymbol{V}^l(\theta_{\leftrightarrow}^K) \right\rangle_{\text{Tr}} + \frac{1}{n} ||\boldsymbol{V}^l(\theta_{\leftrightarrow}^K)||_{\text{Tr}}^2 \right\}$$

$$= \underset{\theta_{\leftrightarrow}^K}{\arg\min} \frac{1}{n} g(\theta_{\leftrightarrow}^K)$$

We note

$$\frac{1}{n} g(\theta_{\leftrightarrow}^K) = -\frac{2}{n} \sum_i^n \sum_k \boldsymbol{v}_{\text{goal}}^l{}_{proj}(\boldsymbol{x}_i)^T \left( \boldsymbol{\alpha}_k^T \boldsymbol{b}_{l-2}(\boldsymbol{x}_i) \right) \boldsymbol{\omega}_k$$

$$+ \frac{1}{n} \sum_{k,j}^K \sum_i^n \left( \boldsymbol{\alpha}_k^T \boldsymbol{b}_{l-2}(\boldsymbol{x}_i) \right) \boldsymbol{\omega}_k^T \boldsymbol{\omega}_j \left( \boldsymbol{\alpha}_j^T \boldsymbol{b}_{l-2}(\boldsymbol{x}_i) \right)$$

$$= -2 \sum_k^K \boldsymbol{\alpha}_k^T \left( \frac{1}{n} \sum_i^n \boldsymbol{b}_{l-2}(\boldsymbol{x}_i) \boldsymbol{v}_{\text{goal}}^l{}_{proj}(\boldsymbol{x}_i)^T \right) \boldsymbol{\omega}_k$$

$$+ \sum_{k,j}^K \boldsymbol{\omega}_k^T \boldsymbol{\omega}_j \boldsymbol{\alpha}_k^T \left( \frac{1}{n} \sum_i^n \boldsymbol{b}_{l-2}(\boldsymbol{x}_i) \boldsymbol{b}_{l-2}(\boldsymbol{x}_i)^T \right) \boldsymbol{\alpha}_j$$

$$= -2 \sum_k^K \boldsymbol{\alpha}_k^T \boldsymbol{N} \boldsymbol{\omega}_k + \sum_{k,j}^K \boldsymbol{\omega}_k^T \boldsymbol{\omega}_j \boldsymbol{\alpha}_k^T \boldsymbol{S} \boldsymbol{\alpha}_j$$

with $\boldsymbol{N} := \frac{1}{n} \boldsymbol{B}_{l-2} \left( \boldsymbol{V}_{\text{goal}}^l{}_{proj} \right)^T$ and $\boldsymbol{S} := \frac{1}{n} \boldsymbol{B}_{l-2} \boldsymbol{B}_{l-2}^T$.

Consider the SVD of $\boldsymbol{S} = \boldsymbol{U} \boldsymbol{\Sigma} \boldsymbol{U}^T$. Define $\boldsymbol{S}^{\frac{1}{2}} := \boldsymbol{U} \sqrt{\boldsymbol{\Sigma}} \boldsymbol{U}$ and $\boldsymbol{S}^{-\frac{1}{2}} := \boldsymbol{U} \sqrt{\boldsymbol{\Sigma}}^{-1} \boldsymbol{U}^T$.

Consider also the SVD of $\boldsymbol{S}^{-\frac{1}{2}} \boldsymbol{N} = \sum_{r=1}^R \lambda_r \boldsymbol{v}_r \boldsymbol{e}_r^T$.

Note also $\boldsymbol{\gamma}_k := \boldsymbol{S}^{\frac{1}{2}}{}^T \boldsymbol{\alpha}_k$. Then :

$$-\sum_{k=1}^K \boldsymbol{\alpha}_k^T \boldsymbol{N} \boldsymbol{\omega}_k = -\sum_k \boldsymbol{\gamma}_k^T \boldsymbol{S}^{-\frac{1}{2}} \boldsymbol{N} \boldsymbol{\omega}_k$$

$$= -\text{Tr} \left( \sum_k \sum_r \lambda_r \left( \boldsymbol{\gamma}_k^T \boldsymbol{v}_r \boldsymbol{e}_r^T \right) \boldsymbol{\omega}_k \right)$$

Using the linearity of the Trace and that $\text{Tr}(AB) = \text{Tr}(BA)$, we have :

$$
\begin{aligned}
-\sum_{k=1}^{K} \boldsymbol{\alpha}_k^T N \boldsymbol{\omega}_k &= -\text{Tr}\Big(\sum_k \sum_r \lambda_r \boldsymbol{\omega}_k \boldsymbol{\gamma}_k^T \boldsymbol{v}_r \boldsymbol{e}_r^T\Big) \\
&= -\text{Tr}\Big(\sum_k \boldsymbol{\omega}_k \boldsymbol{\gamma}_k^T \sum_r \lambda_r \boldsymbol{v}_r \boldsymbol{e}_r^T\Big) \\
&= -\Big\langle \sum_k \boldsymbol{\gamma}_k \boldsymbol{\omega}_k^T, \sum_r \lambda_r \boldsymbol{v}_r \boldsymbol{e}_r^T\Big\rangle_{\text{Tr}} \quad \text{with } \langle \boldsymbol{A}, \boldsymbol{B}\rangle_{\text{Tr}} = \text{Tr}(\boldsymbol{A}^T \boldsymbol{B})
\end{aligned}
$$

For the second sum :

$$
\begin{aligned}
\sum_{k,j}^{K} \boldsymbol{\omega}_k^T \boldsymbol{\omega}_j \boldsymbol{\alpha}_k^T S \boldsymbol{\alpha}_j &= \sum_{k,j} \big(\boldsymbol{\omega}_k^T \boldsymbol{\omega}_j\big)\big(\boldsymbol{\gamma}_j^T \boldsymbol{\gamma}_k\big) \\
&= \text{Tr}\Big(\sum_{k,j} \big((\boldsymbol{\omega}_k^T \boldsymbol{\omega}_j)\boldsymbol{\gamma}_j^T\big)\boldsymbol{\gamma}_k\big)\Big) \\
&= \text{Tr}\Big(\big(\sum_{k,j} \boldsymbol{\gamma}_k \boldsymbol{\omega}_k^T \boldsymbol{\omega}_j \boldsymbol{\gamma}_j^T\big)\Big) \\
&= \|\sum_k \boldsymbol{\omega}_k \boldsymbol{\gamma}_k^T\|_{\text{Tr}}^2 \quad \text{with } \|\boldsymbol{A}\|_{\text{Tr}} = \sqrt{\text{Tr}(\boldsymbol{A}^T \boldsymbol{A})} \\
&= \|\sum_k \boldsymbol{\gamma}_k \boldsymbol{\omega}_k^T\|_{\text{Tr}}^2
\end{aligned}
$$

Then we have :

$$
\underset{K, \theta_{\leftrightarrow}}{\arg\min} \ \frac{1}{n} g(\boldsymbol{\alpha}, \boldsymbol{\omega}) = \underset{K, \boldsymbol{\alpha}=\boldsymbol{S}^{-1/2^T}\boldsymbol{\gamma}, \boldsymbol{\omega}}{\arg\min} \left\|\boldsymbol{S}^{-1/2}\boldsymbol{N} - \sum_{k=1}^{K} \boldsymbol{\gamma}_k \boldsymbol{\omega}_k^T\right\|_{\text{Tr}}^2
$$

The solution of such problems is given by the paper Eckart & Young (1936), by choosing $K = \text{rank}(\boldsymbol{S}^{-1/2}\boldsymbol{N})$ and $\sum_{k=1}^{K} \boldsymbol{\gamma}_k \boldsymbol{\omega}_k^T = \sum_{r=1}^{K} \lambda_r \boldsymbol{v}_r \boldsymbol{e}_r^T$. Choosing $K = R$ is thus the best option. Thus we have that $\sum_k^R \boldsymbol{\omega}_k \boldsymbol{\alpha}_k^T = \boldsymbol{S}^{-1/2} \sum_k^R \lambda_k \boldsymbol{\gamma}_k \boldsymbol{\omega}_k^T = \boldsymbol{S}^{-1}\boldsymbol{N}$.

We now consider the matrix $\boldsymbol{S}^{-1/2}\boldsymbol{N}$. The minimization also yields the following properties at the optimum:

$$
\textbf{for } k \neq j, \quad \langle \boldsymbol{\gamma}_k \boldsymbol{\omega}_k^T, \boldsymbol{\gamma}_j \boldsymbol{\omega}_j^T\rangle_{\text{Tr}} = 0
$$

$$
\begin{aligned}
\|\boldsymbol{S}^{-1/2}\boldsymbol{N} - \sum_{k=1}^{K} \boldsymbol{\gamma}_k \boldsymbol{\omega}_k^T\|_{\text{Tr}}^2 &= \sum_{r=K+1}^{R} \lambda_r^2 \\
&= \|\boldsymbol{S}^{-1/2}\boldsymbol{N}\|_{\text{Tr}}^2 - \|\sum_{k=1}^{K} \boldsymbol{\gamma}_k \boldsymbol{\omega}_k^T\|_{\text{Tr}}^2
\end{aligned}
$$

Furthermore :

$$
\begin{aligned}
\frac{1}{n}\|\boldsymbol{V}_{\text{goal}\,proj}^l - \boldsymbol{V}(\theta_{\leftrightarrow}^{K,*})\|_{\text{Tr}}^2 &= \frac{1}{n}\|\boldsymbol{V}_{\text{goal}\,proj}^l\|_{\text{Tr}}^2 + \|\boldsymbol{S}^{-\frac{1}{2}}\boldsymbol{N} - \sum_k \boldsymbol{\gamma}_k \boldsymbol{\omega}_k^T\|^2 - \|\boldsymbol{S}^{-\frac{1}{2}}\boldsymbol{N}\|_{\text{Tr}}^2 \\
&= \sum_{r=K+1}^{R} \lambda_r^2 + \frac{1}{n}\|\boldsymbol{V}_{\text{goal}\,proj}^l\|_{\text{Tr}}^2 - \|\boldsymbol{S}^{-\frac{1}{2}}\boldsymbol{N}\|_{\text{Tr}}^2 \\
&= -\sum_{r=1}^{K} \lambda_r^2 + \frac{1}{n}\|\boldsymbol{V}_{\text{goal}\,proj}^l\|_{\text{Tr}}^2
\end{aligned}
$$

We note $\boldsymbol{V}_{\text{goal}_{proj}}^{l}(\delta \boldsymbol{W}_l^*) := \boldsymbol{V}_{\text{goal}_{proj}}^{l}$. Suppose that $\boldsymbol{B}_{l-1}$ and $\boldsymbol{B}_{l-2}$ are orthogonal for the trace scalar product, then when adding the new neurons, the impact on the global loss is :

$$\mathcal{L}(f_{\theta \oplus \theta_\leftrightarrow^K}) = \frac{1}{n}\sum_{i=1}^{n} L(f_\theta(\boldsymbol{x}_i), \boldsymbol{y}_i) - \frac{\gamma}{\eta}\frac{1}{n}\sigma_l'(0)\Big\langle \boldsymbol{V}^l(\theta_\leftrightarrow^{K,*}), \boldsymbol{V}_{\text{goal}_{proj}}^{l}\Big\rangle_{\text{Tr}} + o(1)$$

We also have the following property :

$$\underset{\theta_\leftrightarrow^K}{\arg\min}\left\{\frac{1}{n}||\boldsymbol{V}_{\text{goal}_{proj}}^{l} - \boldsymbol{V}^l(\theta_\leftrightarrow^K)||_{\text{Tr}}^2\right\}$$

$$= \underset{H \geq 0}{\arg\min}\ \underset{\theta_\leftrightarrow^K, ||\boldsymbol{V}^l(\theta_\leftrightarrow^K)||_{\text{Tr}}=H}{\arg\min}\left\{\frac{1}{n}||\boldsymbol{V}_{\text{goal}_{proj}}^{l} - \boldsymbol{V}^l(\theta_\leftrightarrow^K)||_{\text{Tr}}^2\right\}$$

$$= \underset{H \geq 0}{\arg\min}\ \underset{\theta_\leftrightarrow^K, ||\boldsymbol{V}^l(\theta_\leftrightarrow^K)||_{\text{Tr}}=H}{\arg\min}\left\{-\frac{2}{n}\Big\langle \boldsymbol{V}_{\text{goal}_{proj}}^{l}(, \boldsymbol{V}^{l+1}(\theta_\leftrightarrow^K)\Big\rangle_{\text{Tr}} + \frac{1}{n}||\boldsymbol{V}_K(\theta_\leftrightarrow^K)||_{\text{Tr}}^2\right\}$$

$$= \underset{H \geq 0}{\arg\min}\ \underset{\theta_\leftrightarrow^K, ||\boldsymbol{V}^l(\theta_\leftrightarrow^K)||_{\text{Tr}}=H}{\arg\min}\left\{-\frac{2}{n}\Big\langle \boldsymbol{V}_{\text{goal}_{proj}}^{l}, \boldsymbol{V}^{l+1}(\theta_\leftrightarrow^K)\Big\rangle_{\text{Tr}} + \frac{1}{n}H^2\right\}$$

$$= \underset{H \geq 0}{\arg\min}\ \underset{\theta_\leftrightarrow^K, ||\boldsymbol{V}^l(\theta_{l\leftrightarrow}^K)^*||_{\text{Tr}}=1}{\arg\min}\left\{-H\Big\langle \boldsymbol{V}_{\text{goal}_{proj}}^{l}, \boldsymbol{V}^{l+1}(\theta_{l\leftrightarrow}^K)*\Big\rangle_{\text{Tr}} + \frac{1}{2}H^2\right\}$$

with $\boldsymbol{V}^l(\theta_\leftrightarrow^K)^*$ the solution of the second arg min (ie for $H = 1$).
Then the norm minimizing the first argmin is given by :

$$H^* = \Big\langle \boldsymbol{V}_{\text{goal}_{proj}}^{l}, \boldsymbol{V}^l(\theta_\leftrightarrow^K)^*\Big\rangle_{\text{Tr}}$$

Furthermore

$$\underset{\theta_{l\leftrightarrow l}^K}{\min}\left\{\frac{1}{n}||\boldsymbol{V}_{\text{goal}_{proj}}^{l} - \boldsymbol{V}^{l+1}(\theta_{l\leftrightarrow l}^K)||_{\text{Tr}}^2\right\} = -\sum_{r=1}^{K}\lambda_r^2 + \frac{1}{n}||\boldsymbol{V}_{\text{goal}_{proj}}^{l}||_{\text{Tr}}^2$$

$$\underset{\theta_\leftrightarrow^K}{\min}\left\{\frac{1}{n}||\boldsymbol{V}_{\text{goal}_{proj}}^{l} - \boldsymbol{V}^l(\theta_{l\leftrightarrow l}^K)||_{\text{Tr}}^2\right\} = -\frac{1}{n}H^{*2} + \frac{1}{n}||\boldsymbol{V}_{\text{goal}_{proj}}^{l}||_{\text{Tr}}^2$$

$$\implies H^* = \Big\langle \boldsymbol{V}_{\text{goal}_{proj}}^{l}, \boldsymbol{V}^l(\theta_\leftrightarrow^K)^*\Big\rangle_{\text{Tr}} = \sqrt{\sum_{r=1}^{K}\lambda_r^2} \times \sqrt{n}$$

$$\boldsymbol{V}^l(\theta_\leftrightarrow^{K,*}) = H^*\boldsymbol{V}^l(\theta_\leftrightarrow^K)^*$$

$$\Big\langle \boldsymbol{V}^l(\theta_\leftrightarrow^{K,*}), \boldsymbol{V}_{\text{goal}_{proj}}^{l}\Big\rangle_{\text{Tr}} = H^* \times \Big\langle \boldsymbol{V}_{\text{goal}_{proj}}^{l}, \boldsymbol{V}^l(\theta_\leftrightarrow^K)^*\Big\rangle_{\text{Tr}} = H^{*2}$$

where the last equality is given by the optimisation of $||\boldsymbol{S}^{-\frac{1}{2}}\boldsymbol{N} - \sum_{k=1}^{K}\boldsymbol{u}_k\boldsymbol{\omega}_k^T||_{\text{Tr}}^2$. So minimizing the scalar product $-\Big\langle \boldsymbol{V}_{\text{goal}_{proj}}^{l}(\delta\boldsymbol{W}_l^*), \boldsymbol{V}^l(\theta_\leftrightarrow^K)^*\Big\rangle_{\text{Tr}}$ for fixed norm of $\boldsymbol{V}^l(\theta_\leftrightarrow^K)$ is equivalent to minimizing the norm $||\boldsymbol{V}_{\text{goal}_{proj}}^{l}(\delta\boldsymbol{W}_l^*) - \boldsymbol{V}^l(\theta_\leftrightarrow^K)||_{\text{Tr}}^2$.

$$\mathcal{L}(f_{\theta \oplus \theta_\leftrightarrow^K}) = \frac{1}{n}\sum_{i=1}^{n}\mathcal{L}(f_\theta(\boldsymbol{x}_i), \boldsymbol{y}_i) - \frac{1}{\eta}\sigma_l'(0)\sum_{r=1}^{K}\lambda_k^2 + o(1)$$

For convolutional layers, the same reasoning can be applied. Consider one adds one convolution layer, ie $K = 1$. Use $\boldsymbol{B}_i^c$ defined in the first proof and $\boldsymbol{T}_j$ the linear application selecting the

activities for the $j-$pixel , then one has to minimize the expression :

$$g(\theta_{\leftrightarrow}^1) = \overset{\text{out channels}}{\underset{m}{\sum}} \overset{\text{examples}}{\underset{i}{\sum}} \overset{\text{preactivity size}}{\underset{j=1}{\sum}} (\boldsymbol{\omega}_m^T \boldsymbol{T}_j \boldsymbol{B}^c \boldsymbol{\alpha} - \boldsymbol{V}_{\text{goal}}{}_{j,m}^i)^2$$

$$= \sum_m \sum_i \sum_{j=1} (\boldsymbol{\omega}_m^T \boldsymbol{T}_j \boldsymbol{B}_i^c \boldsymbol{\alpha})^2 - 2\boldsymbol{\omega}_m^T \boldsymbol{T}_j \boldsymbol{B}_i^c \boldsymbol{\alpha} \boldsymbol{V}_{\text{goal}}{}_{j,m}^i + C$$

$$= \sum_m \sum_i \sum_{j=1} \text{Tr}(\boldsymbol{\omega}_m^T \boldsymbol{T}_j \boldsymbol{B}_i^c \boldsymbol{\alpha})^2 - 2\boldsymbol{\omega}_m^T \boldsymbol{T}_j \boldsymbol{B}_i^c \boldsymbol{\alpha} \boldsymbol{V}_{\text{goal}}{}_{j,m}^i + C$$

$$= \sum_m \sum_i \sum_{j=1} \text{Tr}(\boldsymbol{T}_j \boldsymbol{B}_i^c \boldsymbol{\alpha} \boldsymbol{\omega}_m^T)^2 - 2\boldsymbol{\omega}_m^T \boldsymbol{T}_j \boldsymbol{B}_i^c \boldsymbol{\alpha} \boldsymbol{V}_{\text{goal}}{}_{j,m}^i + C$$

for some constant $C$. We have the property that $\langle \boldsymbol{T}_j^T, \boldsymbol{B}_i^c \boldsymbol{\alpha} \boldsymbol{\omega}_m^T \rangle_{\text{Tr}}^2 = \text{Tr}(\boldsymbol{T}_j \boldsymbol{B}_i^c \boldsymbol{\alpha} \boldsymbol{\omega}_m^T)^2 = ||\boldsymbol{T}_j \boldsymbol{B}_i^c \boldsymbol{\alpha} \boldsymbol{\omega}_m^T||_{\text{Tr}}^2$. Ignoring the constant $C$:

$$g(\theta_{\leftrightarrow}^1) = \sum_m \text{Tr}\Big(\boldsymbol{\omega}_m \boldsymbol{\alpha}^T \Big(\sum_i \boldsymbol{B}_i^{cT} \sum_j \boldsymbol{T}_j^T \boldsymbol{T}_j \boldsymbol{B}_i^c\Big) \boldsymbol{\alpha} \boldsymbol{\omega}_m^T\Big) - 2\boldsymbol{\omega}_m^T \sum_{i,j} \boldsymbol{T}_j \boldsymbol{B}_i^c \boldsymbol{V}_{\text{goal}}{}_{j,m}^i \boldsymbol{\alpha}$$

$$= \sum_m \boldsymbol{\alpha}^T \Big(\sum_i \boldsymbol{B}_i^{cT} \sum_j \boldsymbol{T}_j \boldsymbol{T}_j^T \boldsymbol{B}_i^c\Big) \boldsymbol{\alpha} \boldsymbol{\omega}_m^T \boldsymbol{\omega}_m - 2\boldsymbol{\omega}_m^T \sum_{i,j} \boldsymbol{T}_j \boldsymbol{B}_i^c \boldsymbol{V}_{\text{goal}}{}_{j,m}^i \boldsymbol{\alpha}$$

$$= \boldsymbol{\alpha}^T \Big(\sum_i \boldsymbol{B}_i^{cT} \sum_j \boldsymbol{T}_j \boldsymbol{T}_j^T \boldsymbol{B}_i^c\Big) \boldsymbol{\alpha} \boldsymbol{\omega}^T \boldsymbol{\omega} - 2\sum_m \boldsymbol{\omega}_m^T \sum_{i,j} \boldsymbol{T}_j 1_{full}^T \boldsymbol{V}_{\text{goal}}{}^i 1_{j,m} \boldsymbol{B}_i^c \boldsymbol{\alpha}$$

$$= \boldsymbol{\alpha}^T \boldsymbol{S} \boldsymbol{\omega}^T \boldsymbol{\omega} - 2\sum_m \boldsymbol{\omega}_m^T \sum_i \boldsymbol{F}_i^m \boldsymbol{B}_i^c \boldsymbol{\alpha}$$

$$= \boldsymbol{\alpha}^T \boldsymbol{S} \boldsymbol{\omega}^T \boldsymbol{\omega} - 2\boldsymbol{\omega} \boldsymbol{N} \boldsymbol{\alpha}$$

with $T_j = \begin{pmatrix} \overbrace{0 \ . \ 0}^{j-1+\lfloor j/(p-1)\rfloor} & 1 & 0 & . & . & . & . \\ \overbrace{0 \ . \ 0}^{j-1+\lfloor j/(p-1)\rfloor} & 0 & 1 & . & . & . & . \\ \underset{j-1+\lfloor j/(p-1)\rfloor}{\overbrace{0 \ . \ 0}} & \underset{31}{\overbrace{0 \ . \ 0}} & 1 & 0 & . & . & . \\ \underset{j-1+\lfloor j/(p-1)\rfloor}{\overbrace{0 \ . \ 0}} & \underset{31}{\overbrace{0 \ . \ 0}} & 0 & 1 & 0 & . & . \end{pmatrix}$ for a kernel size equal to $(2,2)$. $\qquad\square$

## C.3 PROPOSITION AND REMARK 3.3

Suppose $\boldsymbol{S}$ is semi definite, we note $\boldsymbol{S} = \boldsymbol{S}^{\frac{1}{2}} \boldsymbol{S}^{\frac{1}{2}}$. Solving (7) is equivalent to find the K first eigenvectors $\boldsymbol{\alpha}_k$ associated to the K largest eigenvalues $\lambda$ of the generalized eigenvalue problem :

$$\boldsymbol{N} \boldsymbol{N}^T \boldsymbol{\alpha}_k = \lambda \boldsymbol{S} \boldsymbol{\alpha}_k$$

*Proof*

This is equivalent to maximizing the following generalized Rayleigh quotient (which is solvable by the LOBPCG technique):

$$\boldsymbol{\alpha}^* = \max_x \frac{\boldsymbol{\alpha}^T \boldsymbol{N} \boldsymbol{N}^T \boldsymbol{\alpha}}{\boldsymbol{\alpha}^T \boldsymbol{S} \boldsymbol{\alpha}}$$

$$\boldsymbol{p}^* = \max_{\boldsymbol{p}=\boldsymbol{S}^{1/2}\boldsymbol{\alpha}} \frac{\boldsymbol{p}^T \boldsymbol{S}^{-\frac{1}{2}} \boldsymbol{N} \boldsymbol{N}^T \boldsymbol{S}^{-\frac{1}{2}} \boldsymbol{p}}{\boldsymbol{p}^T \boldsymbol{p}}$$

$$\boldsymbol{p}^* = \max_{||\boldsymbol{p}||=1} ||\boldsymbol{N}^T \boldsymbol{S}^{-\frac{1}{2}} \boldsymbol{p}||$$

$$\boldsymbol{\alpha}^* = \boldsymbol{S}^{-\frac{1}{2}} \boldsymbol{p}^*$$

Considering the SVD of $\boldsymbol{S}^{-\frac{1}{2}}\boldsymbol{N} = \sum_{r=1}^R \lambda_r \boldsymbol{e}_r \boldsymbol{f}_r^T$, then $\boldsymbol{S}^{-\frac{1}{2}}\boldsymbol{N}\boldsymbol{N}^T\boldsymbol{S}^{-\frac{1}{2}} = \sum_{r=1}^R \lambda_r^2 \boldsymbol{f}_r \boldsymbol{f}_r^T$, because $j \neq i \implies \boldsymbol{e}_i^T \boldsymbol{e}_j = 0$ and $\boldsymbol{f}_i^T \boldsymbol{f}_j = 0$. Hence maximizing the first quantity is equivalent to $\boldsymbol{p}_k^* = \boldsymbol{f}_k$, then $\boldsymbol{\alpha}_k = \boldsymbol{S}^{-\frac{1}{2}}\boldsymbol{e}_k$. The same reasoning is used to find $\boldsymbol{\omega}_k$.

We prove second corollary 3.2 by induction. For $m = m' = 1$ :

$$\boldsymbol{a}_l(\boldsymbol{x})^{t+1} = \boldsymbol{a}_l(\boldsymbol{x})^t + \boldsymbol{V}(\theta_\leftrightarrow^{1,*}, \boldsymbol{x})\gamma + o(\gamma)$$

$$\boldsymbol{v}_{\text{goal}}{}^{l,t+1}(\boldsymbol{x}) = \boldsymbol{v}_{\text{goal}}{}^{l,t}(\boldsymbol{x}) + \nabla_{\boldsymbol{a}_l(\boldsymbol{x})}\mathcal{L}(f_\theta(\boldsymbol{x}), \boldsymbol{y})^T \boldsymbol{v}(\theta_\leftrightarrow^{1,*}, \boldsymbol{x})\gamma + o(\gamma)$$

Adding the second neuron we obtain the minimization problem:

$$\underset{\boldsymbol{\alpha}_2, \boldsymbol{\omega}_2}{\arg\min} ||\boldsymbol{V}_{\text{goal}}{}^{l,t} - \boldsymbol{V}^l(\boldsymbol{\alpha}_2, \boldsymbol{\omega}_2)||_{\text{Tr}} + o(1)$$

$\square$

## C.4 ABOUT EQUIVALENCE OF QUADRATIC PROBLEMS

Problems 6 and 5 are generally not equivalent, but might be very close, depending on layer sizes and number of samples. The difference between the two problems is that in one case one minimizes the quadratic quantity:

$$\left\| \boldsymbol{V}^l(\theta_\leftrightarrow^K) + \boldsymbol{V}^l(\delta\mathbf{W}_l) - \boldsymbol{V}_{\text{goal}}{}^l \right\|_{\text{Tr}}^2$$

w.r.t. $\delta\mathbf{W}_l$ and $\theta_\leftrightarrow^K$ **jointly**, while in the other case the problem is first minimized w.r.t. $\delta\mathbf{W}_l$ and then w.r.t. $\theta_\leftrightarrow^K$. The latter process, being greedy, might thus provide a solution that is not as optimal as the joint optimization.

We chose this two-step process as it intuitively relates to the spirit of improving upon a standard gradient descent: we aim at adding neurons that complement what the other ones have already done. This choice is debatable and one could solve the joint problem instead, with the same techniques.

The topic of this section is to check how close the two problems are. To study this further, note that $\boldsymbol{V}^l(\delta\mathbf{W}_l) = \delta\mathbf{W}_l\boldsymbol{B}_{l-1}$ while $\boldsymbol{V}^l(\theta_\leftrightarrow^K) = \sum_{k=1}^K \boldsymbol{\omega}_k \boldsymbol{B}_{l-2}^T \boldsymbol{\alpha}_k$. The rank of $\boldsymbol{B}_{l-1}$ is $\min(n_S, n_{l-1})$ where $n_S$ is the number of samples and $n_{l-1}$ the number of neurons (post-activities) in layer $l-1$, while the rank of $\boldsymbol{B}_{l-2}$ is $\min(n_S, n_{l-2})$ where $n_{l-2}$ is the number of neurons (post-activities) in layer $l-2$. Note also that the number of degrees of freedom in the optimization variables $\delta\mathbf{W}_l$ and $\theta_\leftrightarrow^K = (\boldsymbol{\omega}_k, \boldsymbol{\alpha}_k)$ is much larger than these ranks.

**Small sample case.** If the number $n_S$ of samples is lower than the number of neurons $n_{l-1}$ and $n_{l-2}$ (which is potentially problematic, see Section D.1), then it is possible to find suitable variables $\delta\mathbf{W}_l$ and $\theta_\leftrightarrow^K$ to form any desired $\boldsymbol{V}^l(\delta\mathbf{W}_l)$ and $\boldsymbol{V}^l(\theta_\leftrightarrow^K)$. In particular, if $n_S \leqslant n_{l-1} \leqslant n_{l-2}$, one can choose $\boldsymbol{V}^l(\theta_\leftrightarrow^K)$ to be $\boldsymbol{V}_{\text{goal}}{}^l - \boldsymbol{V}^l(\delta\mathbf{W}_l)$ and thus cancel any effect due to the greedy process in two steps. The two problems are then equivalent.

**Large sample case.** On the opposite, if the number of samples is very large (compared to the number of neurons $n_{l-1}$ and $n_{l-2}$), then the lines of matrices $\boldsymbol{B}_{l-1}$ and $\boldsymbol{B}_{l-2}$ become asymptotically uncorrelated, under the assumption of their independence (which is debatable, depending on the type of layers and activation functions). Thus the optimization directions available to $\boldsymbol{V}^l(\delta\mathbf{W}_l)$ and $\boldsymbol{V}^l(\theta_\leftrightarrow^K)$ become orthogonal, and proceeding greedily does not affect the result, the two problems are asymptotically equivalent.

In the general case, matrices $\boldsymbol{B}_{l-1}$ and $\boldsymbol{B}_{l-2}$ are not independent, though not fully correlated, and the number of samples (in the minibatch) is typically larger than the number of neurons; the problems are then different.

Note that technically the ranks could be lower, in the improbable case where some neurons are perfectly redundant, or, e.g., if some samples yield exactly the same activities.

## C.5 SECTION *Theory behind Greedy Growth* WITH PROOFS

One might wonder whether a greedy approach on layer growth might get stuck in a non-optimal state. We derive the following series of propositions in this regard. Since in this work we add neurons layer per layer independently, we study here the case of a single hidden layer network, to spot potential layer growth issues. For the sake of simplicity, we consider the task of least square regression towards an explicit continuous target $f^*$, defined on a compact set. That is, we aim at minimizing the loss:

$$\inf_f \sum_{x \in \mathcal{D}} \|f(x) - f^*(x)\|^2$$

where $f(x)$ is the output of the neural network and $\mathcal{D}$ is the training set.

**Proposition C.2** (Greedy completion of an existing network). *If $f$ is not $f^*$ yet, there exists a set of neurons to add to the hidden layer such that the new function $f'$ will have a lower loss than $f$.*

One can even choose the added neurons such that the loss is arbitrarily well minimized.

*Proof.* The classic universal approximation theorem about neural networks with one hidden layer Pinkus (1999) states that for any continuous function $g$ defined on a compact set $\boldsymbol{\omega}$, for any desired precision $\gamma$, and for any activation function $\sigma$ provided it is not a polynomial, then there exists a neural network $\widehat{g}$ with one hidden layer (possibly quite large when $\gamma$ is small) and with this activation function $\sigma$, such that

$$\forall x, \|g(x) - g^*(x)\| \leqslant \gamma$$

We apply this theorem to the case where $g^* = f^* - f$, which is continuous as $f^*$ is continuous, and $f$ is a shallow neural network and as such is a composition of linear functions and of the function $\sigma$, that we will suppose to be continuous for the sake of simplicity. We will suppose that $f$ is real-valued for the sake of simplicity as well, but the result is trivially extendable to vector-valued functions (just concatenate the networks obtained for each output independently). We choose $\gamma = \frac{1}{10}\|f^* - f\|_{L2}$, where $\langle a| b\rangle_{L2} = \frac{1}{|\boldsymbol{\omega}|} \int_{x \in \boldsymbol{\omega}} a(x) b(x)\, dx$. This way we obtain a one-hidden-layer neural network $g$ with activation function $\sigma$ such that:

$$\forall x \in \boldsymbol{\omega}, \;\; -\gamma \leqslant g(x) - g^*(x) \leqslant \gamma$$

$$\forall x \in \boldsymbol{\omega}, \;\; g(x) = f^*(x) - f(x) + a(x)$$

with $\forall x \in \boldsymbol{\omega}, \; |a(x)| \leqslant \gamma$.

Then:

$$\forall x \in \boldsymbol{\omega}, \;\; f^*(x) - (f(x) + g(x)) = -a(x)$$

$$\forall x \in \boldsymbol{\omega}, \;\; (f^*(x) - h(x))^2 = a^2(x) \tag{12}$$

with $h$ being the function corresponding to a neural network consisting in concatenating the hidden layer neurons of $f$ and $g$, and consequently summing their outputs.

$$\|f^* - h\|_{L2}^2 = \|a\|_{L2}^2$$

$$\|f^* - h\|_{L2}^2 \leqslant \gamma^2 = \frac{1}{100}\|f^* - f\|_{L2}^2$$

and consequently the loss is reduced indeed (by a factor of 100 in this construction).

The same holds in expectation or sum over a training set, by choosing $\gamma = \frac{1}{10}\sqrt{\frac{1}{|\mathcal{D}|}\sum_{x \in \mathcal{D}}\|f(x) - f^*(x)\|^2}$, as Equation (12) then yields:

$$\sum_{x \in \mathcal{D}}(f^*(x) - h(x))^2 = \sum_{x \in \mathcal{D}} a^2(x) \leqslant \frac{1}{100}\sum_{x \in \mathcal{D}}(f^*(x) - f(x))^2$$

which proves the proposition as stated.

For more general losses, one can consider order-1 (linear) developpment of the loss and ask for a network $g$ that is close to (the opposite of) the gradient of the loss.

$\square$

*Proof of the additional remark.* The proof in Pinkus (1999) relies on the existence of real values $c_n$ such that the $n$-th order derivatives $\sigma^{(n)}(c_n)$ are not 0. Then, by considering appropriate values arbitrarily close to $c_n$, one can approximate the $n$-th derivative of $\sigma$ at $c_n$ and consequently the polynomial $c^n$ of order $n$. This standard proof then concludes by density of polynomials in continuous functions.

Provided the activation function $\sigma$ is not a polynomial, these values $c_n$ can actually be chosen arbitrarily, in particular arbitrarily close to 0. This corresponds to choosing neuron input weights arbitrarily close to 0. $\square$

**Proposition C.3** (Greedy completion by one single neuron). *If $f$ is not $f^*$ yet, there exists a neuron to add to the hidden layer such that the new function $f'$ will have a lower loss than $f$.*

*Proof.* From the previous proposition, there exists a finite set of neurons to add such that the loss will be decreased. In this particular setting of $L2$ regression, or for more general losses if considering small function moves, this means that the function represented by this set of neurons has a strictly negative component over the gradient $g$ of the loss ($g = 2(f^* - f)$ in the case of the $L2$ regression). That is, denoting by $a_i\sigma(\boldsymbol{W}_i \cdot \boldsymbol{x})$ these $N$ neurons:

$$\big\langle \sum_{i=1}^{N} a_i\sigma(\boldsymbol{w}_i \cdot \boldsymbol{x}) \,\big|\, g \big\rangle_{L2} = K < 0$$

i.e.

$$\sum_{i=1}^{N} \langle a_i\sigma(\boldsymbol{w}_i \cdot \boldsymbol{x})|\, g \rangle_{L2} = K < 0$$

Now, by contradiction, if there existed no neuron $i$ among these ones such that

$$\langle a_i\sigma(\boldsymbol{w}_i \cdot \boldsymbol{x}) \,|\, g \rangle_{L2} \leqslant \frac{1}{N}K$$

then we would have:

$$\forall i \in [1, N], \ \ \langle a_i\sigma(\boldsymbol{w}_i \cdot \boldsymbol{x})|\, g \rangle_{L2} > \frac{1}{N}K$$

$$\sum_{i=1}^{N} \langle a_i\sigma(\boldsymbol{w}_i \cdot \boldsymbol{x})|\, g \rangle_{L2} > K$$

hence a contradiction. Then necessarily at least one of the $N$ neurons satisfies

$$\langle a_i\sigma(\boldsymbol{w}_i \cdot \boldsymbol{x})|\, g \rangle_{L2} \leqslant \frac{1}{N}K < 0$$

and thus decreases the loss when added to the hidden layer of the neural network representing $f$. Moreover this decrease is at least $\frac{1}{N}$ of the loss decrease resulting from the addition of all neurons.

$\square$

As a consequence, our greedy approach will not get stuck in a situation where one would need to add many neurons simultaneously to decrease the loss: it is always feasible by a single neuron. On can express a lower bound on how much the loss has improved (for the best such neuron), but it not a very good one without further assumptions on $f$.

**Proposition C.4** (Greedy completion by one infinitesimal neuron). *The neuron in the previous proposition can be chosen to have arbitrarily small input weights.*

*Proof.* This is straightforward, as, following a previous remark, the neurons found to collectively decrease the loss can be supposed to all have arbitrarily small input weights. $\square$

This detail is important in that our approach is based on the tangent space of the function $f$ and consequently manipulates infinitesimal quantities. Though we perform line search in a second step and consequently add non-infinitesimal neurons, our first optimization problem relies on the linearization of the activation function by requiring the added neuron to have infinitely small input weights, without which it would be much harder to solve. This proposition confirms that such neuron does exist indeed.

# D  TECHNICAL DETAILS

## D.1  VARIANCE OF THE ESTIMATOR AND BATCHSIZE FOR ESTIMATION

In this section we study the variance of the matrices $\delta\mathbf{W}_l^*$ and $\boldsymbol{S}^{-1/2}\boldsymbol{N}$ computed using a minibatch of $n$ samples, seeing the samples as random variables, and the matrices computed as estimators of the true matrices one would obtain by considering the full distribution of samples. Those two matrices are the solutions of the multiple linear regression problems defined in (9) and in (11), as we are trying to regress the desired update noted $Y$ onto the span of the activities noted $X$. We suppose we have the following setting :

$$Y \sim AX + \varepsilon, \;\; \varepsilon \sim \mathcal{N}(0, \sigma^2), \;\; \mathbb{E}[\varepsilon|X] = 0$$

where the $(X_i, Y_i)$ are *i.i.d.* and $A$ is the oracle for $\delta\mathbf{W}_l^*$ or matrix $\boldsymbol{S}^{-1/2}\boldsymbol{N}$. If $Y$ is multidimensional, the the total variance of our estimator can be seen as the sum of the variances of the estimator on each dimension of $Y$.

We now suppose that $Y \in \mathbb{R}$. The estimator $\hat{A} := (\mathbf{X}\mathbf{X}^T)^{-1}\mathbf{X}\mathbf{Y}^T$ has variance $\text{var}(\hat{A}) = \sigma^2(\mathbf{X}\mathbf{X}^T)^{-1}$. If $n$ is large, and if matrix $\frac{1}{n}\mathbf{X}\mathbf{X}^T \to Q$, with $Q$ non singular, then, asymptotically, we have $\hat{A} \sim \mathcal{N}(A, \sigma^2\frac{Q^{-1}}{n})$, which is equivalent to $(\hat{A} - A)\frac{\sqrt{n}}{\sigma}Q^{1/2} \sim \mathcal{N}(0, I)$. Then $||(\hat{A} - A)\frac{\sqrt{n}}{\sigma}Q^{1/2}||^2 \sim \chi^2(k)$ where $k$ is the dimension of $X$. It follows that $\mathbb{E}\left[||(\hat{A} - A)Q^{1/2}||^2\right] = \frac{k\sigma}{n}$ and as $Q^{1/2}Q^{1/2^T}$ is positive definite, we conclude that $\text{var}(\hat{A}) \leqslant \frac{k\sigma}{n\lambda_{min}(Q)}$.

In practice and to keep the variance of our estimators stable during architecture growth, for the estimation of the best neuron to add we use batch size

$$n \propto \frac{(SW)^2}{P},$$

with the notations defined in Figure 10, since the matrices we estimate have side size $SW$ and that each input sample contains $P$ values, i.e. $P$ quantities that each play the role of $X$ here.

## D.2  BATCH SIZE FOR LEARNING

Batchsize, learning rate and number of neurons are known to be related. As our architecture grows with time, we need to adapt the training batchsize as well.

The batch size is set to $b_{t=0} = 500$ at the beginning of each experiment, and it is scheduled to increase as the square root of the complexity of the model (ie number of additions in a test). If at time $t$ the network has complexity $C_t$ parameters, then at time $t + 1$ the training batch size is equal to $b_{t+1} = b_t \times \sqrt{\frac{C_{t+1}}{C_t}}$.

## D.3  SIGNIFICANCE OF THE EIGENVALUES.

Using the SVD on matrix $\boldsymbol{B} = \boldsymbol{U}\boldsymbol{D}\boldsymbol{Q}^T$, we have: $\boldsymbol{S}^{-\frac{1}{2}}\boldsymbol{N} = \boldsymbol{U}{I_d}^{rk(D)}\boldsymbol{Q}^T\boldsymbol{V}_{\text{goal}}$. Thus the $\lambda_k$ are the square roots of the eigenvalues of the matrix $\boldsymbol{N}^T\boldsymbol{S}^{-1}\boldsymbol{N} = \boldsymbol{V}_{\text{goal}}{}^T\boldsymbol{Q}{I_d}^{rk(D)}\boldsymbol{Q}^T\boldsymbol{V}_{\text{goal}}$, which is similar to the covariance matrix of the desired update. To avoid adding neurons that are non significant, we evaluate which amplitude those eigenvalues would have under the hypothesis $(\mathcal{H}_0)$ that $\boldsymbol{V}_{\text{goal}} \sim \mathcal{N}(0, \Sigma)$ is uncorrelated with the projection matrix $\boldsymbol{Q}{I_d}^{rk(D)}\boldsymbol{Q}^T$ (i.e. what values are obtained when estimating eigenvalues that are actually 0).

Remark that under the hypothesis $\mathcal{H}_0$, the eigenvalues of covariance matrix $\boldsymbol{V}_{\text{goal}}{}^T\boldsymbol{V}_{\text{goal}}$ follow the Marchenko-Pastur distribution, which is known in closed form. This enables to set a threshold on eigenvalues based on statistical significance.

Unfortunately regarding the distribution of the eigenvalues of $\boldsymbol{V}_{\text{goal}}{}^T\boldsymbol{Q}{I_d}^{rk(D)}\boldsymbol{Q}^T\boldsymbol{V}_{\text{goal}}$, getting closed-form expression is trickier, so we numerically estimate a threshold $\lambda_-^2$ by generating $v_{\text{gauss}}(X) \sim \mathcal{N}(0, \Sigma)$ and considering $\lambda_-^2 := \lambda_{\max}\left(\boldsymbol{V}_{\text{gauss}}^T\boldsymbol{Q}{I_d}^{rk(D)}\boldsymbol{Q}^T\boldsymbol{V}_{\text{gauss}}\right)$. Conversely, if the sample size required for reasonably estimating an eigenvalue $\lambda_k$ exceeds the dataset size, associated neurons necessarily overfit, and could be chosen not to be added. The overfit risk in this

addition step is thus controlled. However, standard gradient descent performed afterwards cancels such guarantees.

## D.4 AMPLITUDE FACTOR

Once the neurons defined in Proposition 3.2 have been computed, they are added with an amplitude factor to the current architecture, *i.e.* $\boldsymbol{\alpha} \leftarrow \varepsilon_1 \boldsymbol{\alpha}$ and $\boldsymbol{\omega} \leftarrow \varepsilon_2 \boldsymbol{\omega}$, with $\varepsilon_i$ real factors. Those factors have to be chosen such that the optimization process stays smooth, in the sense that the gradient of each parameter keeps being smooth. Otherwise the current learning rates defined by the optimizer are no longer correct and the system can get unstable. In the work of Maile et al. (2022); Evci et al. (2022) the amplitude of the new neurons are set to an arbitrary constant $10^{-4}$. In our paper we normalize $\alpha, \omega$, such that $||\boldsymbol{v}^l(\sqrt{\varepsilon}\alpha, \sqrt{\varepsilon}\omega, \boldsymbol{X})|| = \varepsilon||\boldsymbol{a}^l(X)||$ then a line search is performed in $\varepsilon$. We processed with this pseudo code :

---
**Algorithm 2:** AmplitudeFactor

---
**Data:** $(\alpha_k, \omega_k)_{k=1}^m$
**Result:** amplitude factor $\varepsilon$ to be applied to $\alpha$ and $\omega$
Take a minibatch $\mathbf{X}$ of size 500;
$\varepsilon^* = \arg\min_{\varepsilon=2^{-k},\, k\in\mathbb{N}} \quad \sum_{\boldsymbol{x}\in\boldsymbol{X}} \ell(f_{\theta\oplus(\sqrt{\varepsilon}\boldsymbol{\alpha}_k^*,\sqrt{\varepsilon}\boldsymbol{\omega}_k^*)_k}(\boldsymbol{x}), \boldsymbol{y}(\boldsymbol{x}))$

---

Note that it is also possible to perform line searches to estimate the best amplitude factors at each neuron addition. This improves the loss much faster, however it also yields later training instabilities (due to the need of different learning rates), that yet have to be solved, which is why we do not present this approach variation here. Interestingly this allows training a neural network without gradient descent (i.e. no parameter update, using just backpropagation).

## D.5 FULL ALGORITHM

In this section we decribe in detail the pseudo code of the main paper (Algorithm 1). In its copy below (Algorithm 3), we have replaced the references by the name of the functions used to compute each non-trivial step.

The function NewNeurons($l$, Sp), in Algorithm 4, computes the new neurons defined at Proposition 3.2 for layer $l$. The argument Sp $= True$, for Spurious, changes the desired update into a random variable (useful to estimate statistical relevance). The function call NewNeurons($l$, Sp $= True$) computes a sample of neurons and eigenvalues ($\{\lambda_k^{\mathcal{N}}\}_k$) that one would have obtained if neuron were not to add. The maximum eigenvalues $max(\lambda_k^{\mathcal{N}})$

is the threshold on the eigenvalues from the function call $NewNeurons(l, \mathrm{Sp} = False)$.

---
**Algorithm 3:** Algorithm to plot Figure 5.

---
**for** *each method [Ours, GradMax]* **do**
    Start from a given small neural network $NN$;
    **for** *j in $nb_{pass}$* **do**
        **for** *each layer l* **do**
            $A, \Lambda, \Omega = $ NewNeurons(layer, method = method);
                *// with our method the above also yields $\delta \boldsymbol{W}_l^*$ as a by-product*;
            $\_, \Lambda^{\mathcal{N}}, \_ = $ NewNeurons(layer, method = method, Sp = $True$ );
            $\lambda_- = max(diag(\Lambda^{\mathcal{N}}))$;
            $A, \Lambda, \Omega \leftarrow A[:, : \lambda_-], \Lambda[: \lambda_-, : \lambda_-], \Omega[: \lambda-, :]$;
            **if** $len(A) > 0$ **then**
                $\gamma = $ AmplitudeFactor$(A, \Omega)$;
                **if** $\gamma > 0$ **then**
                    $A, \Omega \leftarrow \sqrt{\gamma}A, \sqrt{\gamma}\Omega$;
                    Add the neurons $A, \Omega$;
                **end**
            **end**
            **if** $method == \ Our$ **then**
                Update the architecture with the best update $\delta \boldsymbol{W}_l^*$ at $l+1$
            **end**
        **end**
    **end**
    Get accuracy $acc_0$ of $NN$;
    Train $NN$ for 15 epochs with Adam(lr = 1e-4) ;
    Get final accuracy $acc_\infty$ of $NN$;
**end**

---

---
**Algorithm 4:** NewNeurons

---
**Data:** $l$, method $= Our$, Sp $= False$
**Result:** Best neurons at $l$
**if** $method = Our$ **then**
    $\delta \mathbf{W}_l = $ BestUpdate$(l + 1)$;
**else**
    $\delta \mathbf{W}_l = None$
**end**
$\boldsymbol{S}, \boldsymbol{N} = $ MatrixSN$(l - 1, l + 1, \delta \mathbf{W}_l = \delta \mathbf{W}_l, \mathrm{Sp} = \mathrm{Sp})$;
Compute the SVD of $\boldsymbol{S} := U\Sigma U^T$;
Compute the SVD of
$U\sqrt{\Sigma}^{-1}U\boldsymbol{N} := \boldsymbol{A}\Lambda\boldsymbol{\Omega}$;
Use the columns of $\boldsymbol{A}$, the ligns of $\boldsymbol{\Omega}$
  and the diagonal of $\Lambda$ to construct the
  new neurons of Prop. 3.2;

---

---
**Algorithm 5:** MatrixSN

---
**Data:** $p_1, p_2$ (layer indexes), $\delta \mathbf{W}_l = $
    None, Sp = False
**Result:** Construct matrices $\boldsymbol{S}$ and $\boldsymbol{N}$
**if** $not(Sp)$ **then**
    Take a minibatch $\mathbf{X}$ of size $\propto \frac{(SW)^2}{P}$;
    Propagate and backpropagate $\mathbf{X}$;
    Compute $\mathbf{V}_{goal}$ at $p_2$, ie $-\frac{\partial \mathcal{L}^{tot}}{\partial \mathbf{A_{p_2}}}$;
    **if** $\delta \mathbf{W}_l \neq None$ **then**
        $\mathbf{V}_{goal} - = \delta \mathbf{W}_l \boldsymbol{B}_{p_1}$
    **end**
**else**
    $\mathbf{V}_{goal} = E, \ E \sim \mathcal{N}(0, I)$
**end**
$\mathbf{S}, \mathbf{N}_{-2} = \boldsymbol{B}_{p_1}\boldsymbol{B}_{p_1}{}^T, \boldsymbol{B}_{p_1}\boldsymbol{V}_{\text{goal}}{}^T$;

---

---
**Algorithm 6:** BestUpdate

---
**Data:** $l$, index of a layer
**Result:** Best update at $l$
Take a minibatch $\mathbf{X}$ of size $\propto \frac{(SW)^2}{P}$;
Compute $(\mathbf{S}, \mathbf{N})$ with the function
  S_N$(l, l)$;
$\delta \mathbf{W}_l = N^T S^{-1}$;

---

### D.6 COMPUTATIONAL COMPLEXITY

We estimate here the computational complexity of the above algorithm for architecture growth.

**Theoretical estimate.** We use the following notations:

- number of layers: $L$
- layer width, or number of kernels if convolutions: $W$ (assuming for simplicity that all layers have same width or kernels)
- number of pixels in the image: $P$ ($P = 1$ for fully-connected)
- kernel filter size: $S$ ($S = 1$ if fully-connected)
- minibatch size used for standard gradient descent: $M$
- minibatch size used for new neuron estimation: $M'$
- minibatch size used in the line-search to estimate amplitude factor: $M''$
- number of classical gradients steps performed between 2 addition tentatives: $T$

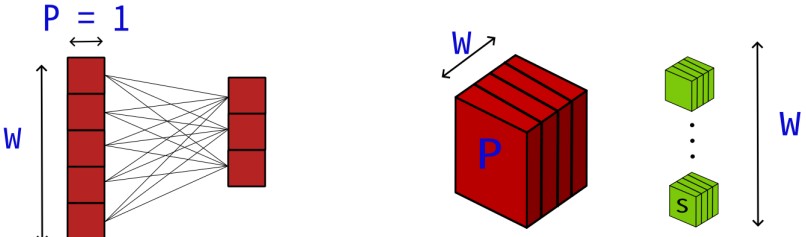

Figure 10: Notation and size for convolutional and linear layers

Complexity, estimated as the number of basic operations, cumulated over all calls of the functions:

- of the standard training part: $TMLW^2SP$
- of the computation of matrices of interest (function MatrixSN): $LM'(SW)^2P$
- of SVD computations (function NewNeurons): $L(SW)^3$
- of line-searches (function AmplitudeFactor): $L^2M''W^2SP$
- of weight updates (function BestUpdate): $LSW$

The relative added complexity w.r.t. the standard training part is thus:

$$M'S\,/\,TM \;\; + \;\; S^2W\,/\,TMP \;\; + \;\; M''L\,/\,TM \;\; + \;\; 1\,/\,WTMP.$$

**SVD cost is negligible.** The relative cost of the SVD w.r.t. the standard training part is $S^2W\,/\,TMP$. In the fully-connected network case, $S = 1$, $P = 1$, and the relative cost of the SVD is then $W/TM$. It is then negligible, as layer width $W$ is usually much smaller than $TM$, which is typically $10 \times 100$ for instance. In the convolutional case, $S = 9$ for $3 \times 3$ kernels, and $P \approx 1000$ for CIFAR, $P \approx 100000$ for ImageNet, so the SVD cost is negligible as long as layer width $W \ll 10000$ or $1\,000\,000$ respectively. So one needs no worrying about SVD cost.

Likewise, the update of existing weights using the "optimal move" (already computed as a by-product) is computationally negligible, and the relative cost of the line searches is limited as long as the network is not extremely deep ($L < TM/M''$).

On the opposite, the estimation of the matrices (to which SVD is applied) can be more ressource demanding. The factor $M'S/TM$ can be large if the minibatch size $M'$ needs to be large for statistical significance reasons. One can show that an upper bound to the value required for $M'$ to ensure estimator precision (see Appendix D.1) is $(SW)^2/P$. In that case, if $W > \sqrt{TMP/S^3}$, these matrix estimations will get costly. In the fully-connected network case, this means $W >$

$\sqrt{TM} \approx 30$ for $T = 10$ and $M = 100$. In the convolutional case, this means $W > \sqrt{TMP/S^3} \approx 30$ for CIFAR and $\approx 300$ for ImageNet. We are working on finer variance estimation and on other types of estimators to decrease $M'$ and consequently this cost. Actually $(SW)^2/P$ is just an upper bound on the value required for $M'$, which might be much lower, depending on the rank of computed matrices.

**In practice.** In practice the cost of a full training with our architecture growth approach is similar (sometimes a bit faster, sometimes a bit slower) than a standard gradient descent training using the final architecture from scratch. This is great as the right comparison should take into account the number of different architectures to try in the classical neural architecture search approach. Therefore we get layer width hyper-optimization for free.

## E ADDITIONAL EXPERIMENTAL RESULTS AND REMARKS

### E.1 MNIST

In this section we work with feed forward networks with two hidden layers. We note by $[a, \ b]$ such network with $a$ neurons on the first hidden layer and $b$ neurons on the second one. The experiments of this section are performed on 7 CPU. We use the optimizer $SGD(lr = 1e - 4)$ and a constant batch of size 100 (we do not apply method in D.2). The training criterion is the cross-entropy loss.

#### E.1.1 RANDOM VS OPTIMIZATION

When performing the quadratic optimization (6), we obtain the optimal direction for $(\boldsymbol{\alpha}_k^*, \boldsymbol{\omega}_k^*)_{k=1}^R$. It is also possible to generate randomly the new neurons and compute the amplitude factors. This second option have the benefit of being less time consuming, but it would project the desired direction on those random vectors and would affect the accuracy score compared to optimal solution defined in 3.1.

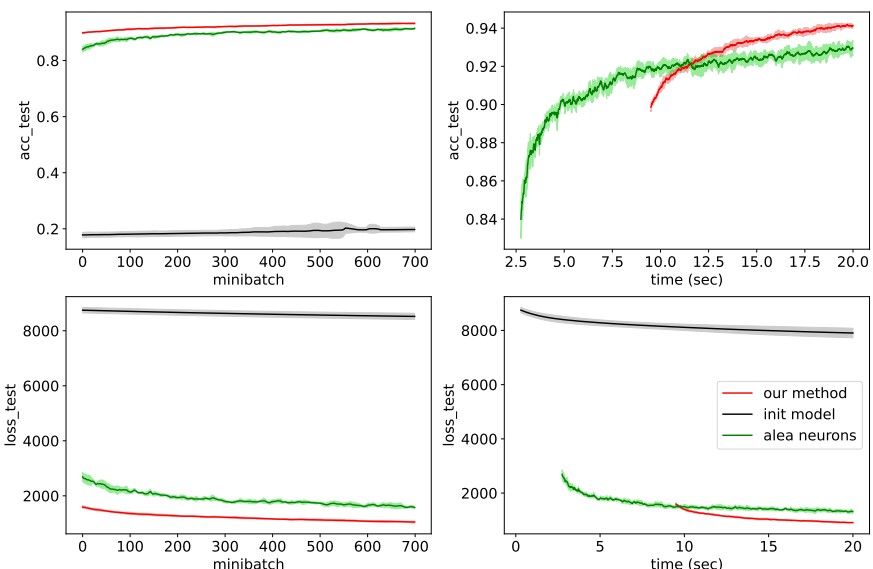

Figure 11: Experiment performed 5 times on the MNIST dataset : a starting model in black $[1, 1]$ is initialized according to normal Kaiming, then is duplicated to give the red and the green model. The structure of the red model is modified by our method to reach the structure $[110, 51]$ while the green model is extended with random neurons. Then all models are trained for 30 seconds. The white space for the red model corresponds to the quadratic optimisation and the computation of the amplitude factor while for the green model it corresponds only to the computation of the amplitude factor.

### E.1.2 COMPARISON WITH BIG MODEL

On figure 12, we plot mean and standard deviation for the same experiement repeated 5 times. We start with a network of size $[1, 1]$ and we increase its architecture by applying our method six times. Each increase of architecture is followed by a training period of 0.5 seconds. Once the network has reached its final structure of $[110, 51]$ it is trained for a limited time of 25 seconds. We compared our performance with a huge network of size $[1000, 1000]$ which is trained with all its neurons from scratch. Compared to the big model our TINY network converge faster in time because its architecture is smaller, converge slower in epochs because it is less expressive.

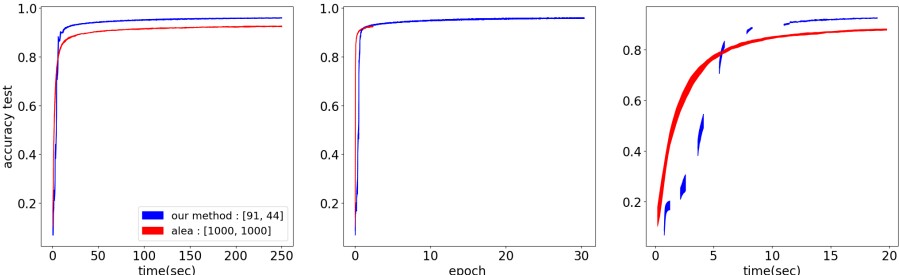

Figure 12: All graphics represent the values for accuracy on test of the same experiment but from a different perspective. ***Left*** : after partitioning computational time on intervals of size 0.1 seconds, we compute a linear interpolation for the accuracy value. ***Middle*** : the accuracy value against number of epochs, where the time needed to compute the optimal neurons is not noticeable. ***Right*** : accuracy against computational time, where durations due to Cholesky decompositions and their happening instants are averaged over experiments, for better visualisation purposes.

### E.1.3 COMPARISON WITH THE SAME STRUCTURE RETRAINED FROM SCRATCH

In figure 13 we compare our method with a neural network retrained from scratch with the same architecture. The protocol is the same as defined in section E.1.2.

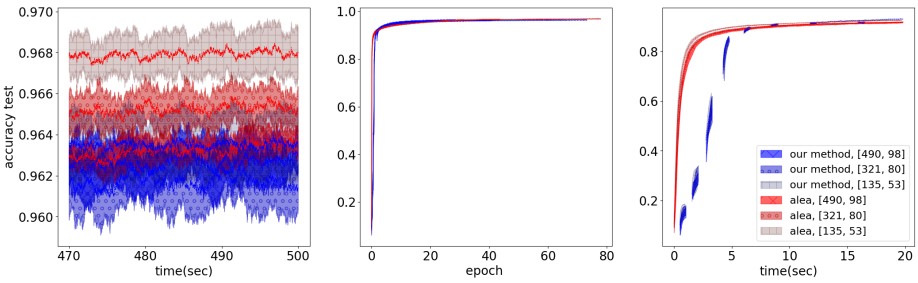

Figure 13: Same description as 12 but for different architecture and that the architecture of the classical training matches the architecture of our method.

### E.2 CIFAR-10

In this section we work with convolutional network forward networks with an architecture of two blocks consisting of 2 convolutions and 1 MaxPooling each, followed by two fully-connected layers, using the selu activation function. The training criterion is the cross-entropy loss. We increase the size of the network on the fly (during training), with a batch size of 500 for training using D.2 on 1 GPU.

### E.2.1 ABOUT NUMBER OF NEURONS AND OVERFITTING

In Figure 14, 100% accuracy is achieved on training dataset, thus fully overfitting the data, which proves that TINY can effectively bypass any expressivity bottleneck. Interestingly, this is done with fewer parameters than the theorem in Zhang et al. (2017), which mentions $2n + d$ parameters to overfit a classification problem with $n$ samples of dimension $d$. This TINY architecture has about twice fewer parameters, due to generalization across samples, the labels in the CIFAR-10 classification task being not random. It should also be noted that the TINY architecture tends to overfit less than the same final model retrained with all neurons from the beginning (FS), or at least, never more, which suggests that optimization and generalization abilities are not necessarily functions of neural network width anymore if one leaves the standard fixed-architecture training paradigm.

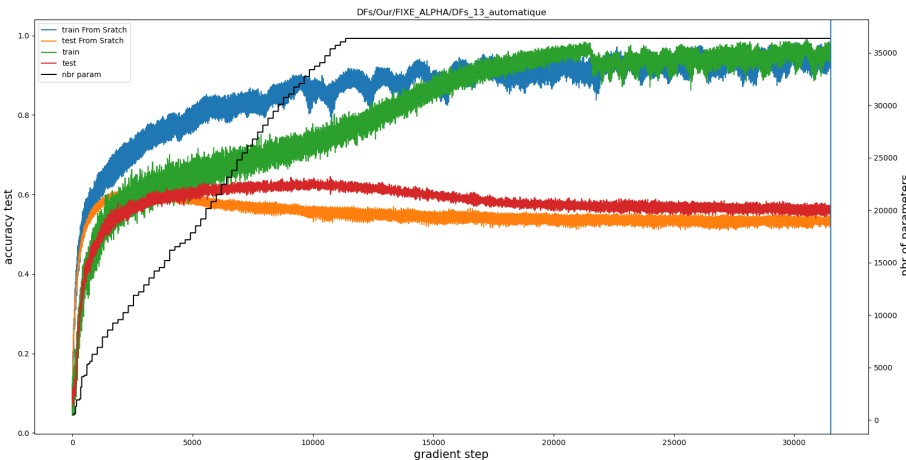

Figure 14: Experiment on CIFAR-10 dataset : Evolution of the train/test accuracy wrt gradient steps for TINY and FS

### E.3 RESNET18 ON CIFAR-100

In this section we compare TINY to GradMax, on CIFAR-100 with the ResNet-18 architecure. For this particular architecture, the size of a convolutional layer $l$ may be increase if no skip connection feeds the output of the layer $l + 1$, see Figure 17.
For both methods we start with the architecture shown in Table 1, initialized with Kaiming Normal. Every 0.1 epoch of standard training with $Adam(batchsize = 100, lr = 1e - 3)$, we add neurons where it is the most needed by:

1. computing the 5 best new neurons $(\alpha_k, \omega_k)_{k=1}^5$ and their amplitude factor $\gamma$ for each layer
2. evaluating the gain of loss associated to each potential addition
3. adding the neurons where the decrease of loss is the largest.

The performances of the models are registered at each gradient step during standard training and after each attempt of architecture increase, while the complexity of the model (as number of basic operation performed at test time) is only evaluated after each addition trial. Figure 15 plots the performance of both methods and the computational cost of models at step $t$, averaged over three runs. After 2300 steps the performance of TINY is higher than GradMax' one (+2%) but its complexity is much lower (a third less). This dramatic difference can be explained by the fact that TINY avoids redundancy when adding neurons, while GradMax does not.

The accuracies reached in this experiment are far from the state of the art on this dataset, but one has to keep in mind that we performed no optimization hyper-parameter tuning and that we did

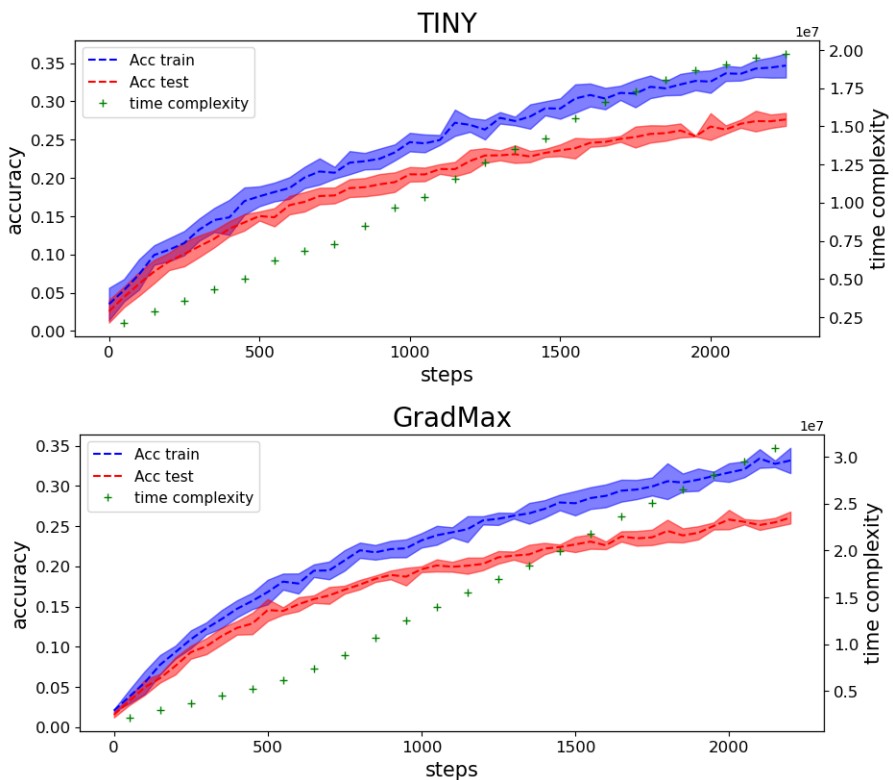

Figure 15: Accuracy and time complexity of ResNet-18 model for TINY and GradMax methods.

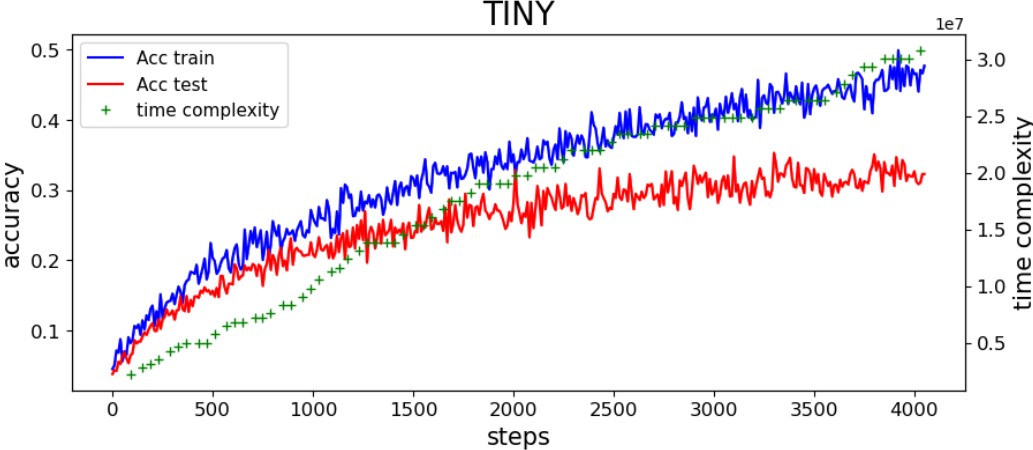

Figure 16: Accuracy and time complexity of ResNet-18 model for one run with TINY method.

not use techniques such as batch-norm, drop-out or data augmentation, which are necessary for convolutional models to go beyond 30-40% accuracy. Additional Figure 16 with twice more training steps shows that with our approach the network keeps learning afterwards (reaching 32% accuracy on test). Note that the number of parameters is very low compared to traditional architectures, for example the model of Figure 16 has $119\ 866$ parameters at $step = 4025$. The final architecture obtained is shown in Table 1. We see that neurons were mostly added to first layers.

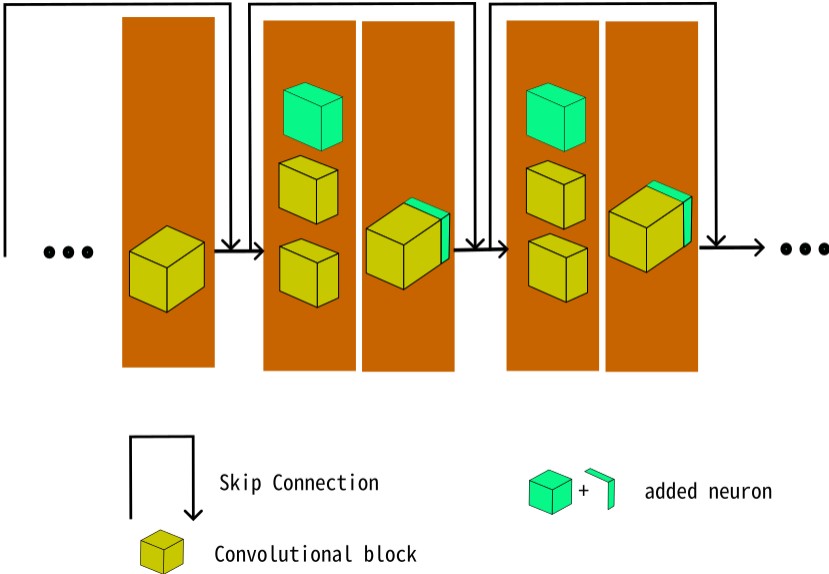

Figure 17: ResNet blocks: in green, the convolutions of the current structure; in cyan, the added convolutions.

Table 1: Initial architecture in the experiments of Figure 15 and 16, and final architecture at the end of training in Figure 16. Numbers in color indicate where TINY was allowed to add neurons (middle of ResNet blocks).

| ResNet18 | | | |
|---|---|---|---|
| Layer name | Output size | Initial layers (kernel=(3,3), padd.=1) | Final layers (end of Fig 16) |
| Conv 1 | $32 \times 32 \times 8$ | $\begin{bmatrix} 3 \times 3, 8 \end{bmatrix}$ | $\begin{bmatrix} 3 \times 3, 8 \end{bmatrix}$ |
| Conv 2 | $32 \times 32$ | $\begin{bmatrix} 3 \times 3, 8 \\ 3 \times 3, 1 \end{bmatrix} \begin{bmatrix} 3 \times 3, 1 \\ 3 \times 3, 8 \end{bmatrix}$ | $\begin{bmatrix} 3 \times 3, 8 \\ 3 \times 3, 91 \end{bmatrix} \begin{bmatrix} 3 \times 3, 91 \\ 3 \times 3, 8 \end{bmatrix}$ |
| Conv 3 | $16 \times 16 \times 8$ | $\begin{bmatrix} 3 \times 3, 8 \\ 3 \times 3, 1 \end{bmatrix} \begin{bmatrix} 3 \times 3, 1 \\ 3 \times 3, 8 \end{bmatrix}$ | $\begin{bmatrix} 3 \times 3, 8 \\ 3 \times 3, 76 \end{bmatrix} \begin{bmatrix} 3 \times 3, 76 \\ 3 \times 3, 8 \end{bmatrix}$ |
| Conv 4 | $16 \times 16 \times 8$ | $\begin{bmatrix} 3 \times 3, 16 \end{bmatrix}$ | $\begin{bmatrix} 3 \times 3, 16 \end{bmatrix}$ |
| Conv 5 | $16 \times 16 \times 16$ | $\begin{bmatrix} 3 \times 3, 16 \\ 3 \times 3, 1 \end{bmatrix} \begin{bmatrix} 3 \times 3, 1 \\ 3 \times 3, 16 \end{bmatrix}$ | $\begin{bmatrix} 3 \times 3, 16 \\ 3 \times 3, 41 \end{bmatrix} \begin{bmatrix} 3 \times 3, 41 \\ 3 \times 3, 16 \end{bmatrix}$ |
| Conv 6 | $8 \times 8 \times 16$ | $\begin{bmatrix} 3 \times 3, 16 \\ 3 \times 3, 1 \end{bmatrix} \begin{bmatrix} 3 \times 3, 1 \\ 3 \times 3, 16 \end{bmatrix}$ | $\begin{bmatrix} 3 \times 3, 16 \\ 3 \times 3, 11 \end{bmatrix} \begin{bmatrix} 3 \times 3, 11 \\ 3 \times 3, 16 \end{bmatrix}$ |
| Conv 7 | $8 \times 8 \times 32$ | $\begin{bmatrix} 3 \times 3, 32 \end{bmatrix}$ | $\begin{bmatrix} 3 \times 3, 32 \end{bmatrix}$ |
| Conv 8 | $8 \times 8 \times 32$ | $\begin{bmatrix} 3 \times 3, 32 \\ 3 \times 3, 1 \end{bmatrix} \begin{bmatrix} 3 \times 3, 1 \\ 3 \times 3, 32 \end{bmatrix}$ | $\begin{bmatrix} 3 \times 3, 32 \\ 3 \times 3, 6 \end{bmatrix} \begin{bmatrix} 3 \times 3, 6 \\ 3 \times 3, 32 \end{bmatrix}$ |
| Conv 9 | $4 \times 4 \times 32$ | $\begin{bmatrix} 3 \times 3, 32 \\ 3 \times 3, 1 \end{bmatrix} \begin{bmatrix} 3 \times 3, 1 \\ 3 \times 3, 32 \end{bmatrix}$ | $\begin{bmatrix} 3 \times 3, 32 \\ 3 \times 3, 3 \end{bmatrix} \begin{bmatrix} 3 \times 3, 3 \\ 3 \times 3, 32 \end{bmatrix}$ |
| Conv 10 | $4 \times 4 \times 64$ | $\begin{bmatrix} 3 \times 3, 64 \end{bmatrix}$ | $\begin{bmatrix} 3 \times 3, 64 \end{bmatrix}$ |
| Conv 11 | $4 \times 4 \times 64$ | $\begin{bmatrix} 3 \times 3, 64 \\ 3 \times 3, 1 \end{bmatrix} \begin{bmatrix} 3 \times 3, 1 \\ 3 \times 3, 64 \end{bmatrix}$ | $\begin{bmatrix} 3 \times 3, 64 \\ 3 \times 3, 1 \end{bmatrix} \begin{bmatrix} 3 \times 3, 1 \\ 3 \times 3, 64 \end{bmatrix}$ |
| Conv 12 | $2 \times 2 \times 64$ | $\begin{bmatrix} 3 \times 3, 64 \\ 3 \times 3, 1 \end{bmatrix} \begin{bmatrix} 3 \times 3, 1 \\ 3 \times 3, 64 \end{bmatrix}$ | $\begin{bmatrix} 3 \times 3, 64 \\ 3 \times 3, 1 \end{bmatrix} \begin{bmatrix} 3 \times 3, 1 \\ 3 \times 3, 64 \end{bmatrix}$ |
| FC 1 | 100 | $256 \times 100$ | $256 \times 100$ |
| FC 2 | 100 | $100 \times 100$ | $256 \times 100$ |
| FC 3 | 100 | $100 \times 100$ | $100 \times 100$ |
| SoftMax | 100 | | |

