}^1) = \overbrace{\sum_m}^{\text{out channels}} \overbrace{\sum_i}^{\text{examples}} \overbrace{\sum_{j=1}^{\text{preactivity size}}} (\boldsymbol{\omega}_m^T \boldsymbol{T}_j \boldsymbol{B}^c \boldsymbol{\alpha} - \boldsymbol{V}_{\text{goal}\,j,m}^i)^2$$

$$= \sum_m \sum_i \sum_{j=1} (\boldsymbol{\omega}_m^T \boldsymbol{T}_j \boldsymbol{B}_i^c \boldsymbol{\alpha})^2 - 2\boldsymbol{\omega}_m^T \boldsymbol{T}_j \boldsymbol{B}_i^c \boldsymbol{\alpha} \boldsymbol{V}_{\text{goal}\,j,m}^i + C$$

$$= \sum_m \sum_i \sum_{j=1} \text{Tr}(\boldsymbol{\omega}_m^T \boldsymbol{T}_j \boldsymbol{B}_i^c \boldsymbol{\alpha})^2 - 2\boldsymbol{\omega}_m^T \boldsymbol{T}_j \boldsymbol{B}_i^c \boldsymbol{\alpha} \boldsymbol{V}_{\text{goal}\,j,m}^i + C$$

$$= \sum_m \sum_i \sum_{j=1} \text{Tr}(\boldsymbol{T}_j \boldsymbol{B}_i^c \boldsymbol{\alpha} \boldsymbol{\omega}_m^T)^2 - 2\boldsymbol{\omega}_m^T \boldsymbol{T}_j \boldsymbol{B}_i^c \boldsymbol{\alpha} \boldsymbol{V}_{\text{goal}\,j,m}^i + C$$

for some constant $C$. We have the property that $\langle \boldsymbol{T}_j^T, \boldsymbol{B}_i^c \boldsymbol{\alpha} \boldsymbol{\omega}_m^T \rangle_{\text{Tr}}^2 = \text{Tr}(\boldsymbol{T}_j \boldsymbol{B}_i^c \boldsymbol{\alpha} \boldsymbol{\omega}_m^T)^2 = ||\boldsymbol{T}_j \boldsymbol{B}_i^c \boldsymbol{\alpha} \boldsymbol{\omega}_m^T||_{\text{Tr}}^2$. Ignoring the constant $C$:

$$g(\theta_{\leftrightarrow}^1) = \sum_m \text{Tr}\Big(\boldsymbol{\omega}_m \boldsymbol{\alpha}^T \Big(\sum_i \boldsymbol{B}_i^{cT} \sum_j \boldsymbol{T}_j^T \boldsymbol{T}_j \boldsymbol{B}_i^c\Big) \boldsymbol{\alpha} \boldsymbol{\omega}_m^T\Big) - 2\boldsymbol{\omega}_m^T \sum_{i,j} \boldsymbol{T}_j \boldsymbol{B}_i^c \boldsymbol{V}_{\text{goal}\,j,m}^i \boldsymbol{\alpha}$$

$$= \sum_m \boldsymbol{\alpha}^T \Big(\sum_i \boldsymbol{B}_i^{cT} \sum_j \boldsymbol{T}_j \boldsymbol{T}_j^T \boldsymbol{B}_i^c\Big) \boldsymbol{\alpha} \boldsymbol{\omega}_m^T \boldsymbol{\omega}_m - 2\boldsymbol{\omega}_m^T \sum_{i,j} \boldsymbol{T}_j \boldsymbol{B}_i^c \boldsymbol{V}_{\text{goal}\,j,m}^i \boldsymbol{\alpha}$$

$$= \boldsymbol{\alpha}^T \Big(\sum_i \boldsymbol{B}_i^{cT} \sum_j \boldsymbol{T}_j \boldsymbol{T}_j^T \boldsymbol{B}_i^c\Big) \boldsymbol{\alpha} \boldsymbol{\omega}^T \boldsymbol{\omega} - 2\sum_m \boldsymbol{\omega}_m^T \sum_{i,j} \boldsymbol{T}_j 1_{full}^T \boldsymbol{V}_{\text{goal}}^i 1_{j,m} \boldsymbol{B}_i^c \boldsymbol{\alpha}$$

$$= \boldsymbol{\alpha}^T \boldsymbol{S} \boldsymbol{\omega}^T \boldsymbol{\omega} - 2\sum_m \boldsymbol{\omega}_m^T \sum_i \boldsymbol{F}_i^m \boldsymbol{B}_i^c \boldsymbol{\alpha}$$

$$= \boldsymbol{\alpha}^T \boldsymbol{S} \boldsymbol{\omega}^T \boldsymbol{\omega} - 2\boldsymbol{\omega} \boldsymbol{N} \boldsymbol{\alpha}$$

with $T_j = \begin{pmatrix} \overbrace{0 \quad . \quad 0}^{j-1+\lfloor j/(p-1)\rfloor} & 1 & 0 & . & . & . & . \\ \overbrace{0 \quad . \quad 0}^{j-1+\lfloor j/(p-1)\rfloor} & 0 & 1 & . & . & . & . \\ \overbrace{0 \quad . \quad 0}^{j-1+\lfloor j/(p-1)\rfloor} & \overbrace{0 \quad . \quad 0}^{31} & 1 & 0 & . & . & . \\ \overbrace{0 \quad . \quad 0}^{j-1+\lfloor j/(p-1)\rfloor} & \overbrace{0 \quad . \quad 0}^{31} & 0 & 1 & 0 & . & . \