# OpenReview forum: "Growing Tiny Networks: Spotting Expressivity Bottlenecks and Fixing Them Optimally"
_ICLR.cc/2024/Conference — Submitted to ICLR 2024_

### Official Review · Reviewer_qBe2 · 2023-10-31

**Soundness:** 3 good
**Presentation:** 3 good
**Contribution:** 4 excellent
**Rating:** 6
**Confidence:** 3

**Summary:**

This work proposes a method to augment network architectures by finding layers with an "expressivity bottlenecks" and widening the network there.

Specifically, they calculate the functional derivative $v_\text{goal} = - \nabla_{u = f(x)} \mathcal{L}(u)$ characterizing the best infinitesimal variation of the outputs of $f$ to decrease the loss at $x$. The derivative $v_\text{goal}$ is then compared with its projection on the tangent space $T_\mathcal{A}^{f_\theta}$ of the manifold $F_\mathcal{A}$ (networks with architecture $\mathcal{A}$) at the point $f_\theta$. The norm of the difference between these two directions is used to quantify the expressivity bottleneck. This gets also generalized not just for the logits $u$ but for all pre-activation values $a_l$.

The authors continue by providing a procedure how one calculates the best variation of the parameters for one layer $\delta W_l^*$ (Proposition 3.1) in order to calculate the expressivity gap $\Psi^l$, as well as a procedure to add neurons and initialize them optimally (Proposition 3.2). A series of Propositions (4.1 - 4.3) follow, shining light on the properties regarding the greediness of the approach.

The proposed approach is evaluated on the CIFAR-10 dataset. The authors start with an architecture consisting of two blocks of 2 convolutions and 2 MaxPooling each followed by two fully-connected layers using selu activation. The proposed method outperforms GradMax. The authors attribute this to the redundancy of GradMax.

Minor comments:
- in 2.2, paragraph "Optimal move direction" should $\Theta$ be $\Theta_\mathcal{A}$?
- Proof for 3.2 in the appendix: the first sentence seems incomplete

**Strengths:**

- The idea is interesting
- The approach appears to be sound, although the reviewer could not verify the proofs (maybe due to some misunderstandings, see more in the questions).
- Overall well written, although the technical reasoning should be improved.
- Mentioned Limitations where insightful.
- Helpful Appendix.

**Weaknesses:**

- The limitations regarding more complex datasets remains unclear.
- There don't seem to be many experiments: How does the method perform on other seed architectures?
- The math, especially the proofs should be more detailed.

**Questions:**

In no particular order:

- How does the approach compare to NEAT based techniques? What are differences / communalities?

- How well would your approach work on datasets with lower signal to noise ratio compared to CIFAR-10? Would you expect to see overfitting?
- How do you terminate the training procedure? Is there some schedule according to which you pick which layer gets widened? Do you pick the layer with the largest (normalized) expressivity gap?
- what exactly do they improve?
- Did you consider combination of standard gradient descent and your proposed method? How would they work out?

- I could not find a formal definition for $\partial / \partial t$ in Section 2.2
- Proposition 3.1 (Appendix):
	- What happens in the step where after "$M^+$, we get:"? To me it looks like you substituted $\delta W_l$ with $\delta W_l^*$ in $V_\text{goal}^l B_{l-1}^T = \delta W_l B_{l-1} B_{l-1}^T$, as this is where the gradient of $g(\delta W_l)$ vanishes, and then multiplied $\tfrac{1}{n} (B_{l-1} B_{l-1}^T)^+$ from the right. However, i am missing the reasoning why $B_{l-1} B_{l-1}^T \tfrac{1}{n} (B_{l-1} B_{l-1}^T)^+ = I$.
- Proposition 3.2:
	- As $S$ is just positive semi-definite ($S := \tfrac{1}{n} B_{l-2} B_{l-2}^T$) and not necessarily positive definite (i.e. may not have full rank), how do we know that $S^{-1/2}$ exists in Proposition 3.2?
	- Of which matrix are $\lambda_k$ the Eigenvalues? Currently i see that they are the singular values of $S^{-1/2} N$.
Overall, especially Proposition 3.1 and 3.2 would really benefit from detailed explanations.

- Does you method scale to ImageNet?
- Did you compare the three different initialization approaches? Random initialization, zero initialization and your in Proposition 3.2 proposed initialization?
- Which hardware did you use to run your experiments?

---

> ### Author Response · Authors · 2023-11-23
>
> $\newcommand{\mM}{\mathbf{\delta W}}$
> $\newcommand{\Ir}{\begin{pmatrix} I_r & 0 \\
>                 0 & 0\end{pmatrix}}$
>
> Thank you for your review and your good appreciation of our paper.
>
> To answer the questions:
> > How does the approach compare to NEAT based techniques? What are differences / communalities?
>
> Both approaches have the similar goals: the optimization of the architecture of a neural network, for a given machine learning task. However they proceed very differently. NEAT is based on evolutionary techniques, with a mix of exploration (random architecture changes), selection and cross-over. On the opposite we are driven by the gradient of the task loss, which enables us to directly know which part of the architecture we should change, and how (i.e. where to add neurons, and which ones). While NEAT needs to perform many random tries (and retrain each generated architecture) to find good moves, we exploit the information from the backpropagation to directly apply the optimal move. However our moves are optimal in a first order derivative meaning, and computing these moves is costly, while NEAT estimates the real impact of each tested move (but this comes at a computational cost as well, and arguably a higher one). We expect that in a near future, neural architecture optimization techniques involving both types of approaches will arise.
>
> > How well would your approach work on datasets with lower signal to noise ratio compared to CIFAR-10? Would you expect to see overfitting?
>
> Definitely, yes, if the user forgot to add a regularizer to the loss to optimize. We suppose that the user provides a loss that they actually want to optimize. This is not the case though in current practice of neural networks, as architectures are selected for known good training biases, and as batteries of indirect regularizers are used (early-stopping, drop-out, batch-norm, data augmentation, etc.). In case the user forgot to add a regularizer to the loss, we could still use these indirect regularizers to fight against overfit (except for early-stopping). We also have other ways to detect of prevent overfitting, as we can estimate the statistical significance of the candidate neurons to add. If the eigenvalue associated to the candidate neuron is too low, the performance gain provided by that neuron is likely to be due to spurious correlations, so we skip it. We detail this in Appendices D.1 and D.3. We actually plan to work further on the design of statistical tests.
>
> > _How do you terminate the training procedure? Is there some schedule according to which you pick which layer gets widened? Do you pick the layer with the largest (normalized) expressivity gap?_
>
> There are several possibilities to select the best layer to widen:
> a) as you suggest, compute the (normalized) expressivity gap for each layer and pick the largest one;
> b) compute also for each layer the best neurons to add, and compare the associated eigenvalues accross layers (as these eigenvalues indicate the first order variation of the loss when adding such neurons). Such a comparison might need special normalization though.
> c) for each layer, also perform line-search to find the best way to add the neurons, and then compare the real loss gains.
> As b) and c) have small computational cost compared to a), the last option is the most interesting, as it is the most reliable (being based on the real loss). This is what we have done for training the ResNet-18 architecture on CIFAR-100.
>
> Regarding termination, we stop when both conditions are met:
> 1) classical gradient descent is not improving performance anymore
> 2) no neuron is added anymore, that is, there is no way to add a neural anywhere to the architecture that would improve the loss.
>
> > what exactly do they improve?
>
> Could you please detail your question more?
>
> > _Did you consider combination of standard gradient descent and your proposed method? How would they work out?_
>
> Yes, this is what we do in practice, for instance with the new experiment on CIFAR-100: after each neuron addition we classically train the network for a certain time. This dramatically improves the results, compared to just adding neurons. However we noticed with a previous implementation that in some cases this would come with unstabilities due to the fact that different parts of the network might need different learning rates. These unstabilities have been tempered by better distributing the amplitude factor on added neurons: we now consider $(\sqrt{\varepsilon} \alpha, \sqrt{\varepsilon} \omega)$ instead of $(\alpha, \varepsilon \omega)$, which makes future gradients more homogeneous.

---

> > ### Author Response · Authors · 2023-11-23
> >
> > > _I could not find a formal definition for_ $\frac{\partial}{\partial t}$ _in Section 2.2_ ?
> >
> > This variable $t$ is the time of the gradient descent. Formally, one considers a family of functions $f_t$ indexed by $t$ that satisfies the Partial Derivative Equation  $\frac{\partial f}{\partial t}  = -\nabla_f \mathcal{L}(f)$, where $f: t,x \mapsto f_t(x)$ is seen as a single function of 2 variables. In practice after time discretization, this just means that we consider a series of updates defined as:
> > $f_{t+1} = f_t - \eta \nabla_f \mathcal{L}(f)$ for some learning rate $\eta$.
> >
> > > _Proposition 3.1 (Appendix):_
> >
> > Indeed $\frac{1}{n} B_{l-1} B_{l-1}^T (\frac{1}{n} B_{l-1} B_{l-1}^T)^+$ is not the identity matrix, but a projection on a subspace, whose dimension is the rank $r$ of the matrix $\frac{1}{n} B_{l-1} B_{l-1}^T$. Vectors in the image of $\frac{1}{n} B_{l-1} B_{l-1}^T$ are kept identical while the other ones are sent to 0.
> >
> > Step by step:
> > $$v_{goal}^{l}B_{l-1}^T = \mM B_{l-1}     B_{l-1}^T $$
> > $$\frac{1}{n} v_{goal}^{l}B_{l-1}^T = \mM \left( \frac{1}{n} B_{l-1}     B_{l-1}^T \right)$$
> > $$\frac{1}{n}v_{goal}^{l}B_{l-1}^T (\frac{1}{n}B_{l-1} B_{l-1}^T)^+ = \mM \left( \frac{1}{n} B_{l-1}     B_{l-1}^T \right)\left( \frac{1}{n} B_{l-1}     B_{l-1}^T \right)^+$$
> >
> > If $\left( \frac{1}{n} B_{l-1}     B_{l-1}^T \right)$ is full rank, then the projector $\left( \frac{1}{n} B_{l-1}     B_{l-1}^T \right)\left( \frac{1}{n} B_{l-1}     B_{l-1}^T \right)^+$ is the identity, which yields the solution.
> >
> > If it is not full rank, one can chose $\mM$ in the image of $\left( \frac{1}{n} B_{l-1}     B_{l-1}^T \right)$, and then we obtain the same formula (as the projection of $\mM$ is itself). Other solutions exist, by adding terms that are in the kernel of $\left( \frac{1}{n} B_{l-1}     B_{l-1}^T \right)$, i.e. whose image by the projection will be 0. The standard least square approach does not add such (useless) terms.
> >
> >
> > > _Proposition 3.2 (paper)_
> >
> > In Proposition $3.2$, the matrix $S^{-1/2} := U\sqrt{\Sigma}^{-1}U$ is defined with the convention that  $0^{-1} = 0$ when taking the inverse of the diagonal of $\sqrt{\Sigma}$. We added that information to the Proposition, that we indeed had forgotten to specify.
> >
> > The $\lambda_k$ are the singular values of $S^{-1/2}N$ and $\lambda_k^2$ are the eigenvalues of $S^{-1/2}NN^TS^{-1/2}$ and $NS^{-1}N^T$. We have updated the article by renaming everywhere the $\lambda_k$ "singular values", to avoid confusion.
> >
> >
> > > Does you method scale to ImageNet?
> >
> > We have performed a complexity analysis in the common reply ["Computational complexity"](https://openreview.net/forum?id=Qp33jnRKda&noteId=wMiGwiU7Bs) that we will also add as an appendix. When considering large images and a convolutional network trained by classical gradient descent, the additional cost of adding neurons with our method is actually small compared to the one of the gradient descent, so, yes, our method would scale to large images.
> >
> > Regarding scaling to larger architectures, as a proof of concept we have trained a ResNet-18 on CIFAR-100, which we are adding at the end of the appendix.
> >
> > > Did you compare the three different initialization approaches? Random initialization, zero initialization and your in Proposition 3.2 proposed initialization?
> >
> > Definitely. For 0 initialization, one cannot set both incoming and output weights $\alpha$ and $\omega$ of the new neuron to 0, otherwise gradient descent will never update them (two 0 in a row make gradient 0). If one sets only $\omega$ (or only $\alpha$) to 0, then the network will be able to learn.
> >
> > Now, if one adds the new neuron with small weights, whether they are random or well-chosen, the difference in training will be small, as all initializations with small weights are close to each other in the space of weights.
> >
> > However if one performs line-search to multiply the neuron by potentially a big factor, the difference is dramatic, and our approach performs much better. Indeed, from the theory of random projections, one can expect any random optimization direction to be able to improve the loss, but only marginally. The best improvements will be realized for directions that align with the loss landscape, i.e., at first order, the gradient ("optimal move") that we compute.
> >
> >
> > > Which hardware did you use to run your experiments?
> >
> > We usually use a standard GPU card (such as RTX2080), but some experiments are run on CPU only, when willing to estimate the full computational cost in practice.
> >
> > > Weaknesses
> >
> > We hope we have addressed the weaknesses you mentioned, in particular with the new experiment with ResNet-18 on CIFAR-100. Regarding proofs, we have added, as announced in the common reply ["New version of the paper, with improved clarity"](https://openreview.net/forum?id=Qp33jnRKda&noteId=7ZWDNGZNl9), several pages of proofs in the appendix, for the paper to be self-contained and to better explain the mathematical objects.

---

### Official Review · Reviewer_vNAT · 2023-10-31

**Soundness:** 3 good
**Presentation:** 2 fair
**Contribution:** 2 fair
**Rating:** 5
**Confidence:** 3

**Summary:**

TINY is proposed to grow neural network architectures with the aim to remove expressivity bottlenecks. The authors propose a scheme to increase the width of a considered feed-forward neural network architecture (with either fully-connected or convolutional layers) by adding neurons and thus increasing the width of the network during the growth process. (No dynamic addition of layers or other modules is considered.)
The proposed method is similar to GradMax but tries to avoid adding redundant neutrons.

**Strengths:**

- The authors work on a timely problem and try to reduce the computational requirements of deep learning by growing relatively small neural networks rather than training large ones from scratch.
- The authors aim to reduce redundancy in the addition of neurons when neural networks are grown.
- The proposal is theoretically motivated based on potential optimal additions of neurons in function space.
- The experiments show improvements in test accuracy over GradMax on CIFAR10.

**Weaknesses:**

- The exposition lacks a related literature discussion. While the introduction mentions different lines of research, it mostly focuses on early works in the related directions. Only Section 3 mentions a few related works on neural architecture growth and redundancy, which are easy to miss in the middle of the paper. As a result, an overview of the state of the art is missing.
- A similar criticism also holds for the experiment section, which only compares with GradMax and not different types of approaches.
- It is impossible to deduce from Section 3 what the actual algorithmic proposal is. The links in the algorithm to the supplement are broken. (The actual description is on page 15.) The actual update equations are not provided in the main paper.
- Limitation: It appears that the number of neutrons that are added in each step is a hyper-parameter.
- The update seems to involve a spectral decomposition to avoid neuron redundancy that is computationally costly.
- The computational complexity of the full training process (including the network growth) has not been evaluated, even though it forms in integral part of the claimed contributions.
- The method is not very flexible in adding layers or different kind of modules. It only grows the width of a chosen architecture.
- The novelty of the method appears to be limited in comparison with GradMax.
- Experiments are limited to CIFAR-10, a relatively small dataset of low complexity.
(Note that GradMax was evaluated also on CIFAR-100 and ImageNet.)
- The performance of the learned models on CIFAR-10 lacks far behind the test accuracy that can be achieved on this dataset with standard, relatively small ResNet architectures (like ResNet18).


Minor points:
- Broken figure link in Section 3 on page 4.
- The supplement is not included in the main paper so that important links are broken (see algorithm, for instance).

**Questions:**

- How does the proposed method perform on CIFAR100 and ImageNet?
- How is the matrix N on page 15 computed, since it depends on the (unknown?) $V_{goal}$?
- What is the runtime complexity of an update step?
- How do the computational requirements compare with training just a wider model from scratch once?
- How does training the obtained end neural network from scratch compare to the proposed training + growing process? Is a real improvement in generalization performance observed?

---

> ### Author Response · Authors · 2023-11-22
>
> Thank you for your detailed review and remarks that we answer below.
>
> ### Answers to questions
>
> > How does the proposed method perform on CIFAR100 and ImageNet?
>
> We explain in the common reply ["About experiments and architectures"](https://openreview.net/forum?id=Qp33jnRKda&noteId=bimw8pDivO) why we have not performed such tests, and that we are currently running them (on CIFAR-100 with a ResNet-style architecture), to show that our approach can scale to larger architectures and larger datasets. However we do not hope for competitive accuracy with the state-of-the-art as our current implementation does not include yet batch-norm, drop-out or data augmentation, which are needed to reach good scores.
>
> > How is the matrix N on page 15 computed, since it depends on the (unknown?) $V_{goal}$?
>
> $V_{goal}$ is actually known and easy to compute. We have rewritten the paper to make this more explicit (beginning of Section 2.2). The quantity $v_{goal}(x)$ is actually what the first step of the usual backpropagation computes, when the network has been shown sample $x$. So we get this quantity for free. We have added a detailed explanation in Appendix A.1 to show this property.
>
> > What is the runtime complexity of an update step?
>
> In practice, running our approach takes about the same time as training from scratch with standard gradient descent the final architecture we find.
>
> We detail the theoretical complexity in the common reply ["Computational complexity"](https://openreview.net/forum?id=Qp33jnRKda&noteId=wMiGwiU7Bs). In particular, we show that the computational cost of the SVD is neglectible relatively to standard gradient descent. This is because the matrices we apply SVD to are very small compared to other computations done to run the network (their size is the number of channels in a layer, and not  minibatch size, neither image size).
>
> > How do the computational requirements compare with training just a wider model from scratch once?
>
> This comparison is performed in Appendix E.1.2 COMPARISON WITH BIG MODEL. The left figure shows the accuracy as a function of time, while middle figure plots the accuracy as a function of gradient step. We easily see a persistent gap between the two curves on the left plot. While we achieve convergence around 90% accuracy on the test after only 15 seconds, the large models do not reach this performance even after 250 seconds of training.
>
>
>
> > How does training the obtained end neural network from scratch compare to the proposed training + growing process? Is a real improvement in generalization performance observed?
>
> This comparison is done in Appendix E.1.3 COMPARISON WITH THE SAME STRUCTURE RETRAINED FROM SCRATCH, where after obtaining an architecture with our method, we compare its performance with the same model randomly intialized and retrained from scratch. Regarding the left plot, which is a zoom of the final performances, the difference of performance is not significant (except for one model where the gap is 0.006 point of accuracy at our disadvantage). The take-away message is that we are able to obtain the same accuracy, in a single run with growing architecture, than the classical gradient descent run using an oracle to tell the best architecture, and this with similar training time. Consequently, we get architecture width tuning for free.
>
> ### Answers to other remarks
>
> > The method is not very flexible in adding layers or different kind of modules. It only grows the width of a chosen architecture.
>
> Our approach naturally extends to the addition of new layers (or nodes) in general computational graphs (any differentiable DAG), as it turns out that layer creation is the same as adding neurons to an empty layer. We discuss it briefly in the last paragraph of ["About experiments and architectures"](https://openreview.net/forum?id=Qp33jnRKda&noteId=bimw8pDivO).
>
> > Limitation: It appears that the number of neutrons that are added in each step is a hyper-parameter.
>
> As we discuss it in the common reply ["Neuron addition strategy"](https://openreview.net/forum?id=Qp33jnRKda&noteId=aJJH3Fi8jC), one can conceive neuron addition strategies that do really get rid of such hyper-parameters. In the paper we chose on purpose a very simple experimental setting (with fixed number of neuron additions) just to compare the quality of neurons added when using our approach or GradMax.
>
> > [About presentation]
>
> As advised, we moved the Related Works paragraphs from Section 3 to the rest of the state of the art in the main introduction. We have put the appendix back to the end of the main paper, for links to work again. We have added a high-level description of the algorithm in the main paper, and we are adding the description of the sub-routines in the Appendix, with detailed complexity.
>
> We hope we have answered your questions as well as adressed the weaknesses.

---

> > ### Comment · Reviewer_vNAT · 2023-11-22
> >
> > I thank the authors for their detailed response. At this point, I do not have any further questions.
> >
> > The project does not seem to be at a completed and fully polished state, yet, as multiple experiments are still running and a neuron addition strategy is not worked out yet. For that reason, I keep my score for now, but would be open to raising it during a discussion with the other reviewers.

---

> > > ### Author Response · Authors · 2023-11-22
> > >
> > > As we have stated, neuron addition strategies are out of the scope of this paper. We believe the paper is already sufficiently dense and that neuron addition strategies will deserve a paper (or series of papers) on their own. GradMax was published at ICLR 2022 and explicitly stated neuron addition strategies were out of the scope of their paper.
> > >
> > > We are still adding an appendix about experiments (we will submit a new version soon) but the main paper itself is already polished and in its final state. We ran supplementary experiments only because reviewers asked us to. We managed to run ResNet-18 on CIFAR-100, got a decent accuracy given the context (no drop-out, etc.) and compare very favorably to GradMax on all aspects (twice smaller network for same given accuracy, or 10 points of accuracy more for a same given size, while training faster). We hope this will be considered as sufficient validation for a theoretical paper.

---

> > > > ### Comment · Reviewer_vNAT · 2023-12-04
> > > > **Update of assessment**
> > > >
> > > > I thank the authors for the additional experiments and went through their results in detail.
> > > > To acknowledge their efforts, I have raised my score but still vote for rejection for the following reasons:
> > > >
> > > > 1. The experiments are conducted only in settings that achieve an extremely low accuracy that is far away from an acceptable range. This creates doubts regarding the application relevance of the presented results. In the small scale setting of the considered experiments (mostly on MNIST, CIFAR10, and one case of ResNet18 on CIFAR100), growing networks is irrelevant to save computations because it is no problem to train the well performing original models even on a laptop. Growing networks.
> > > > For a primarily theoretical paper, the theory would need to introduce more novel proof ideas or provide critical insights into general open questions. Furthemore, also theory needs experimental backup. But the performance of the learned models is far away from convincing.
> > > >
> > > > 2. Presentation: At least the appendix is still full of typos. A simple spell checker could alleviate most of this. Figure captions are not sufficiently informative. If one wants to determine the learning task of an experiment, one sometimes needs to follow links back to previous sections, which is inconvenient. Fig. 15 does not allow for a direct comparison between GradMax and TINY. The used red and green color scheme is unfriendly to color blind people.

---

### Official Review · Reviewer_XQVK · 2023-11-01

**Soundness:** 3 good
**Presentation:** 1 poor
**Contribution:** 3 good
**Rating:** 5
**Confidence:** 2

**Summary:**

This paper shows how to grow tiny networks by leveraging the functional gradient to optimize the network architecture on the fly. They define the expressivity bottlenecks by the distance between the desired activity update and the reachable update from the current parameter space. And they greedy reduce the expressivity bottlenecks during training when neurons are added.

**Strengths:**

The problem is interesting and well-defined mathematically. Theoretically, they show how to solve the problem and have solid propositions and proofs, although I did not follow most of them. Empirically they compared their method to the previous method showing that they achieve better accuracy on cifar10 when growing from a tiny model.

**Weaknesses:**

The paper is not easy to follow and there are a lot of typos in the paper, i.e., missing figure number in section 3, no caption for the algorithm. I do think we should have a main algorithm that describes the whole training process, like how function gradients are calculated and how the optimization problem is solved according to which proposition. Empirical results seems very limited even compared to the baseline methods, such as gradmax.

**Questions:**

How do we add neurons to the convolutional layers? Are we structurally adding kernels or adding neurons treating them as fully connected layers?

What are the benefits of the proposed method? Are we trying to have a method that tries to get the best model among a certain size or a method that can efficiently and effectively grow a tiny network to an arbitrary size? If it is the latter one, can we have some experiments with models that people use in practice?

What is the computational cost of the proposed methods?

---

> ### Author Response · Authors · 2023-11-22
>
> Thank you for your review and questions.
>
> ### About presentation
> We are very sorry for the typos, we hope we have corrected all of them in the revision, which we introduce in the common reply ["New version of the paper, with improved clarity"](https://openreview.net/forum?id=Qp33jnRKda&noteId=7ZWDNGZNl9). We have added a description of the main algorithm and are adding details of subroutines in an appendix. More generally we have reworked the paper to make it more clear and easier to follow.
>
> Calculating function gradients is actually very easy, it is part of what the standard backpropagation computes. We have added details in Section 2 and in Appendix A.1 to show this theoretically. We will also release the code (open-source). Technically one just has to call usual backpropagation and intercept an auxiliary computation, for instance with a ["hook"](https://pytorch.org/docs/stable/generated/torch.Tensor.register_hook.html) in the desired layer.
>
>
> > How do we add neurons to the convolutional layers? Are we structurally adding kernels or adding neurons treating them as fully connected layers?
>
> Adding one neuron to a convolutional layer $l$ consists in adding one kernel to layer 𝑙 and consequently also one dimension to each kernel at layer $l+1$. We added a scheme, in dedicated Appendix A.6, to make this more clear.
>
> > What are the benefits of the proposed method?
> > Are we trying to have a method that tries to get the best model among a certain size or a method that can efficiently and effectively grow a tiny network to an arbitrary size? If it is the latter one, can we have some experiments with models that people use in practice?
>
> Our goal is to build models of the right size by iteratively growing the architecture, starting from a tiny model, instead of training large models and pruning them.
>
> So we are interested in minimizing architecture size for a given target accuracy, or, equivalently, maximizing accuracy for a given size. We actually jointly optimize both size and accuracy.
>
> For this we also need indeed to be able to grow models to potentially arbitrary-large sizes, in case the task at hand requires it. We are right now training a ResNet-18 model (with initially very few neurons per block), on CIFAR-100, to check hows the architecture grows (cf common reply ["Experiments and architectures"](https://openreview.net/forum?id=Qp33jnRKda&noteId=bimw8pDivO)).
>
>
>
> > What is the computational cost of the proposed methods?
>
> We have addressed computational complexity in the common reply ["Computational complexity"](https://openreview.net/forum?id=Qp33jnRKda&noteId=wMiGwiU7Bs).

---

### Official Review · Reviewer_7xtb · 2023-11-10

**Soundness:** 1 poor
**Presentation:** 2 fair
**Contribution:** 2 fair
**Rating:** 5
**Confidence:** 4

**Summary:**

The submission presents a novel method to increase the width of a network during optimization, inspired from a functional argument. The method starts from the gradient of the loss wrt the output of the network, and finds weights by trying to align the output to this desired change.

**Strengths:**

The method is novel and the problem of fitting both the weights and the architecture at the same time is relevant and very much open. I have heard the idea of "let's do gradient descent on the architecture" multiple time, but little in the way of actual attempts to define what is meant by that statement, which is welcome.

**Weaknesses:**

$\newcommand{\R}{\mathbb{R}}$$\newcommand{\L}{\mathcal{L}}$

The paper presents an interesting idea and I am generally positive towards it, but I found it hard to get the intended message. I think it would greatly benefit from an update to improve the clarity of the message, especially on the following points
- The presentation of the functional analysis viewpoint. I found it hard to follow, probably due to notation overload.
- The submission seems to not directly address how to trade-off increasing number of parameters vs. updating the parameters we already have.
- Given that a stated contribution of the submission is a definition of optimality, what is meant by that should be stated explicitly.
- Some statements should be made more carefully to avoid overly general claims.

I give more details and specific examples of each of the points below. I will increase my score if these points are adressed by a revision during the discussion period.

As my issues have to do with clarity, I tried giving specific and clear descriptions, leading to a possibly (overly long) review. The length of the section below should not be taken as a negative assesment of the submission. My hopes is that those can help the authors make the message of the paper clearer.

---

## Details

**Clarity of the functional view**

I might be missing some key background reference, but I struggle to follow section 2.2. My understanding of the high-level idea is that $v_{\text{goal}}(x)$ indicates the desired change in output of the network by indicating what infinitesimal change in the output of the network is desired. This goal reasonable and I wouldn't have an issue if it had been stated as such, but I don't understand how it follows from the functional perspective outlined in §2.2.

The notation seems overloaded to represent the functional and the "standard" ML notation. The lack of distinction makes the text hard to parse. For example, the expression $\nabla_{f}\L(f)$ implies that $\L$ takes a function, but in $\nabla_{u=f(x)} L(u)$, it is evaluated at the output of the network, a vector in $\R^p$.

The text also implies that $\nabla_f \L(f) = \nabla_{u=f(x)} \L(u)$" by the definition and evaluation of $v_{\text{goal}}$. Assuming my interpretation above is correct, this equivalence is not obvious to me. It would benefit from an explanation as to why it holds, or a reference. It is also unclear to me why this holds without defining the space of functions, for example whether $\mathcal{F}$ is the class of function representable by any width-$M$ networks and taking the union over all $M$s, some RKHS, or whether we any arbitrary pathologic discontinuous functions is allowed.

**Balancing optimization and adding parameters**

The last contribution states that the submission "naturally obtain[s] a series of compromises between performance and number of neurons, in a single run, thus removing the need for width hyper-optimization". I would expect this contribution to refer to a particular result highlighting how the proposed methods trades-off (a) fitting the current architecture/doing more traditional GD steps vs. (b) adding neurons. Unless I missed something, the proposed method does not inherently address this tradeoff, and instead adds a fixed number of neurons. This seems to be replacing the width hyperparameter by a "how-much-width-to-add" hyperparameter. The method can still be an improvement by lowering the dependency of the performance on the hyper-parameter, but should be discussed more directly in the main text.

**Definition of optimality**

The submission uses the term optimal in many instances with what appears to be different meanings. It is not clear what criteria is used to establish optimality. To take an example from optimization, gradient descent being optimal could refer to the fact that it is the result of minimizing a surrogate quadratic problem, or to say that it attains the best rate of convergence among first-order algorithms in some problem class.

As the goal of the submission is to "mathematically define the best possible neurons" and fixing expressivity bottlenecks "optimally", it would be beneficial to be explicit about what is meant by "optimal". Especially as the submission can be interpreted as proposing two definitions; one implied in §2.2 as minimizing the distance between $v_{\text{goal}}$ and the actual update, and another looking at the layers independently in 2.3 to make the problem tractable (especially as the submission states in §3.3 that "this move is sub-optimal").

For example, "picking optimal directions that avoid redundancy in the pre-activation space" at the end of the submission seems to reflect that "optimal" is taken to mean the optimal direction to decrease the first-order approximation of the loss, a concept that is missing from other instances such as "Optimal functional move", "The optimal update of the weights at a given layer", or the optimality in Prop 3.2.

**Overly broad claims**

- (Intro) "This removes the optimization issues (local minima) that usually arise when considering thin architectures"; "remove optimization issues" is too broad, and might imply that local minima are the only optimization problem. The contribution should state that it is possible to avoid some local minima (with a forward reference to the specific result in §4), or specify that this result applies to 1-hidden-layer networks.
- (§3.2) "[adding random neurons] would not yield any guarantee regarding the impact on the system loss"; I read this sentence as implying that this is in contrast with the proposed method, which then should have a guarantee that adding the proposed neurons decreases the loss. As no such results are presented, the description should be changed.
- Going into §2.3, I interpreted the description of "recursive" as implying that some invariant would be maintained, and specifically that the resulting update wouldn't change. To avoid this confusion, I would suggest being explicit are the start of §2.3 that what follows is a an approximation to what is desired in §2.2, as this only spelled out in §3.1.


**Related work**
The description of prior work could be more detailled to help readers unfamiliar with the literature. For example, it is not clear how the description of Net2Net, AdaptNet and MorphNet ("propose different strategies to explore possible variations of a given architecture") differs from the approach proposed here.

I was also surprised to not see a citation to the classical works of neuron boosting/incrementally learning a neural network one neuron at a time (For example, Bengio, Le Roux, Vincent, Delalleau, and Marcotte. Convex neural networks. 2006, or other references found in the GradMax paper of Evci et al.), which I think would be relevant for historical context.

Although focused on optimization of a fixed architecture, there is a line work in optimization for deep learning that takes a constrained optimization/Lagrangian view to obtain per-layer updates that look similar to the recursion argument in §2.3--§3.1. The following works might be of interest to the authors if they were previously unaware of those. _(to be explicit; although I do believe there is a connection and that some discussion could be beneficial, I am not requesting that the submission cite those works)_
- Lecun. A Theoretical Framework for Back-Propagation. Proceedings of the 1988 connectionist summer school
- Carreira-Perpiñán and Wang. Distributed optimization of deeply nested systems. AISTATS 2014.
- Taylor, Burmester, Xu, Singh, Patel and Goldstein. Training Neural Networks Without Gradients: A Scalable ADMM Approach. ICML 2016.
- Frerix, Mollenhoff, Moeller and Cremers. Proximal Backpropagation. ICLR 2018.
- Amid, Anil and Warmuth: LocoProp: Enhancing BackProp via Local Loss Optimization, AISTATS 2022

---

**Minor points (no need for a response):**
- "Under standard weak assumptions (A.1)" made me think I should look for a an "Assumption 1", as this style of reference is common. I'd suggest spelling out "(see Appendix A.1)".
- (after Prop 4.3) "by requiring the added neuron to have infinitely small input weights"; a literal interpretation of this sentence requires the inputs to be 0. I suggest rephrasing in terms of "direction" instead.
- What "time" in $\frac{\partial\theta}{\partial t}$ is not defined,
- "shown empirically to be better optimized than small ones Jacot et al. 2018" seems to imply that this is what Jacot et al. shown. I assume the citation should have been moved earlier in the sentence, for the theoretical part.
- There are multiple instances where \citet and \citep are mixed, leading to missing parentheses around citations, especially in paragraphs discussing related works ("Notions of expressivity" paragraph)
- The "amplitude factor $\gamma$" used in the Algorithm given in Fig. 6 seems undefined in the main paper.

**Questions:**

The specifics of my points above do not require a response and can instead be adressed through a revision, although I am open to a discussion if the authors disagree with my comments.

For specific questions;

- **Clarity of the functional view**

  Are the notation issues identified above correct, or did I completely miss something? If so, what is the formal definitions of the objects used, and why is the functional gradient the same as the derivative wrt the output of the network?
  (Those questions be adressed by a revision to the paper and need not be written in openreview posts)

- **Balancing optimization and adding parameters**

  Shouldn't the functional view provide a way to perform this trade-off, for example through some regularization parameter that could impact how much better the "adding new weights" step should be vs. updating existing weights?

- **Complexity of the methods**

  The introduction claims that the method is is "competitive" in computational complexity with standard training. However, it seems that the methods requires the computation of SVDs of matrices whose size dependent on the width of the network, and the complexity should scale (at least) with that width squared, which is much more than gradient descent. Could the authors clarify what was meant?

---

> ### Author Response · Authors · 2023-11-22
>
> Thank you a lot for your review and the raised points, which incited us to rework some parts of the paper and to add important details to the appendix.
>
> In particular:
> > Clarity of the functional view
>
> We apologize for the notation overload. We have corrected this, notably by distinguishing the sample-loss $\ell$ from the global loss $\mathcal{L}$, and by correcting a few typos.
> As stated in the common reply ["New version of the paper, with improved clarity"](https://openreview.net/forum?id=Qp33jnRKda&noteId=7ZWDNGZNl9), we have added Appendix A.1 to get the paper self-contained and to explain what precisely is the functional gradient, as well as how to obtain the property $${v_{goal}(x)} = -\left( \nabla_f \mathcal{L}(f) \right)(x) = -\nabla_{u} \ell(u, y(x))\big|_{u = f(x)}$$
> which is not trivial indeed. We hope that with the new version of the paper we have answered all of your concerns regarding the mathematical soundness of the paper.
>
> > Balancing optimization and adding parameters
>
> This is of course a very important question. We provide a detailed answer in the common reply ["Neuron addition strategy"](https://openreview.net/forum?id=Qp33jnRKda&noteId=aJJH3Fi8jC), that we will also add to the paper as an appendix.
>
> To answer the remark that the method "adds a fixed number of neurons", note that the algorithm (described in "Algorithm 1", page 9) for that experiment is actually more complex. A fixed number of candidate neurons are proposed indeed, but two selection steps take place afterwards: 1) a selection by statistical relevance based on the singular values (detection of spurious correlations, to avoid overfit), 2) a selection by performance gain (only consider neurons that do improve the loss on a validation set). Thus the number of added neurons may vary.
>
> The number of neuron candidates is fixed in that experiment in order to compare with GradMax, but one can instead select this number in an adaptive manner based on the singular values (which are linked to expected loss improvement as stated by Proposition 3.2). A simple threshold on singular values enables indeed to make layers grow only when needed / useful. In that respect architecture growth can be not scheduled in advance but happen only on demand.
>
> > Complexity of the methods
>
> We have addressed this point in the common reply ["Computational complexity"](https://openreview.net/forum?id=Qp33jnRKda&noteId=wMiGwiU7Bs), which is being included in a dedicated appendix as well. TL;DR: SVD cost is actually low.
>
> > Definition of optimality
>
> Thank you for noticing this. Indeed we were using the same word "optimal" for different meanings. We have added a paragraph in Section 2 to make this more clear, and this was the opportunity to better introduce the criterion considered in the next sections. A supplementary appendix (A.3) details these concepts and their intertwinement.
>
> > Misc.
>
> Thank you for the minor points and noticing formulations leading to "overly broad claims", that we have addressed.
> Regarding  "[adding random neurons] would not yield any guarantee regarding the impact on the system loss" however, what we mean is that our method yields the best neuron to add to decrease the loss at first order (in a sense now well defined), while a random neuron does not have this property.
>
> Thank you also for the references, which we did not know and are very interesting and indeed related to our work. We will discuss them, in an appendix section though, as space is too limited in the main paper, especially with the imposed bibliographic citation style. Speaking of which:
>
> > I have heard the idea of "let's do gradient descent on the architecture" multiple time,
>
> Are you referring to DARTS and to the Lottery ticket hypothesis, or to other papers? in which case, we would be delighted if you could provide references.

---

> > ### Comment · Reviewer_7xtb · 2023-11-23
> >
> > Thanks for your response.
> >
> > Given the short time-frame, I did not have time to go through the entirety of the response and paper update yet, but am leaving this message before the end of the discussion period.
> >
> > **Clarity of the functional view**
> > From a short look, §2 looks much better, but will need more time to go through it again.
> >
> > **Balancing optimization and adding parameters**
> > The response claims that a "neuron addition strategy" is out of scope, but I would argue that the last contribution ("naturally obtain a series of compromises between performance and number of neurons, in a single run, thus removing the need for layer width hyper-optimization") is putting "how to add neurons" in scope. My understanding of this point is as a claim that the proposed methodology provides a way to measure the benefit of adding neurons vs. further optimizing the current ones. If this is not the case, and the core contribution is limited in scope to "how to add a neuron when we need one", this point needs clarification.
> >
> > I understand that the submission has a more subtle selection procedure, but it seems ad-hoc and disconnected from the formalism proposed in the earlier sections. This issue might be addressed by making the contribution point more explicit.
> >
> > **Misc**
> >
> > > Thank you also for the references
> >
> > I probably missed some relevant papers, and would recommend a deeper dive if the authors are interested (in my view, this is not strictly necessary for the current submission). Per-layer updates show up regularly in different contexts, and the  connection between those methods are often not obvious, leading to a very disconnected citation graph.
> >
> > > Are you referring to DARTS and to the Lottery ticket hypothesis, or to other papers?
> >
> > I was referring to informal conversations with colleagues at conferences/workshops, where I've heard the idea floated at the level of "it would be great _if_ we could GD on the architecture". DARTS or the RL-based methods referenced in the submission are the closest match I know.

---

> > > ### Author Response · Authors · 2023-11-23
> > >
> > > > My understanding of this point is as a claim that the proposed methodology provides a way to measure the benefit of adding neurons vs. further optimizing the current ones.
> > >
> > > Indeed we do not provide this (we could, actually, by comparing the norm of the classical gradient with the performance gain associated to the candidate neurons, but we did not do it in the paper). We understand your point now.
> > >
> > > > This issue might be addressed by making the contribution point more explicit.
> > >
> > > Ok, we rephrase the last contribution as:
> > > "We automatically adapt the architecture to the task at hand by making it grow where needed, and this in a single run, in competitive computational complexity with respect to classically training a large model just once. To remove any need for layer width hyper-optimization, one could define a target accuracy and stop adding neurons when it is reached."

---

### Author Response · Authors · 2023-11-21
**New version of the paper, with improved clarity**

First, we would like to thank the reviewers for the quality of their reviews.
Second, we would like to apologize for the delay in answering: we wanted to finish the revision of the article first, but as time is ticking, we upload a new version, yet not final, and start replying.

### Changes in this new version:
In order to improve clarity and presentation, we significantly reworked the article; in particular:
- we improved notations, in particular we now distinguish the sample-loss $\ell$ from the overall loss $\mathcal{L}$ to avoid notation overload and confusion. We also corrected typos and in particular missing parentheses for citations (we are deeply sorry for this; as our pdf viewer displays boxes around links, parentheses around citations or their absence were not very visible).
- we improved presentation by removing some technical details (postponed to appendix) and moving the "Related work" paragraphs from Section 3 to the introduction.
- we added a detailed appendix (A.1) to properly introduce the functional gradient. The article is now self-contained in that respect. Furthermore we show how to prove that:
$${v_{goal}(x)} = -\left( \nabla_f \mathcal{L}(f) \right)(x) = -\nabla_{u} \ell(u, y(x))\big|_{u = f(x)}$$ This is not trivial and could actually be turned into a proposition.
- we added a paragraph in Section 2.2 in order to explain the links between gradients, closeness to the functional update, and loss optimization. This yields a natural criterion to optimize w.r.t. $\delta\theta$, makes the connexion with proximal gradients, and then Section 3 comes more naturally. In particular this clarifies in which sense the moves are *optimal*.
- To prove the assertions in this new paragraph, we added a detailed appendix (A.3), reminding the definition of "gradient" and showing why the projection of the functional gradient onto the tangent space $\mathcal{T}_\mathcal{A}$ is optimal for the loss at first order: it is a parametric gradient for a particular, natural inner product.
- we moved the appendix to the main pdf, for links to work.

We still have to:
- add an appendix about detailed full algorithm
- as well as its complexity
- add an appendix about links with literature on per-layer updates
- add an appendix to discuss about the propositions in part 3 and explain their consequences.

---

> ### Author Response · Authors · 2023-11-21
> **Experiments and architectures**
>
> ### About experiments and architectures
>
> We agree that the experiments in the paper may seem to be limited, in that the datasets are relatively small (MNIST, CIFAR-10) and the architectures are not very deep. This said, a 6-layer network, with 4 convolutional layers and 2 fully-connected ones, trained on CIFAR-10 (which is much more varied than MNIST), starts being a good proof of concept (it is not just a 1- or 2-hidden layer MLP as in many theoretical articles), and is sufficient to see a significant difference between our approach and GradMax in terms of performance/complexity ratio.
>
> To show that our approach can scale up, we are currently trying to train a ResNet-18 architecture on CIFAR-10 and on CIFAR-100, as advised by several reviewers. However for this we first need to adapt our code. Besides, though there is no theoretical difficulty, we have not yet implemented our approach for layers such as batch-norm or drop-out, and we are training with a standard setting, without data augmentation or transfer learning for instance, and will have no time till the deadline for standard hyper-parameter tuning (such as learning rate, learning schedule, momentum, etc.). Given that context, we do not expect to meet top ResNet accuracies that can be found in the literature. We can still compare with GradMax in the same setting as ours. We expect same final accuracy, slightly slower training (as GradMax is precisely designed to learn as fast as possible, and both approaches have similar computational complexity), and smaller final architecture (since our approach is designed to avoid redundancy while GradMax is not). Thus we expect a similar Pareto front as in Figure 5. It will be interesting to compare the number of parameters at same accuracy and we are eager to check how much we are able to reduce computational complexity for a same accuracy performance.
>
> The reason why we have not performed such large-scale experiments before is that intensive experimental validation of the current approach does not match our line of research in that it is too early. Indeed, we have been working on extending our approach to the addition of new layers (or nodes) in general computational graphs (any differentiable DAG), as it turns out that layer creation is the same as adding neurons to an empty layer. The theoretical extension is straightforward but the implementation is more tricky, and graph algorithmics have to be considered. We feel that it makes more sense to spend time on experiments once the whole architecture can evolve freely rather than when it is constrained as it is now to a fixed graph (with only layer growth).
> The paper submitted here is more about the mathematical foundations of the approach (quantifying expressivity bottlenecks, solving them, etc.).

---

> ### Author Response · Authors · 2023-11-21
> **Neuron addition strategy**
>
> ### About neuron addition strategy
>
> #### Scope of the paper
> The precise strategy to add neurons (how many, how frequently, etc.) is out of the scope of this paper, in the same way that is explicitly out of the scope of the GradMax paper. We chose on purpose a naive strategy for this paper, in order to focus on the way we compute new neurons, and not on the evaluation of the strategy. This way we are able to check that even in very simple settings, step by step, GradMax produces larger architectures that ours for a similar performance gain (and this is expected, as the two approaches do not have the same goal).
>
> #### Yet, some examples of strategies
> However for daily use our implementation does of course include less naive strategies, when considering candidate neurons to be added, based on:
> - statistical significance, as spurious correlations can trick the estimation of singular values. We therefore threshold singular values in a theoretically appropriate manner.
> - a compromise between performance gain (in terms of loss variation) and added computational complexity (which is related to the number of added parameters). Before adding a neuron, we check that this ratio is above a certain threshold. This can be seen as a modification of the general loss $\mathcal{L}$, to which a criterion "$+ \mu$ Complexity" would be added, similarly to the spirit of Kolmogorov complexity or Minimum Description Length paradigm, except that this is about computational complexity. Fortunately, the hyper-parameter $\mu$ does not need to be chosen: one could start training with a very high value of it, and decrease it progressively each time the architecture growth gets stuck. This way one would obtain a kind of Pareto front between performance and complexity (with a single run!), and the user would just have to select the compromise they prefer *afterwards*. In this sense, our approach does get rid of all architecture hyper-parameters.
>
> #### Mixing neuron addition with gradient descent
> Regarding how to mix standard gradient descent with neuron addition:
> - a) one extreme is to only add neurons, without any form of gradient descent (while still relying on backpropagation); surprisingly, this works not so bad;
> - b) a more flexible way is train classically and sometimes add neurons (according to some schedule that itself might depend on last expressivity bottleneck estimations);
> - c) an orthogonal option, that can be used with either of the above, is to make use of the optimal move (kind of natural gradient) which is a by-product of neuron estimation, hence bringing a supplementary performance boost after each neuron addition.
>
> Method c) brings flexibility to a), in that old neurons can be updated.
> If one adds new neurons after a line search on amplitude factor (as in the paper), then each neuron addition brings a significant change in the network, and combinations involving b) have the potential issue that several different learning rates might be needed at the same time by different parts of the network (as some are nearly converged and some are disturbed by recent neuron additions), in which case one has to be careful with learning rates. Method c) does not have this issue, as an amplitude factor line search is performed on these optimal updates as well.
> On the opposite if one adds neurons with 0 output weights (or 0 input weights), then the point of using a)+b) is mostly to know when and where to add neurons: the difference between various ways to choose neuron input (or output, resp.) weights, such as random initialization, GradMax, or ours, is much less noticeable, as it will be quickly overwritten by standard gradient descent.
>
> #### About overfit
> We assume that the loss $\mathcal{L}$ to be minimized is the real loss that the user wants to optimize, possibly including regularizers to avoid overfitting.
> In the absence of regularizer indeed, our method applied without safeguards (such as thresholds depending on statistical significance and/or added computational complexity vs. performance gain) should fully optimize the loss and as such completely overfit.
> Our safeguards are expected to prevent strong overfitting as they actually are regularizers.

---

> ### Author Response · Authors · 2023-11-21
> **Computational complexity**
>
> ### About Computational complexity
>
> The algorithm for architecture growing, when considering a naive strategy of the type a)+b)+c), is:
>
>     For a certain number of times (typically, number of epochs/T):
>       Train classically for T gradient descent steps, using minibatches of size M;
>       For each layer:
>         Compute the first K neurons to add at that layer, using a minibatch of size M' (1);
>         Select the neurons with significant eigenvalue ;
>         Compute the amplitude factor for the new neuron, using another minibatch, of size M" (2);
>       Add the best neurons of the best layer;
>       For each layer:
>         Update its weights using the optimal move (3);
>
>
> using in plus the following notations:
> - number of layers: $L$
> - layer width: $W$ (assuming for simplicity that all layers have same width), number of kernels if convolutions
> - number of pixels in the image: $P$ ($P=1$ for fully-connected)
> - kernel filter size: $S$ ($S=1$ if fully-connected)
>
>
> Complexity:
> - of the standard training part: $T M L W^2 S P$
> - of step (1):
>    - computation of matrices of interest: $LM' (SW)^2 P$
>    - SVD cost: $L (SW)^3$
> - of step (2): $L^2 M'' W^2 S P$
> - of step (3): $L S W$
>
> The relative added complexity w.r.t. the standard training part is thus:
> $M'S/TM +S^2W/TMP + M''L/TM + 1/WTMP$.
>
> ### SVD cost is neglectible
> The relative cost of the SVD w.r.t. the standard training part is $S^2W/TMP$.
> In the fully-connected network case, S=1, P=1, and the relative cost of the SVD is then W/TM. It is then neglectible, as layer width W is usually much smaller than TM, which is typically 10x100 for instance.
> In the convolutional case, S=9 for 3x3 kernels, and $P\approx 1000$ for CIFAR, $P\approx 100000$ for ImageNet, so the SVD cost is neglectible as long as layer width W << 10000 or 1 000 000 respectively. So one needs no worrying about SVD cost.
>
> Likewise, the update of existing weights using the "optimal move" (already computed as a by-product) is computationally neglectible, and the relative cost of the line searches is limited as long as the network is not extremely deep (L < T M/M").
>
> On the opposite, the estimation of the matrices (to which SVD is applied) can be more ressource demanding. The factor M'S/TM can be large if the minibatch size M' needs to be large for statistical significance reasons. One can show that an upper bound to the value required for M'  to ensure estimator precision (details to come soon in Apprendix) is $(SW)^2/P$. In that case, if $W > \sqrt{TMP/S^3}$, these matrix estimations will get costly. In the fully-connected network case, this means $W > \sqrt{TM} \approx 30$ for T=10 and M=100. In the convolutional case, this means $W > \sqrt{TMP/S^3} \approx 30$ for CIFAR and $\approx 300$ for ImageNet. We are working on finer variance estimation and on other types of estimators to decrease M' and consequently this cost. Actually $(SW)^2/P$ is just an upper bound on the value required for M', which might be much lower, depending on the rank of computed matrices.
>
> ### In practice
> In practice the cost of a full training with our architecture growth approach is similar (sometimes a bit faster, sometimes a bit slower) than a standard gradient descent training using the final architecture from scratch. This is great as the right comparison should take into account the number of different architectures to try with the classical neural architecture search approach. Therefore we get layer width hyper-optimization for free.

---

> ### Author Response · Authors · 2023-11-23
> **New revision**
>
> We have added:
> - Appendix D.1 about how to pick a suitable minibatch size for estimating matrices of interest, based on estimator variance
> - Appendix D.5 with the full algorithm and subroutines
> - Appendix D.6 about computational complexity
> - Appendix E.3 about experiments with ResNet-18 on CIFAR-100. Our method learns slightly faster than GradMax (which is a surprise, as neuron addition in GradMax is designed to learn as fast as possible), and this with final architectures that are significantly thinner (-30%). This shows that our method does succeed in avoiding redundancy.

---

### Meta-Review · Area_Chair_v9Xa · 2023-12-24

**Metareview:**

The paper introduces TINY, a method that widens layers by identifying expressivity bottlenecks. The authors present a systematic approach to detect and quantify these bottlenecks and provide a mechanism for expanding networks. While reviewers appreciate the novel perspective on exprssivity, reviewers think the current experiments are too small to substantiate the theoretical claims.

The authors argue that as a proof of concept, a 6-layer CNN on CIFAR-10 is enough, and to achieve good results on CIFAR100 or ImageNet, extra techniques like BatchNorm and data augmentation are needed. In contrast, the reviewers think CIFAR-100 with ResNet18, or even ImageNet (as GradMax has done) is a minimum to justify the proposed method. I can understand why theoreticians think 6-layer CNN on CIFAR-10 is enoughm, as many theoretical papers only analyze 3-lalyer FCN. However, I think in ICLR, ResNet18 on CIFAR100 with a good test accuracy is a somewhat standard requirement. In addition, training techniques like BatchNorm is too standard to ignore in the experiments. The current shape of the paper might be sufficient in a more theoretical venue like COLT, but in venues like ICLR, I support the reviewers' opinions on CIFAR100 experiments (or even TinyImageNet experiment).

Additionally, presentation issues like numerous typos and unclear figures hinder the readability of the paper. Overall, despite recognizing the effort in conducting additional experiments, more comprehensive experiments are still needed to demonstrate the proposed method's efficacy. I recommend rejection.

**Justification For Why Not Higher Score:**

Experiments on a 6-layer CIFAR10 is not enough to justify the proposed method.

**Justification For Why Not Lower Score:**

N/A

---

### Decision · Program_Chairs · 2024-01-16

Reject